# SGD: The Role of Implicit Regularization, Batch-size and Multiple Epochs

**Satyen Kale**
Google Research, NY
satyen@google.com

**Ayush Sekhari**
Cornell University
as3663@cornell.edu

**Karthik Sridharan**
Cornell University
ks999@cornell.edu

## Abstract

Multi-epoch, small-batch, Stochastic Gradient Descent (SGD) has been the method of choice for learning with large over-parameterized models. A popular theory for explaining why SGD works well in practice is that the algorithm has an implicit regularization that biases its output towards a good solution. Perhaps the theoretically most well understood learning setting for SGD is that of Stochastic Convex Optimization (SCO), where it is well known that SGD learns at a rate of $O(1/\sqrt{n})$, where $n$ is the number of samples. In this paper, we consider the problem of SCO and explore the role of implicit regularization, batch size and multiple epochs for SGD. Our main contributions are threefold:

1. We show that for any regularizer, there is an SCO problem for which Regularized Empirical Risk Minimzation fails to learn. This automatically rules out any implicit regularization based explanation for the success of SGD.

2. We provide a separation between SGD and learning via Gradient Descent on empirical loss (GD) in terms of sample complexity. We show that there is an SCO problem such that GD with any step size and number of iterations can only learn at a suboptimal rate: at least $\widetilde{\Omega}(1/n^{5/12})$.

3. We present a multi-epoch variant of SGD commonly used in practice. We prove that this algorithm is at least as good as single pass SGD in the worst case. However, for certain SCO problems, taking multiple passes over the dataset can significantly outperform single pass SGD.

We extend our results to the general learning setting by showing a problem which is learnable for any data distribution, and for this problem, SGD is strictly better than RERM for any regularization function. We conclude by discussing the implications of our results for deep learning, and show a separation between SGD and ERM for two layer diagonal neural networks.

## 1 Introduction

We consider the problem of stochastic optimization of the form:

$$\text{Minimize } F(w) \tag{1}$$

where the objective $F : \mathbb{R}^d \mapsto \mathbb{R}$ is given by $F(w) = \mathbb{E}_{z \sim D}[f(w; z)]$. The goal is to perform the minimization based only on samples $S = \{z_1, \ldots, z_n\}$ drawn i.i.d. from some distribution $\mathcal{D}$. Standard statistical learning problems can be cast as stochastic optimization problems, with $F(w)$ being the population loss and $f(w, z)$ being the instantaneous loss on sample $z$ for the model $w$.

**Stochastic Convex Optimization (SCO).** Perhaps the most well studied stochastic optimization problem is that of SCO. We define a problem to be an instance of a SCO problem if,

$$\textbf{Assumption I}: \text{Population loss } F \text{ is convex.} \tag{2}$$

35th Conference on Neural Information Processing Systems (NeurIPS 2021).

Notice above that we only require the population loss to be convex and do not impose such a condition on the instantaneous loss functions $f$.

Algorithms like Stochastic Gradient Descent (SGD), Gradient Descent on training loss (GD), and methods like Regularized Empirical Risk Minimization (RERM) that minimize training loss with additional penalty in the form of a regularizer are all popular choices of algorithms used to solve the above problem and have been analyzed theoretically for various settings of Stochastic Optimization problems (convex and non-convex). We discuss below a mix of recent empirical and theoretical insights about SGD algorithms that provide motivation for this work.

**SGD and Implicit Regularization.** A popular theory for why SGD generalizes so well when used on large over-parameterized models has been that of implicit regularization. It has been oberved that in large models, often there are multiple global minima for the empirical loss. However not all of these empirical minima have low suboptimality at the population level. SGD when used as the training algorithm often seems to find empirical (near) global minima that also generalize well and have low test loss. Hence while a general Empirical Risk Minimization (ERM) algorithm might fail, the implicit bias of SGD seems to yield a well-generalizing ERM. The idea behind implicit regularization is that the solution of SGD is equivalent to the solution of a Regularized Empirical Risk Minimizer (RERM) for an appropriate implicit regularizer.

The idea of implicit regularization of SGD has been extensively studied in recent years. In Gunasekar et al. [2018a], Zou et al. [2021], the classical setting of linear regression (with square loss) is considered and it was shown that when considering over-parameterized setting, the SGD algorithm is equivalent to fitting with a linear predictor with the smallest euclidean norm. In Soudry et al. [2018], Ji and Telgarsky [2018] linear predictors with logistic loss are considered and it was noted that SGD for this setting can be seen as having an implicit regularization of $\ell_2$ norm. Gunasekar et al. [2018b] considered multi-layer convolutional networks with linear activation are considered and showned that SGD for this model can be seen as having an implicit regularization of $\ell_{2/L}$ norm (bridge penality for depth $L$ networks) of the Fourier frequencies corresponding to the linear predictor. Gunasekar et al. [2018c] considered matrix factorization and showed that running SGD is equivalent to having a nuclear norm based regularizer. More recent work of Arora et al. [2019], Razin and Cohen [2020] shows that in particular in the deep matrix factorization setting, SGD cannot be seen as having any norm based implicit regularizer but rather a rank based one. However, in all these cases, the behavior of SGD corresponds to regularizers that are independent of the training data (e.g. rank, $l_p$-norm, etc).

One could surmise that a grand program for this line of research is that for problems where SGD works well, perhaps there is a corresponding implicit regularization explanation. That is, there exists a regularizer $R$ such that SGD can been seen as performing exact or approximate RERM with respect to this regularizer. In fact, one can ask this question specific to SCO problems. That is, for the problem of SCO, is there an implicit regularizer $R$ such that SGD can be seen as performing approximate RERM? In fact, a more basic question one can ask: is it true that SCO problem is always learnable using some regularized ERM algorithms? *We answer both these questions in the negative.*

**SGD vs GD: Smaller the Batch-size Better the Generalization.** It has been observed that in practice, while SGD with small batch size, and performing gradient descent (GD) with empirical loss as the objective function both minimize the training error equally well, the SGD solution generalizes much better than the full gradient descent one [Keskar et al., 2016, Kleinberg et al., 2018]. However, thus far, most existing literature on theorizing why SGD works well for over-parameterized deep learning models also work for gradient descent on training loss [Allen-Zhu and Li, 2019, Allen-Zhu et al., 2018a,b, Arora et al., 2018]. In this work, we construct a convex learning problem where single pass SGD provably outperforms GD run on empirical loss, which converges to a solution that has large excess risk. We thus provide *a problem instance where SGD works but GD has strictly inferior sample complexity (with or without early stopping).*

**Multiple Epochs Help.** The final empirical observation we consider is the fact that multiple epochs of SGD tends to further continually decrease not only the training error but also the test error [Zhang et al., 2016, Ma et al., 2018, Bottou and Bousquet, 2011]. In this paper, we construct *an SCO instance for which multiple epochs of single sample mini-batched SGD significantly outperforms single pass SGD*, and for the same problem, *RERM fails to converge to a good solution*, and hence, so does GD when run to convergence.

## 1.1 Our Contributions

We now summarize our main contributions in the paper:

1. **SGD and RERM Separation.** In Section 3, we demonstrate a SCO problem where a single pass of SGD over $n$ data points obtains a $1/\sqrt{n}$ suboptimality rate. However, for any regularizer $R$, regularized ERM does not attain a diminishing suboptimality. We show that this is true even if the regularization parameter is chosen in a sample dependent fashion. Our result immediately rules out the explanation that SGD is successful in SCO due to some some implicit regularization.

2. **SGD and GD Separation.** In Section 4, we provide a separation between SGD and GD on training loss in terms of sample complexity for SCO. To the best of our knowledge, this is the first[1] such separation result between SGD and GD in terms of sample complexity. In this work, we show the existence of SCO problems where SGD with $n$ samples achieves a suboptimality of $1/\sqrt{n}$, however, irrespective of what step-size is used and how many iterations we run for, GD cannot obtain a suboptimality better than $1/(n^{5/12}\log^2(n))$.

3. **Single-pass vs Multi-pass SGD.** In Section 5, we provide an adaptive multi-epoch SGD algorithm that is provably at least as good as single pass SGD algorithm. On the other hand, we also show that this algorithm can far outperform single pass SGD on certain problems where SGD only attains a rate of $1/\sqrt{n}$. Also, on these problems RERM fails to learn, indicating that GD run to convergence fails as well.

4. **SGD and RERM Separation in the Distribution Free Agnostic PAC Setting.** The separation result between SGD and RERM introduced earlier was for SCO. However, it turns out that the problem is not agnostically learnable for all distributions but only for distributions that make $F$ convex. In Section 6, we provide a learning problem that is distribution-free learnable, and where SGD provably outperforms any RERM.

5. **Beyond Convexity (Deep Learning).** The convergence guarantee for SGD can be easily extended to stochastic optimization settings where the population loss $F(w)$ is *linearizable*, but may not be convex. We formalize this in Section 7, and show that for two layer diagonal neural networks with ReLU activations, there exists a distribution for which the population loss is *linearizable* and thus SGD works, but ERM algorithm fails to find a good solution. This hints at the possibility that the above listed separations between SGD and GD / RERM for the SCO setting, also extend to the deep learning setting.

## 1.2 Preliminaries

A standard assumption made by most gradient based algorithms for stochastic optimization problems is the following:

$$\textbf{Assumption II:} \begin{cases} F \text{ is } L\text{-Lipschitz (w.r.t. } \ell_2 \text{ norm)} \\ \exists w^\star \in \operatorname{argmin}_w F(w) \text{ s.t. } \|w_1 - w^\star\| \le B \ . \\ \sup_w \mathbb{E}_{z\sim D} \|\nabla f(w,z) - \nabla F(w)\|^2 \le \sigma^2 \end{cases} \tag{3}$$

In the above and throughout this work, the norm denotes the standard Euclidean norm and $w_1$ is some initial point known to the algorithm. Next, we describe below more formally what the regularized ERM, GD and SGD algorithms are.

**(Regularized) Empirical Risk Minimization.** Perhaps the simplest algorithm one could consider is the Empirical Risk Minimization (ERM) algorithm where one returns a minimizer of training loss (empirical risk)

$$F_S(w) := \frac{1}{n}\sum_{t=1}^{n} f(w; z_t) \ .$$

That is, $w_{\mathrm{ERM}} \in \operatorname{argmin}_{w\in\mathcal{W}} F_S(w)$. A more common variant of this method is one where we additionally penalize complex models using a regularizer function $R : \mathbb{R}^d \mapsto \mathbb{R}$. That is, a Regularized Empirical Risk Minimization (RERM) method consists of returning:

$$w_{\mathrm{RERM}} = \operatorname*{argmin}_{w\in\mathcal{W}} F_S(w) + R(w). \tag{4}$$

---

[1]Despite the way the result is phrased in Amir et al. [2021], their result does not imply separation between SGD and GD in terms of sample complexity but only number of iterations.

**Gradient Descent (GD) on Training Loss.** Gradient descent on training loss is the algorithm that performs the following update on every iteration:

$$w_{i+1}^{\mathrm{GD}} \leftarrow w_i^{\mathrm{GD}} - \eta \nabla F_S(w_i^{\mathrm{GD}}), \tag{5}$$

where $\eta$ denotes the step size. After $t$ rounds, we return the point $\widehat{w}_t^{\mathrm{GD}} := \frac{1}{t} \sum_{i=1}^t w_i^{\mathrm{GD}}$.

**Stochastic Gradient Descent (SGD).** Stochastic gradient descent (SGD) has been the method of choice for training large over-parameterized deep learning models, and other convex and non-convex learning problems. Single pass SGD algorithm runs for $n$ steps and for each step takes a single data point and performs gradient update with respect to that sample. That, is on round $t$,

$$w_{i+1}^{\mathrm{SGD}} \leftarrow w_i^{\mathrm{SGD}} - \eta \nabla f(w_i^{\mathrm{SGD}}; z_i) \tag{6}$$

Finally, we return $\widehat{w}_n^{\mathrm{SGD}} = \frac{1}{n} \sum_{i=1}^n w_i^{\mathrm{SGD}}$. Multi-epoch (also known as multi-pass) SGD algorithm simply cycles over the dataset multiple times continuing to perform the same update specified above. It is well know that single pass SGD algorithm enjoys the following convergence guarantee:

**Theorem 1** (Nemirovski and Yudin [1983]). *On any SCO problem satisfying Assumption I in (2) and Assumption II in (3), running SGD algorithm for $n$ steps with the step size of $\eta = 1/\sqrt{n}$ enjoys the guarantee*

$$\mathbb{E}_S[F(\widehat{w}_n^{\mathrm{SGD}})] - \inf_{w \in \mathbb{R}^d} F(w) \le O\Big(\frac{1}{\sqrt{n}}\Big),$$

*where the constant in the order notation only depends on constants $B, L$ and $\sigma$ in (3) and is independent of the dimension $d$.*

In fact, even weaker assumptions like one-point convexity or star convexity of $F$ w.r.t. an optimum on the path of the SGD suffices to obtain the above guarantee. We use this guarantee of SGD algorithm throughout this work. Up until Section 6 we only consider SCO problems for which SGD automatically works with the above guarantee.

## 2 Related Work

On the topic of RERM, implicit regularization and SGD, perhaps the work most relevant to this paper is that of Dauber et al. [2020]. Just like this work, they also consider the general setting of stochastic convex optimization (SCO) and show that for this setting, for no so-called *"admissible"* data independent regularizer, SGD can be seen as performing implicit regularized ERM. For the implicit regularization part of our work, while in spirit the work aims at accomplishing some of the similar goals, their work is in the setting where the instantaneous losses are also convex and Lipschitz. This means that, for their setting, while SGD and regularized ERM may not coincide, regularized ERM is indeed still an optimal algorithm as shown in Shalev-Shwartz et al. [2009]. In this work we show a strict separation between SGD and RERM with an example where SGD works but no RERM can possibly provide a non-trivial learning guarantee. Second, qualitatively, their separation result in Theorems 2 and 3 are somewhat unsatisfactory. This is because while they show that for every regularizer there is a distribution for which the SGD solution is larger in value than the regularized training loss of the regularized ERM w.r.t. that regularizer, the amount by which it is larger can depend on the regularizer and can be vanishingly small. Hence it might very well be that SGD is an approximate RERM. In fact, if one relaxes the requirement of their admissible relaxations then it is possible that their result doesn't hold. For instance, for square norm regularizer, the gap on regularized objective between RERM and SGD is only shown to be as large as the regularization parameter which is typically set to be a diminishing function of $n$.

On the topic of comparison of GD on training loss with single epoch SGD, one can hope for three kinds of separation. First, separation in terms of work (number of gradient computations), second, separation in terms of number of iterations, and finally separation in terms of sample complexity. A classic result from Nemirovski and Yudin [1983] tells us that for the SCO setting, the number of gradient computations required for SGD to obtain a suboptimality guarantee of $\varepsilon$ is equal to the optimal sample complexity (for $\varepsilon$) and hence is optimal (in the worst case). On the other hand, a single iteration of GD on training loss requires the same number of gradient computations and hence any more than a constant number of iterations of GD already gives a separation between GD and

SGD in terms of work. This result has been explored in Shalev-Shwartz et al. [2007] for instance. The separation of GD and SGD in terms of number of iterations has been considered in Amir et al. [2021]. Amir et al. [2021] demonstrate a concrete SCO problem on which GD on the training loss requires at least $\Omega(1/\varepsilon^4)$ steps to obtain an $\varepsilon$-suboptimal solution for the test loss. Whereas, SGD only requires $O(1/\varepsilon^2)$ iterations. However, their result does not provide any separation between GD and SGD in terms of sample complexity. Indeed, in their example, using the upper bound for GD using Bassily et al. [2020] one can see that if GD is run on $n$ samples for $T = n^2$ iterations with the appropriate step size, then it does achieve a $1/\sqrt{n}$ suboptimality. In comparison our work provides a much stronger separation between GD and SGD. We show a sample complexity separation, meaning that to obtain a specific suboptimality, GD requires more samples than SGD, irrespective of how many iterations we run it for. Our separation result also yields separation in terms of both number of iterations and number of gradient computations.

## 3 Regularized ERM, Implicit Regularization and SGD

In Shalev-Shwartz et al. [2009] (see also Feldman [2016]), SCO problems where not just the population loss $F$ but where also for each $z \in \mathcal{Z}$, the instantaneous loss $f(\cdot, z)$ is convex is considered. In this setting, the authors show that the appropriate $\ell_2$ norm square regularized RERM always obtains the optimal rate of $1/\sqrt{n}$. However, for SGD to obtain a $1/\sqrt{n}$ guarantee, one only needs convexity at the population level. In this section, based on construction of SCO problems that are convex only at population level and not at the empirical level, we show a strict separation between SGD and RERM. Specifically, while SGD is always successful for any SCO problem we consider here, we show that for any regularizer $R$, there is an instance of an SCO problem for which RERM w.r.t. this regularizer has suboptimality lower bounded by a constant.

In Shalev-Shwartz et al. [2009] (see also Feldman [2016]), SCO problems where not just the population loss $F$ but where also for each $z \in \mathcal{Z}$, the instantaneous loss $f(\cdot, z)$ is convex is considered. In this setting, the authors show that the appropriate $\ell_2$ norm square regularized RERM always obtains the optimal rate of $1/\sqrt{n}$. However, for SGD to obtain a $1/\sqrt{n}$ guarantee, one only needs convexity at the population level. In this section, based on construction of SCO problems that are convex only at population level and not at the empirical level, we show a strict separation between SGD and RERM. Specifically, while SGD is always successful for any SCO problem we consider here, we show that for any regularizer $R$, there is an instance of an SCO problem for which RERM with respect to this regularizer has suboptimality lower bounded by a constant.

**Theorem 2.** *For any regularizer $R$, there exists an instance of a SCO problem that satisfies both Assumptions I and II given in Equations (2) and (3), for which*

$$\mathbb{E}_S[F(w_{\mathrm{RERM}})] - \inf_{w \in \mathbb{R}^d} F(w) \geq \Omega(1),$$

*where $w_{\mathrm{RERM}}$ is the solution to (4) with respect to the prescribed regularizer.*

The regularizer $R$ we consider in the above result is sample independent, that is, it has to be chosen before receiving any samples. In general, if one is allowed an arbitrary sample dependent regularizer, one can encode any learning algorithm as an RERM. This is because, for any algorithm, one can simply contrive the regularizer $R$ to have its minimum at the output model of the algorithm on the given sample, and a very high penalty on other models. Hence, to have a meaningful comparison between SGD (or for that matter any algorithm) and RERM, one can either consider sample independent regularizers or at the very least, consider only some specific restricted family of sample dependent regularizers. One natural variant of considering mildly sample dependent regularizers is to first, in a sample independent way pick some regularization function $R$ and then allow arbitrary sample dependent regularization parameters that multiply the regularizer $R$. The following corollary shows that even with such mildly data dependent regularizers, one can still find SCO instances for which the RERM solution has no non-trivial convergence guarantees.

**Corollary 1.** *For any regularizer $R$, there exists an instance of a SCO problem that satisfies both Assumptions I and II, for which the point $w_{\mathrm{RERM}} = \mathrm{argmin}_{w \in \mathcal{W}} F_S(w) + \lambda R(w)$, where $\lambda$ is any arbitrary sample dependent regularization parameter, has the lower bound*

$$\mathbb{E}_S[F(w_{\mathrm{RERM}})] - \inf_{w \in \mathbb{R}^d} F(w) \geq \Omega(1).$$

The SCO problem in Theorem 2 and Corollary 1 is based on the function $f_{(A)}$ given by:

$$f_{(A)}(w; z) = y\|(w - \alpha) \odot x\|, \tag{A}$$

where each instance $z \in \mathcal{Z}$ can be written as a triplet $z = (x, y, \alpha)$ and the notation $\odot$ denotes Hadamard product (entry wise product) of the two vectors. We set $x \in \{0, 1\}^d$, $y \in \{\pm 1\}$ and $\alpha \in \{0, e_1, \ldots, e_d\}$ where $e_1$ to $e_d$ denote the standard basis in $d$ dimensions. We also set $d > 2^n$.

In the following, we provide a sketch for why $\ell_2$-norm square regularization fails and show no regularization parameter works. In the detailed proof provided in the Appendix, we deal with arbitrary regularizers. The basic idea behind the proof is simple. Consider the distribution: $x \sim \mathrm{Unif}(\{0, 1\})^d$, and $y$ is set to be $+1$ with probability $0.6$ and $-1$ with probability $0.4$, and set $\alpha = e_1$ deterministically. In this case, note that the population function is $0.2 \, \mathbb{E}_{x \sim \mathrm{Unif}\{0,1\}^d} \|x \odot (w - e_1)\|_2$ which is indeed convex, Lipchitz and sandwiched between $0.1\|w - e_1\|$ and $0.2\|w - e_1\|$. Hence, any $\varepsilon$ sub-optimal solution $\widehat{w}$ must satisfy $\|\widehat{w} - e_1\| \leq 10\varepsilon$.

Since $d > 2^n$, with constant probability there is at least one coordinate, say $\widehat{j} \in [d]$, such that for any data sample $(x_t, y_t)$ for $t \in [n]$, we have $x_t[\widehat{j}] = 0$ whenever $y_t = +1$ and $x_t[\widehat{j}] = 1$ whenever $y_t = -1$. Hence, ERM would simply put large weight on the $\widehat{j}$ coordinate and attain a large negative value for training loss (as an example, $w = e_{\widehat{j}}$ has a training loss of roughly $-0.4$). However, for test loss to be small, we need the algorithm to put little weight on coordinate $\widehat{j}$. Now for any square norm regularizer $R(w) = \lambda\|w\|_2^2$, to prevent RERM from making this $\widehat{j}$ coordinate large, $\lambda$ has to be at least as large as a constant.

However, we already argued that any $\varepsilon$ sub-optimal solution $\widehat{w}$ should be such that $\|\widehat{w} - e_1\| \leq 10\varepsilon$. When $\lambda$ is chosen to be as large as a constant, the regularizer will bias the solution towards $0$ and thus the returned solution will never satisfy the inequality $\|\widehat{w} - e_1\| \leq 10\varepsilon$. This leads to a contradiction: To find an $\varepsilon$ sub-optimal solution, we need a regularization that would avoid picking large value on the spurious coordinate $\widehat{j}$ but on the other hand, any such strong regularization, will not allow RERM to pick a solution that is close enough to $e_1$. Hence, any such regularized ERM cannot find an $\varepsilon$ sub-optimal solution. This shows that no regularization parameter works for norm square regularization. In the Appendix we expand this idea for arbitrary regularizers $R$.

The construction we use in this section has dimensionality that is exponential in number of samples $n$. However similarly modifying the construction in Feldman [2016], using the $y = \pm 1$ variable multiplying the construction there, one can extend this result to a case where dimensionality is $\Theta(n)$. Alternatively, noting that for the maximum deviation from mean over $d$ coordinates is of order $\sqrt{\log d/n}$, one can use a simple modification of the exact construction here, and instead of getting the strong separation like the one above where RERM does not learn but SGD does, one can instead have $d = n^p$ for $p \in (0, 1]$ and obtain a separation where SGD obtains the $1/\sqrt{n}$ rate but no RERM can beat a rate of order $\sqrt{\log n/n}$.

**Implicit Regularization.** As mentioned earlier, a proposed theory for why SGD algorithms are so successful is that they are finding some implicitly regularized empirical risk minimizers, and that this implicit bias helps them learn effectively with low generalization error. However, at least for SCO problems, the above strict separation result tells us that SGD cannot be seen as performing implicitly regularized ERM, neither exactly nor approximately.

## 4  Gradient Descent (Large-batch) vs SGD (Small-batch)

In the previous section, we provided an instance of a SCO problem on which no regularized ERM works as well as SGD. When gradient descent algorithm (specified in (5)) is used for training, one would expect that GD, and in general large batch SGD, will also eventually converge to an ERM and so after enough iterations would also fail to find a good solution. In this section, we formalize this intuition to provide lower bounds on the performance of GD.

**Theorem 3.** *There exists an instance of a SCO problem such that for any choice of step size $\eta$ and number of iterations $T$, the following lower bound on performance of GD holds:*

$$\mathbb{E}_S[F(\widehat{w}_T^{GD})] - \inf_{w \in \mathbb{R}^d} F(w) \geq \Omega\Big(\frac{1}{n^{5/12}}\Big).$$

Theorem 3 suggests that there is an instance of a SCO problem for which the performance of GD is lower bound by $1/n^{0.42}$. On the other hand, SGD with the step size of $\eta = 1/n^{0.5}$ learns at a rate of $1/n^{0.5}$ for this problem. This suggests that GD (large batch size) is a worse learning algorithm than SGD. We defer the proof to Appendix C and give a brief sketch below.

Our lower bound proof builds on the recent works of Amir et al. [2021], which gives an instance of a SCO problem for which the performance guarantee of GD algorithm is $\Omega(\eta\sqrt{T} + 1/\eta T)$. We provide an instance of a SCO problem for which GD has a lower bound of $\Omega(\eta T/n)$. Adding the two instances together gives us an SCO problem for which GD has lower bound of

$$\Omega\Big(\eta\sqrt{T} + \frac{1}{\eta T} + \frac{\eta T}{n}\Big). \tag{7}$$

This lower bound is by itself not sufficient. In particular, for $\eta = 1/n^{3/2}$ and $T = n^2$, the right hand side of (7) evaluates to $O(1/\sqrt{n})$, which matches the performance guarantee of SGD for this SCO problem. Hence, this leaves open the possibility that GD may work as well as SGD. However, note that in order to match the performance guarantee of SGD, GD needs to be run for quadratically more number of steps and with a smaller step size. In fact, we can show that in order to attain a $O(1/\sqrt{n})$ rate in (7), we must set $\eta = O(1/n^{3/2})$ and $T = \omega(n^2)$.

The lower bound in Theorem 3 follows by adding to the SCO instance in (7) another objective that rules out small step-sizes. This additional objective is added by increasing the dimensionality of the problem by one and on this extra coordinate adding a stochastic function that is convex in expectation. The expected loss on this coordinate is a piecewise linear convex function. However, the stochastic component on this coordinate has random kinks at intervals of width $1/n^{5/4}$ that vanish in expectation, but can make the empirical loss point in the opposite direction with probability $1/2$. Since the problem is still an SCO problem, SGD works as earlier. On the other hand, when one considers training loss, there are up to $n$ of these kinks and roughly half of them make the training loss flat. Thus, if GD is used on training loss with step size smaller than $1/n^{5/4}$ then it is very likely that GD hits at least one such kink and will get stuck there. This function, thus, rules out GD with step size $\eta$ smaller than $1/n^{5/4}$. Restricting the step size $\eta = \Omega(1/n^{5/4})$, the lower bound in (7) evaluates to the $\Omega(1/n^{5/12})$ giving us the result of Theorem 3.

Our lower bound in (7) matches the recently shown performance guarantee for GD algorithm by Bassily et al. [2020]; albeit under slightly different assumptions on the loss function $f(\cdot; z)$. Their work assumes that the loss functions $f(w; z)$ is convex in $w$ for every $z \in \mathcal{Z}$. On the other hand, we do not require convexity of $f$ but only that $F(w)$ is convex in $w$ (see our Assumptions I and II).

## 5 Single-pass vs Multi-pass SGD

State of the art neural networks are trained by taking multiple passes of SGD over the dataset. However, it is not well understood when and why multiple passes help. In this section, we provide theoretical insights into the benefits of taking multiple passes over the dataset. The multi-pass SGD algorithm that we consider is:

1. Split the dataset $S$ into two equal sized datasets $S_1$ and $S_2$.
2. Run $k$ passes of SGD algorithm using a fixed ordering of the samples in $S_1$ where,
   - The step size for the $j$th pass is set as $\eta_j = 1/\sqrt{nj}$.
   - At the end of $j$th pass, compute $\widehat{w}_j = \frac{2}{nj}\sum_{t=1}^{nj/2} w_j^{\text{SGD}}$ as the average of all the iterates generated so far.
3. Output the point $\widehat{w}^{\text{MP}} := \operatorname{argmin}_{w \in \widehat{\mathcal{W}}} F_{S_2}(w)$ where $\widehat{\mathcal{W}} := \{\widehat{w}_1, \ldots, \widehat{w}_k\}$.

The complete pseudocode is given in the Appendix. In the following, we show that the above multi-pass SGD algorithm performs at least as well as taking a single pass of SGD.

**Proposition 1.** *The output $\widehat{w}^{\text{MP}}$ of multipass-SGD algorithm satisfies*

$$\mathbb{E}_S[F(\widehat{w}^{\text{MP}})] \leq \mathbb{E}_S[F(\widehat{w}_{n/2}^{\text{SGD}})] + \widetilde{O}\Big(\frac{1}{\sqrt{n}}\Big),$$

*where $\widehat{w}_{n/2}^{\text{SGD}}$ denotes the output of running (one pass) SGD algorithm for $n/2$ steps with step size $1/\sqrt{n}$.*

This suggests that the output point of the above multi-pass SGD algorithm is not too much worse than that of SGD (single pass). For problems in SCO, SGD has a rate of $O(1/\sqrt{n})$ (see Theorem 1), and in for these problems, the above bound implies that multi-pass SGD also enjoys the rate of $\widetilde{O}(1/\sqrt{n})$.

## 5.1 Multiple Passes Can Help!

In certain favorable situations the output of multi-pass SGD can be much better:

**Theorem 4.** *Let $k$ be a positive integer and let $R(\cdot)$ be a regularization function. There exists an instance of a SCO problem such that:*

*(a) For any step size $\eta$, the output of the SGD algorithm has the lower bound*

$$\mathbb{E}_S[F(\widehat{w}_n^{SGD})] - \inf_{w \in \mathbb{R}^d} F(w) \geq \Omega\Big(\frac{1}{\sqrt{n}}\Big).$$

*Furthermore, running SGD with $\eta = 1/\sqrt{n}$ achieves the above $1/\sqrt{n}$ rate.*

*(b) On the other hand, multi-pass SGD algorithm with $k$ passes has the following guarantee:*

$$\mathbb{E}_S[F(\widehat{w}^{\mathrm{MP}})] - \inf_{w \in \mathbb{R}^d} F(w) \leq O\Big(\frac{1}{\sqrt{nk}}\Big).$$

*(c) RERM algorithm has the lower bound: $\mathbb{E}_S[F(w_{\mathrm{RERM}})] - \inf_{w \in \mathbb{R}^d} F(w) \geq \Omega(1)$.*

We defer the proof details to Appendix D and provide the intuition below. Parts (a) and (b) follow easily by taking a standard construction in [Nesterov, 2014, Section 3.2.1]. Here, a convex deterministic function $F_N$ over an optimization variable $v$ is provided such that Assumption II is satisfied with $L, B = O(1)$, and any gradient based scheme (such as SGD, which is equivalent to GD on $F_N$) has a suboptimality of $\Omega(1/\sqrt{n})$ after $n$ iterations (which corresponds to single pass SGD). On the other hand, because $F_N$ satisfies Assumption II, $k$ passes of SGD, which correspond to $nk$ iterations of GD, will result in suboptimality of $O(1/\sqrt{nk})$ for the given step sizes.

The more challenging part is to prove part (c). It is tempting to simply add the SCO instance from Theorem 2, but that may break the required upper bound of $O(1/\sqrt{nk})$ for multipass SGD. To get around this issue, we construct our SCO instance by making $z$ consist of $k$ components $\{\xi_1, \ldots, \xi_k\}$ drawn independently from the distribution considered in Theorem 2. Furthermore, the loss function considered in Theorem 2 also defines the loss corresponding to each $\xi_i$ component for an optimization variable $w$ that is different from the optimization variable $v$.

The key idea in the construction is that, while each data sample consists of $k$ independently sampled components, at any time step, SGD gets to see only one of these components. Specifically, in the first pass over the dataset, we only observe the $\xi_1$ component of every sample, and in the second pass we only observe the $\xi_2$ component for every sample and so on for further passes. This behavior for SGD is induced by using an independent control variable $u$. The optimization objective for $u$ is such that SGD increases $u$ monotonically with every iteration. The value of $u$ controls which of the $\xi_i$ components is observed during that time step. In particular, when we run our multipass SGD algorithm, during the first pass, the value of $u$ is such that we get to see $\xi_1$ only. However, on the second pass, $u$ has become large enough to reveal $\xi_s 2$, and so on for further passes. Thus, in $k$ passes, multipass SGD gets to see $nk$ "fresh" samples and hence achieves an suboptimality of $O(1/\sqrt{nk})$, as required in part (b). Finally, part (c) follows from the same reasoning as in Theorem 2 since the same SCO instance is used.

## 6 Distribution Free Learning

In this section, we consider the general learning setting [Vapnik, 2013] and aim for a distribution free learnability result. That is, we would like to provide problem settings where one has suboptimality that diminishes with $n$ for any distribution over the instance space $\mathcal{Z}$. For the SCO setting we considered earlier, the assumptions in (2) and the assumption in (3) that the population loss is convex, imposes restrictions on the distributions allowed to be considered. Specifically, since $f(\cdot\,;z)$ need not be convex for every $z$, one can easily construct distributions for which the problem is not SCO and in

fact, may not even be learnable. E.g., consider the distribution that puts all its mass on $z$ for which $f(\cdot\,;z)$ is non-convex. In this section we will consider problems that are so called learnable, meaning that there is a learning algorithm with diminishing in $n$ suboptimality for any distribution $\mathcal{D}$ on the instance space $\mathcal{Z}$. Under this setting, we show a separation between SGD and RERM. Specifically, we provide a problem instance that is learnable at a rate of $c_n > \Omega(1/n^{1/4})$ for any distribution over the instance space $\mathcal{Z}$. In particular, we show that the worst case rate of SGD is $c_n$ for this problem. However, on the subset of distributions for which the problem is SCO, SGD as expected obtains a rate of $1/\sqrt{n}$. On the other hand, for the same problem we show that RERM while having worst case rate no better than $c_n$, has a lower bound of $\Omega(c_n^2)$ on SCO instances.

Our lower bound in this section is based on the following learning setting: For $d > 2^n$, let $\mathcal{Z} = \{0,1\}^d \times \{0, e_1, \ldots, e_d\}$ be the instance space, let the instantaneous loss function be given by:

$$f_{(B)}(w;z) = \frac{1}{2}\|(w-\alpha)\odot x\|^2 - \frac{c_n}{2}\|w-\alpha\|^2 + \max\{1, \|w\|^4\}, \tag{B}$$

where $z = (x, \alpha)$ and $c_n := n^{-(\frac{1}{4}-\gamma)}$ for some $\gamma > 0$.

Additionally, let $\mathscr{D}_c$ denote the set of distributions over instances space $\mathcal{Z}$ for which $F$ is convex, i.e.

$$\mathscr{D}_c := \big\{\mathcal{D} \mid F(w) = \mathbb{E}_{z\sim\mathcal{D}}\big[f_{(B)}(w,z)\big] \text{ is convex in } w\big\}. \tag{8}$$

Note that under the distributions from $\mathscr{D}_c$, the problem is an instance of an SCO problem. We show a separation between SGD and RERM over the class $\mathscr{D}_c$. In the theorem below we claim that the above learning problem is learnable with a worst case rate of order $c_n$ (by SGD), and whenever $\mathcal{D} \in \mathscr{D}_c$, SGD learns at a faster rate of $1/\sqrt{n}$. To begin with we first provide the following proposition for the problem described by the function $f_{(B)}$ that shows that no algorithm can have a worst case rate better than $c_n$ for this problem.

**Proposition 2.** *For any algorithm* ALG *that outputs* $\widehat{w}^{\mathrm{ALG}}$, *there is a distribution* $\mathcal{D}$ *on the instance space, such that for the learning problem specified by function* $f_{(B)}$:

$$\mathbb{E}_S[F(\widehat{w}^{\mathrm{ALG}})] - \inf_{w\in\mathcal{W}} F(w) \geq \frac{c_n}{4}$$

Next, we show that SGD obtains this worst case rate, and a much better rate of $1/\sqrt{n}$ for any distribution in $\mathscr{D}_c$. However, we also show that for this problem, no RERM can obtain a rate better than $c_n^2$ for every distribution in $\mathscr{D}_c$. This shows that while SGD and RERM have the same worst case rate, SGD outperforms any RERM whenever the problem turns out to be convex.

**Theorem 5.** *For the learning problem specified by function* $f_{(B)}$:

  *(a) For every regularizer $R$, there exists a distribution $\mathcal{D} \in \mathscr{D}_c$ such that,*
  $$\mathbb{E}_S[F(w_{\mathrm{RERM}})] - \inf_{w\in\mathbb{R}^d} F(w) \geq \Omega(c_n^2).$$

  *(b) For the SGD algorithm we have the following upper bounds:*
  $$\forall \mathcal{D} \in \mathscr{D}_c, \ \mathbb{E}_S[F(\widehat{w}_n^{\mathrm{SGD}})] - \inf_{w\in\mathbb{R}^d} F(w) \leq O\left(\frac{1}{\sqrt{n}}\right),$$
  $$\forall \mathcal{D} \notin \mathscr{D}_c, \ \mathbb{E}_S[F(\widehat{w}_n^{\mathrm{SGD}})] - \inf_{w\in\mathbb{R}^d} F(w) \leq O\left(c_n\right).$$

As an example, plugging in $c_n = n^{-\frac{1}{8}}$ implies that when $\mathcal{D} \notin \mathscr{D}_c$, the suboptimality of SGD is bounded by $O(1/n^{1/8})$, and when $\mathcal{D} \in \mathscr{D}_c$, the suboptimality of SGD is bounded by $O(1/\sqrt{n})$. However, for any RERM, there exists a distribution $D \in \mathscr{D}_c$, on which the RERM has a suboptimality of $\Omega(1/n^{1/4})$ and the worst case rate of any RERM is also $n^{-1/8}$. This suggests that SGD is a superior algorithm to RERM for any regularizer $R$, even in the distribution free learning setting.

## 7  $\alpha$-Linearizable Functions and Deep Learning

While the classic convergence proof for SGD is shown for SCO setting, a reader familiar with the proof technique will recognize that the same result also holds when we only assume that the population loss $F$ is star-convex or one-point-convex with respect to any optimum $w^*$, or in fact even if it is star-convex only on the path of SGD. The following definition of Linearizable population loss generalizes star-convexity and one-point-convexity.

**Definition 1** ($\alpha$-Linearizable). *A stochastic optimization problem with population loss $F(w)$ is $\alpha$-Linearizable if there exists a $w^* \in \arg\min F(w)$ such that for every point $w \in \mathbb{R}^d$,*

$$F(w) - F(w^*) \leq \alpha \langle \nabla F(w), w - w^* \rangle.$$

For linearizable function, one can upper bound the suboptimality at any point $w$ by a linear function given by $\nabla F(w)$. The convergence guarantee for SGD now follows by bounding the cumulative sum of this linear function using standard arguments, giving us the following performance guarantee.

**Theorem 6.** *On any $\alpha$-Lineariazable stochastic optimization problem satisfying Assumption II in (3), running SGD algorithm for $n$ steps with the step size of $\eta = 1/\sqrt{n}$ enjoys the guarantee:*

$$\mathbb{E}_S[F(\widehat{w}_n^{\mathrm{SGD}})] - \inf_{w \in \mathbb{R}^d} F(w) \leq O\Big(\frac{\alpha}{\sqrt{n}}\Big),$$

*where the constant in the order notation only depends on constants $B, L$ and $\sigma$ in (3) and is independent of the dimension $d$.*

The hypothesis that this phenomenon is what makes SGD successful for deep learning has been proposed and explored in various forms [Zhou et al., 2019, Kleinberg et al., 2018]. On the other hand our lower bound results hold even for the simpler SCO setting. Of course, to claim such a separation of SGD with respect to GD or RERM in the deep learning setting, one would need to show that our lower bound constructions can be represented as deep neural networks with roughly the same dimensionality as the original problem. In fact, all the functions that we considered so far can be easily expressed by restricted deep neural networks (where some weights are fixed) with square activation functions as we show in Appendix F.3. Although, it would be a stretch to claim that practical neural networks would look anything like our restricted neural network constructions; it still opens the possibility that the underlying phenomena we exploit to show these separations hold in practical deep learning setting. In the following, we give an example of a simple two layer neural network with ReLU activation function where SGD enjoys a rate of $1/\sqrt{n}$, but any ERM algorithm fails to find an $O(1)$-suboptimal solution.

Let the input sample $(x, y)$ be such that $x \in \{0, 1\}^d$ and $y \in \{-1, 1\}$. Given the weights $w = (w_1, w_2)$, where $w_1 \in \mathbb{R}^d$ and $w_2 \in \mathbb{R}^d$, we define a two layer ReLU neural network that on the input $x$ outputs

$$h(w; x) = \mathrm{ReLU}(w_2^\top \mathrm{ReLU}(w_1 \odot x)).$$

This is a two layer neural network with the input layer having a diagonal structure and output layer being fully connected (hence the name diagonal neural network). Suppose the network is trained using the absolute loss, i.e. on data sample $z = (x, y)$ and for weights $w = (w_1, w_2)$, we use the loss

$$f(w; z) = |y - h(w; z)| = |y - \mathrm{ReLU}(w_2^\top \mathrm{ReLU}(w_1 \odot x))|. \tag{9}$$

**Theorem 7** (Two layer diagonal network). *For the loss function given in (9) using a two layer diagonal neural network, there exists a distribution $\mathcal{D}$ over the instance space $\mathcal{Z}$ such that:*

(a) *$F(w)$ is $1/2$-Linearizable, and thus SGD run with step-size $1/\sqrt{n}$ has excess risk $O(1/\sqrt{n})$.*

(b) *For $d \geq 2^n$, with probability $0.9$, ERM algorithm fails to find an $O(1)$-suboptimal point.*

**Remark 1.** *The result of Theorem 7 can be extended to diagonal two layer neural networks trained with linear loss $f(w; z) = yh(w; x)$, or with hinge loss $f(w; x) = \max\{0, 1 - yh(w; z)\}$.*

While the above result shows that for a simple two layer neural network, SGD performs better than ERM, our construction requires the first layer to be diagonal. It is an interesting future research direction to explore whether a similar phenomena can be demonstrated in more practical network architectures, for eg. fully connected networks, convolutional neural networks (CNN), recurrent neural networks (RNN), etc. It would also be interesting to extend our lower bounds for GD algorithm from Section 4 to these network architectures. The key idea is that SGD only requires certain nice properties (eg. convex, Linearizable, etc) at the population level, which might fail to hold at the empirical level in large dimensional models; hence, batch algorithms like GD and RERM might fail.

**Acknowledgements**

We thank Roi Livni, Robert Kleinberg and Mehryar Mohri for helpful discussions. AS was an intern at Google Research, NY when a part of the work was performed. KS acknowledges support from NSF CAREER Award 1750575.

**Funding Transparency Statement**

Funding in direct support of this work: NSF CAREER Award 1750575.

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
