# Contents of Appendix

# A Preliminaries

## A.1 Additional notation

For a vector $w \in \mathbb{R}^d$, fo any $j \in [d]$, $w[j]$ denotes the $j$-th coordinate of $w$, $\|w\|$ denotes the Euclidean norm and $\|w\|_\infty$ denotes the $\ell_\infty$ norm. For any two vectors $w_1$ and $w_2$, $\langle w_1, w_2 \rangle$ denotes their inner product, and $w_1 \odot w_2$ denotes the vector generated by taking the Hadamard product of $w_1$ and $w_2$, i.e. $(w_1 \odot w_2)[j] = w_1[j]w_2[j]$ for $j \in [d]$. We denote by $\mathbf{1}_d$ a $d$-dimensional vector of all 1s, and the notation $\mathbb{I}_d$ denotes the identity matrix in $d$-dimensions. The notation $\mathcal{N}(0, \sigma^2)$ denotes Gaussian distribution with variance $\sigma^2$, and $\mathcal{B}(p)$ denotes the Bernoulli distribution with mean $p$.

For a function $f : \mathbb{R}^d \times \mathbb{R}$, we denote the gradient of $f$ at the point $w \in \mathbb{R}^d$ by $\nabla f(w) \in \mathbb{R}^d$. The function $f$ is said to be $L$-Lipschitz if $f(w_1) - f(w_2) \leq L\|w_1 - w_2\|$ for all $w_1, w_2$.

## A.2 Basic algorithmic results

The following convergence guarantee for SGD algorithm is well known in stochastic convex optimization literature, and is included here for the sake of completeness.

**Theorem 8** (Nemirovski and Yudin [1983]). *Let $w \in \mathbb{R}^d$ and $z \in \mathcal{Z}$. Given an initial point $w_1 \in \mathbb{R}^d$, loss function $f(w; z)$ and a distribution $\mathcal{D}$ such that:*

*(a) The population loss $F(w) = \mathbb{E}_{z \sim \mathcal{D}}[f(w; z)]$ is L-Lipschitz and convex in $w$.*

*(b) For any $w$, $\mathbb{E}_{z \sim \mathcal{D}}[\|\nabla f(w; z) - \nabla F(w)\|] \leq \sigma^2$.*

*(c) The initial point $w_1$ satisfies $\|w_1 - w^*\| \leq B$ where $w^* \in \arg\min F(w)$.*

*Further, let $S \sim \mathcal{D}^n$. Then, the point $\widehat{w}_n^{\mathrm{SGD}}$ obtained by running SGD algorithm given in (6), with step size $\eta = 1/\sqrt{n}$ for $n$ steps using the dataset $S$ satisfies*

$$\mathbb{E}\big[F(\widehat{w}_n^{\mathrm{SGD}}) - F^*\big] \leq \frac{1}{\sqrt{n}}\big(\sigma^2 + L^2 + B^2\big),$$

*where $F^* := \min_w F(w)$.*

**Proof.** Let $\{w_t\}_{t \geq 1}$ denote the sequence of iterates generated by the SGD algorithm. We note that for any time $t \geq 1$,

$$\begin{aligned}
\|w_{t+1} - w^*\|_2^2 &= \|w_{t+1} - w_t + w_t - w^*\|_2^2 \\
&= \|w_{t+1} - w_t\|_2^2 + \|w_t - w^*\|_2^2 + 2\langle w_{t+1} - w_t, w_t - w^* \rangle \\
&= \|-\eta\nabla f(w_t; z_t)\|_2^2 + \|w_t - w^*\|_2^2 + 2\langle -\eta\nabla f(w_t; z_t), w_t - w^* \rangle,
\end{aligned}$$

where the last line follows from plugging in the SGD update rule that $w_{t+1} = w_t - \eta\nabla f(w_t; z_t)$.

Rearranging the terms in the above, we get that

$$\langle \nabla f(w_t; z_t), w_t - w^* \rangle \leq \frac{\eta}{2}\|\nabla f(w_t; z_t)\|_2^2 + \frac{1}{2\eta}\big(\|w_t - w^*\|_2^2 - \|w_{t+1} - w^*\|_2^2\big).$$

Taking expectation on both the sides, while conditioning on the point $w_t$, implies that

$$\begin{aligned}
\langle \nabla F(w_t), w_t - w^* \rangle &\leq \frac{\eta}{2}\,\mathbb{E}\Big[\|\nabla f(w_t; z_t)\|_2^2\Big] + \frac{1}{2\eta}\big(\|w_t - w^*\|_2^2 - \|w_{t+1} - w^*\|_2^2\big) \\
&\leq \eta\,\mathbb{E}\Big[\|\nabla f(w_t; z_t) - \nabla F(w_t)\|_2^2\Big] + \eta\|\nabla F(w_t)\|_2^2 \\
&\qquad + \frac{1}{2\eta}\big(\|w_t - w^*\|_2^2 - \|w_{t+1} - w^*\|_2^2\big) \\
&\leq \eta(\sigma^2 + L^2) + \frac{1}{2\eta}\big(\|w_t - w^*\|_2^2 - \|w_{t+1} - w^*\|_2^2\big),
\end{aligned}$$

where the inequality in the second line is given by the fact that $(a - b)^2 \leq 2a^2 + 2b^2$ and the last line follows from using Assumption II (see (3)) which implies that $F(w)$ is $L$-Lipschitz in $w$ and that

$\mathbb{E}[\|\nabla f(w; z_t) - \nabla F(w)\|_2^2] \leq \sigma^2$ for any $w$. Next, using convexity of the function $F$, we have that $F(w^*) \geq F(w_t) - \langle \nabla F(w_t), w_t - w^* \rangle$. Thus

$$F(w_t) - F^* \leq \eta(\sigma^2 + L^2) + \frac{1}{2\eta}\left(\|w_t - w^*\|_2^2 - \|w_{t+1} - w^*\|_2^2\right).$$

Telescoping the above for $t$ from $1$ to $n$, we get that

$$\sum_{t=1}^{n}(F(w_t) - F^*) \leq \eta n(\sigma^2 + L^2) + \frac{1}{2\eta}\left(\|w_1 - w^*\|_2^2 - \|w_{n+1} - w^*\|_2^2\right)$$

$$\leq \eta n(\sigma^2 + L^2) + \frac{1}{2\eta}\|w_1 - w^*\|_2^2.$$

Dividing both the sides by $n$, we get that

$$\frac{1}{n}\sum_{t=1}^{n}(F(w_t) - F^*) \leq \eta(\sigma^2 + L^2) + \frac{1}{2\eta n}\|w_1 - w^*\|_2^2.$$

An application of Jensen's inequality on the left hand side, implies that the point $\widehat{w}_n := \frac{1}{n}\sum_{t=1}^{n} w_t$ satisfies

$$\mathbb{E}[F(\widehat{w}_n) - F^*] \leq \eta(\sigma^2 + L^2) + \frac{1}{2\eta n}\|w_1 - w^*\|_2^2.$$

Setting $\eta = \frac{1}{\sqrt{n}}$ and using the fact that $\|w_1 - w^*\|_2 \leq B$ in the above, we get that

$$\mathbb{E}[F(\widehat{w}_n) - F^*] \leq \frac{1}{\sqrt{n}}(\sigma^2 + L^2 + B^2),$$

which is the desired claim. Dependence on the problem specific constants ($\sigma, L$ and $B$) in the above bound can be improved further with a different choice of the step size $\eta$; getting the optimal dependence on these constants, however, is not the focus of this work . $\qquad\square$

### A.3 Basic probability results

**Lemma 1** (Hoeffding's inequality). *Let $X_1, \ldots, X_n$ be independent random variables with values in the interval $[a, b]$, and the expected value $\mathbb{E}[X] = \mu$. Then, for every $t \geq 0$,*

$$\Pr\left(\left|\frac{1}{n}\sum_{j=1}^{n} X_j - \mu\right| \geq t\right) \leq 2\exp\left(-\frac{2t^2 n}{(b-a)^2}\right).$$

**Lemma 2.** *Let $j^* \in [d]$. Let $X$ be a $\{0, 1\}^d$ valued random variable such that $X[j]$ is sampled independently from $\mathcal{B}(p)$ for every $j \in [d]$. Let the $\{X_1, \ldots, X_n\}$ denote $n$ i.i.d. samples of the random variable $X$. If*

$$n \leq \log_{\frac{1}{1-p}}\left(\frac{d-1}{\ln(10)}\right),$$

*then, with probability at least $0.9$, there exists a coordinate $\widehat{j} \in [d]$ such that $\widehat{j} \neq j^*$, and $X_i[\widehat{j}] = 0$ for all $i \in [n]$.*

**Proof.** Let $E_j$ denote the event the coordinate $X_i[j] = 0$ for all $i \in [n]$. We note that for any $j \in [d]$,

$$\Pr(E_j) = (1-p)^n,$$

Further, let $E$ denote the event that there exists a coordinate $\widehat{j} \in [d] \setminus \{j^*\}$ such that $X_i[\widehat{j}] = 0$ for all $i \in [n]$. Thus,

$$\Pr(E^c) = \Pr\left(\bigcap_{j \in [d]\setminus\{j^*\}} E_j^c\right) \overset{(i)}{=} \prod_{j \in [d]\setminus\{j^*\}} \Pr(E_j^c)$$

where the equality in $(i)$ follows from the fact that the events $\{E_j\}$ are mutually independent to each other. Plugging in the bound for $\Pr(E_j^c)$, we get

$$\Pr(E^c) = (1 - (1-p)^n)^{d-1}.$$

Using the fact that $1 - a \leq e^{-a}$ for $a \geq 0$, we get

$$\Pr(E^c) \leq e^{-(1-p)^n(d-1)} \leq 0.1,$$

where the second inequality above holds for $n \leq \log_{\frac{1}{1-p}}\left(\frac{d-1}{\ln(10)}\right)$. $\qquad\square$

**Lemma 3** (Proposition 7.3.2, Matoušek and Vondrák [2001])**.** *For $n$ even, let $X_1, \ldots, X_n$ be independent samples from $\mathcal{B}(1/2)$. Then, for any $t \in [0, n/8]$,*

$$\Pr\Big(\sum_{i=1}^{n} X_1 \geq \frac{n}{2} + t\Big) \geq \frac{1}{15}e^{-16t^2/n}.$$

**Lemma 4** (Panchenko [2002])**.** *Let $\mathcal{W}$ denote a finite class of $k$ points $\{w_1, \ldots, w_K\}$, and let $f(w; z)$ denote a loss function that is $L$-Lipschitz in the variable $w$ for all $z \in \mathcal{Z}$. Further, let $S = \{z_i\}_{i=1}^{n}$ denote a dataset of $n$ samples, each drawn independently from some distribution $\mathcal{D}$. Define the point $\widehat{w}_S = \operatorname{argmin}_{w \in \mathcal{W}} \sum_{i=1}^{n} \frac{1}{n} f(w, z_i)$. Then, with probability at least $1 - \delta$ over the sampling of the set $S$,*

$$F(\widehat{w}_S) \leq F^* + O\Big(\frac{L\log(K/\delta)}{n} + \sqrt{\frac{F^* L\log(K/\delta)}{n}}\Big),$$

*where $F(w) := \mathbb{E}_{z \sim \mathcal{D}}[f(w; z)]$ and $F^* := \min_{w \in \mathcal{W}} F(w)$.*

# B    Missing proofs from Section 3

Throughout this section, we assume that a data sample $z$ consists of $(x, y, \alpha)$, where $x \in \{0, 1\}^d$, $y \in \{-1, 1\}$ and $\alpha \in \{0, e_1, \ldots, e_d\}$. The loss function $f_{(A)} : \mathbb{R}^d \times \mathcal{Z}$ is given by:

$$f_{(A)}(w; (x, y, \alpha)) = y\|(w - \alpha) \odot x\|. \tag{10}$$

We also assume that $d \geq \ln(10)2^n + 1$. Since $f_{(A)}$ is not differentiable when $w = 0$, we define the sub-gradient $\partial f_{(A)}(w) = 0$ at the point $w = 0$. Further, for the rest of the section, whenever clear from the context, we will ignore the subscript $(A)$, and denote the loss function by $f(w; z)$. Additionally we define the following distribution over the samples $z = (x, y, \alpha)$.

**Definition 2.** *For parameters $\delta \in [0, \frac{1}{2}]$, $p \in [0, 1]$, and $a \in \{0, e_1, \ldots, e_d\}$, define the distribution $\mathcal{D}(\delta, p, a)$ over $z = (x, y, \alpha)$ as follows:*

$$x \sim \mathcal{B}(p)^{\otimes d}, \qquad y = 2r - 1 \quad \text{for} \quad r \sim \mathcal{B}(\tfrac{1}{2} + \delta), \qquad \alpha = a.$$

*The components $x$, $y$ and $\alpha$ are sampled independently.*

## B.1    Supporting technical results

The following lemma provides some properties of the population loss under $\mathcal{D}(\delta, p, a)$ given in Definition 2.

**Lemma 5.** *For any $\delta \in [0, \frac{1}{2}]$, $p \in (0, 1]$, and $a \in \{0, e_1, \ldots, e_d\}$, the population loss under $f_{(A)}$ when the data is sampled i.i.d. from $\mathcal{D}(\delta, p, a)$ is convex in the variable $w$. Furthermore, the population loss has a unique minimizer at $w = a$ with $F(a) = 0$, and any $\varepsilon$-suboptimal minimizer $w$ of the population loss must satisfy $\|w - a\| \leq \varepsilon/2\delta p$.*

**Proof.** Let $F$ be the population loss. Using the definition of $F$, we get

$$\begin{aligned}
F(w) &= \mathbb{E}_{(x,y,\alpha) \sim \mathcal{D}(\delta,p,a)}[y\|(w - \alpha) \odot x\|] \\
&= \Pr(y = 1)\, \mathbb{E}_x[\|(w - a) \odot x\|] - \Pr(y = -1)\, \mathbb{E}_{x,\alpha}[\|(w - a) \odot x\|] \\
&= (\tfrac{1}{2} + \delta)\, \mathbb{E}_x[\|(w - a) \odot x\|] - (\tfrac{1}{2} - \delta)\, \mathbb{E}_x[\|(w - a) \odot x\|]
\end{aligned}$$

$$= 2\delta \, \mathbb{E}_x[\|(w - a) \odot x\|].$$

Since, for any $x$ and $a$, the function $\|(w - a) \odot x\|$ is a convex function of $w$, the above formula implies that $F$ is also a convex function of $w$. Furthermore, $F$ is always non-negative and $F(a) = 0$.

Now note that

$$F(w) = 2\delta \, \mathbb{E}_x[\|(w - a) \odot x\|] \geq 2\delta \|(w - a) \odot \mathbb{E}_x[x]\| = 2\delta p \|w - a\|,$$

where the second inequality follows from Jensen's inequality, and the last equality from the fact that $x \sim \mathcal{B}(p)^d$. Now, if $w$ is an $\varepsilon$-suboptimal minimizer of $F$, then since $F(a) = 0$, the above bound implies that $\|w - a\| \leq \varepsilon/2\delta p$. This also implies, in particular using $\varepsilon = a$, that $a$ is a unique minimizer of $F$. $\qquad\square$

The next lemma establishes empirical properties of a dataset $S$ of size $n$ drawn from the distribution $\mathcal{D}(1/10, 1/2, a)$ given in Definition 2.

**Lemma 6.** *Let $j^* \in [d]$. Let $S$ denote a dataset of size $n \leq \log_2(d/\ln(10))$ sampled i.i.d. from a distribution $\mathcal{D}(\frac{1}{10}, \frac{1}{2}, a)$ for some vector $a \in \{0, e_1, \ldots, e_d\}$. Then, with probability at least $0.9$ over the choice of the dataset $S$, there exists an index $\widehat{j} \in [d]$ such that $\widehat{j} \neq j^*$ and $x[\widehat{j}] = 0$ for all $z \in S$ for which $y = 1$ and $x[\widehat{j}] = 1$ for all $z \in S$ for which $y = -1$.*

**Proof.** Let $S$ denote a set of $n$ samples drawn i.i.d. from $\mathcal{D}(\frac{1}{10}, \frac{1}{2}, a)$. We define the sets $S^+$ and $S^-$ as follows:

$$
\begin{aligned}
S^+ &:= \{z_i \in S \mid y_i = +1\}, \\
S^- &:= \{z_i \in S \mid y_i = -1\}.
\end{aligned}
\tag{11}
$$

Let $E_j$ denote the event that the coordinate $x[j] = 0$ for all $z = (x, y) \in S^+$, and $x[j] = 1$ for all $z = (x, y) \in S^-$. Since, for each sample, $x[j]$ is drawn independently from $\mathcal{B}(1/2)$, we have that

$$\Pr(E_j) = \frac{1}{2^n}.$$

Next, let $E$ denote the event that there exists some $\widehat{j} \in [d] \setminus \{j^*\}$ for which $x[\widehat{j}] = 0$ for all $z = (x, y) \in S^+$, and $x[\widehat{j}] = 1$ for all $z = (x, y) \in S^-$. We thus note that,

$$\Pr(E^c) = \Pr\Big( \bigcap_{j \in [d] \setminus \{j^*\}} E_j^c \Big) \overset{(i)}{=} \prod_{j \in [d] \setminus \{j^*\}} \Pr\big(E_j^c\big)$$

where the equality in $(i)$ follows from the fact that the events $\{E_j\}$ are mutually independent to each other. Plugging in the bound for $\Pr(E_j^c)$, we get

$$\Pr(E^c) = \Big(1 - \frac{1}{2^n}\Big)^{d-1}.$$

Using the fact that $1 - a \leq e^{-a}$ for $a \geq 0$, we get

$$\Pr(E^c) \leq e^{-(d-1)/2^n} \leq 0.1,$$

where the second inequality above holds for $n \leq \log_2((d-1)/\ln(10))$. $\qquad\square$

## B.2 Proof of Theorem 2

We now have all the tools required to prove Theorem 2, which states for any regularization function $R(w)$, there exists an instance of a SCO problem for which RERM fails to find $O(1)$-suboptimal solution, in expectation.

**Proof of Theorem 2.** In this proof, we will assume that $n \geq 300$, $d \geq \log(10)2^n + 1$ and the initial point $w_1 = 0$. Assume, for the sake of contradiction, that there exists a regularizer $R : \mathbb{R}^d \to \mathbb{R}$

such that for any SCO instance over $\mathbb{R}^d$ satisfying Assumption II with[2] $L, B = 1$ the expected suboptimality gap for RERM is at most $\varepsilon = 1/20000$. Then, by Markov's inequality, with probability at least 0.9 over the choice of sample set $S$, the suboptimality gap is at most $10\varepsilon$.

Before delving into the proof, we first define some additional notation based on the regularization function $R(\cdot)$. For $j \in [d]$, define the points $w_j^*$ such that

$$w_j^* \in \operatorname*{argmin}_{w \text{ s.t. } \|w - e_j\| \le 100\varepsilon} R(w). \tag{12}$$

and define the index $j^* \in [d]$ such that $j^* \in \operatorname{argmax}_{j \in [d]} R(w_j^*)$.

Now we will construct an instance of SCO in $d = \lceil 2^n \ln(10) + 1 \rceil$ dimensions. The instance will be based on the function $f_{(A)}$ given in (10). The data distribution of interest is $\mathcal{D}_1 := \mathcal{D}(\frac{1}{10}, \frac{1}{2}, e_{j^*})$ (see Definition 2) and suppose that the dataset $S = \{z_i\}_{i=1}^n$ is sampled i.i.d. from $\mathcal{D}_1$. The population loss $F(w)$ corresponding to $\mathcal{D}_1$ is given by

$$F(w) = \mathbb{E}_{z \sim \mathcal{D}_1}\big[f_{(A)}(w; z)\big] = 0.2\,\mathbb{E}_x[\|(w - e_{j^*}) \odot x\|].$$

Clearly, $F(w)$ is convex in $w$ and so minimizing it is an instance of SCO. Furthermore, $f_{(A)}$ is 1-Lipschitz, and the initial point $w_1 = 0$ satisfies $\|w_1 - w^*\| \le 1$. This implies that $f_{(A)}$ satisfies both Assumptions I and II given in (2) and (3) respectively. Additionally, note that $e_{j^*}$ is the unique minimizer of $F(\cdot)$, and as a consequence of Lemma 5, any $10\varepsilon$-suboptimal minimizer $w'$ for $F(\cdot)$ must satisfy

$$\|w' - e_{j^*}\| \le 100\varepsilon. \tag{13}$$

Using the dataset $S$, we define some additional sets as follows:

- Define the set $S^+$ as the set of all the sample points in $S$ for which $y = +1$, i.e

$$S^+ := \{(x, y, \alpha) \in S \mid y = +1\},$$

and define the set $S^- := S \setminus S^+$.

- Define the set $U$ as the set of all the sample points in $S^+$ for which $x[j^*] = 1$, i.e.

$$U := \big\{(x, y, \alpha) \in S^+ \mid x[j^*] = 1\big\}.$$

- Similarly, define the set $V$ as the set of all the sample points in $S^-$ for which $x[j^*] = 1$, i.e.

$$V := \big\{(x, y, \alpha) \in S^- \mid x[j^*] = 1\big\}.$$

Next, define the event $E$ such that all of the following hold:

(a) $|S^-| \ge 7n/20$.

(b) $|U| \le 39n/100$.

(c) $|V| \ge 7n/50$.

(d) There exists $\widehat{j}$ such that $\widehat{j} \ne j^*$ and $x[\widehat{j}] = 0$ for all $z \in S^+$ and $x[\widehat{j}] = 1$ for all $z \in S^-$.

(e) RERM with regularization $R(\cdot)$ and using the dataset $S$ returns an $10\varepsilon$-suboptimal solution for the test loss $F(w)$.

Using Hoeffding's inequality (Lemma 1) and the fact that $\mathbb{E}[|S^-|] = 2n/5$, $\mathbb{E}[|U|] = 3n/10$, and $\mathbb{E}[|V|] = n/5$, we get that parts (a), (b) and (c) hold simultaneously with probability at least 0.3 for $n \ge 300$. Furthermore, by Lemma 6, part (d) holds with probability at least 0.9 since $d \ge \ln(10)2^n + 1$, and part (e) holds with probability 0.9 by our assumption. Hence, the event $E$ occurs with probability at least 0.1. In the following, we condition on the occurrence of the event $E$.

---

[2] The specific values $L, B = 1$ are used for convenience; the proof immediately yields the required SCO instances for arbitray values of $L$ and $B$ that are $O(1)$ in magnitude as a function of $n$.

Consider the point $w_{\widehat{j}}^*$ defined in (12) corresponding to the coordinate $\widehat{j}$ (that occurs in event $E$). By definition, we have that $\|w_{\widehat{j}}^* - e_{\widehat{j}}\| \leq 100\varepsilon$. Thus, we have

$$\|w_{\widehat{j}}^* - e_{j^*}\| \geq \|e_{\widehat{j}} - e_{j^*}\| - \|w_{\widehat{j}}^* - e_{\widehat{j}}\| \geq \sqrt{2} - 100\varepsilon > 100\varepsilon.$$

The first line above follows from Triangle inequality, and the second line holds because $\widehat{j} \neq j^*$ and because $\varepsilon = 1/20000$. As a consequence of the above bound and the condition in (13), we get that the point $w_{\widehat{j}}^*$ is not an $10\varepsilon$-suboptimal point for the population loss $F(\cdot)$, and thus would not be the solution of the RERM algorithm (as the RERM solution is $10\varepsilon$-suboptimal w.r.t $F(\cdot)$). Since, any RERM must satisfy condition (13), we have that

$$\widehat{F}(w_{\widehat{j}}^*) + R(w_{\widehat{j}}^*) > \min_{w: \, \|w - e_{j^*}\| \leq 100\varepsilon} \left( \widehat{F}(w) + R(w) \right)$$

$$\geq \min_{w: \, \|w - e_{j^*}\| \leq 100\varepsilon} \widehat{F}(w) + \min_{w: \, \|w - e_{j^*}\| \leq 100\varepsilon} R(w)$$

$$\geq -100\varepsilon + R(w_{j^*}^*), \tag{14}$$

where $\widehat{F}(w) = \frac{1}{n} \sum_{i=1}^n f(w; z_i)$ denotes the empirical loss on the dataset $S$, and the inequality in the last line follows from the definition of the point $w_{j^*}^*$ and by observing that $\widehat{F}(w) \geq -100\varepsilon$ for any $w$ for which $\|w - e_{j^*}\| \leq 100\varepsilon$ since $\widehat{F}$ is 1-Lipschitz and $\widehat{F}(e_{j^*}) = 0$. For the left hand side, we note that

$$\widehat{F}(w_{\widehat{j}}^*) = \frac{1}{n} \sum_{z \in S} y \|(w_{\widehat{j}}^* - e_{j^*}) \odot x\|$$

$$\overset{(i)}{\leq} 100\varepsilon + \frac{1}{n} \sum_{z \in S} y \|(e_{\widehat{j}} - e_{j^*}) \odot x\|$$

$$\overset{(ii)}{\leq} 100\varepsilon + \frac{1}{n} \left( \sum_{z \in S^+} \|(e_{\widehat{j}} - e_{j^*}) \odot x\| - \sum_{z \in S^-} \|(e_{\widehat{j}} - e_{j^*}) \odot x\| \right)$$

$$\overset{(iii)}{\leq} 100\varepsilon + \frac{1}{n} \left( \sum_{z \in S^+} |x[j^*]| - \sum_{z \in S^+} \sqrt{1 + (x[j^*])^2} \right)$$

$$\overset{(iv)}{\leq} 100\varepsilon + \frac{1}{n} \left( \sum_{z \in U} 1 - \sum_{z \in V} \sqrt{2} - \sum_{z \in S^- \setminus V} 1 \right)$$

$$= 100\varepsilon + \frac{1}{n} \left( |U| - (\sqrt{2} - 1)|V| - |S^-| \right)$$

where the inequality $(i)$ is due to the definition of the point $w_{\widehat{j}}^*$, and because $\widehat{F}$ is 1-Lipschitz, the inequality in $(ii)$ follows from the definition of the sets $S^+$ and $S^-$, the inequality $(iii)$ holds due to the fact that $x[\widehat{j}] = 0$ for all $(x, \alpha) \in S^+$, and $x[\widehat{j}] = 1$ for all $x \in S^-$, and finally, the inequality $(iv)$ follows from the definition of the sets $U$ and $V$. Plugging the bounds on $|U|, |V|$ and $|S^-|$ from the event $E$ defined above, we get that:

$$\widehat{F}(w_{\widehat{j}}^*) \leq 100\varepsilon - \frac{3}{200}. \tag{15}$$

Combining the bounds in (14) and (15) and rearranging the terms, we get that

$$200\varepsilon \geq \frac{3}{200} + R(w_{j^*}^*) - R(w_{\widehat{j}}^*) \geq \frac{3}{200}.$$

where the second inequality above holds because $j^* \in \operatorname{argmax}_{j \in [d]} R(w_j^*)$ (by definition). Thus, $\varepsilon \geq 3/40000 > 1/20000$, a contradiction, as desired.[3]

Finally, note that since the function $f_{(A)}$ is $1$−Lipschitz, the initial point $w_1 = 0$ satisfies $\|w_1 - w^*\| = \|w_1 - e_{j^*}\| \leq 1$ and $F(w)$ is convex, due to Theorem 1, SGD run with a step size of $1/\sqrt{n}$ for $n$ steps learns at a rate of $O(1/\sqrt{n})$. $\qquad \square$

---

[3] The constant in the lower bound for $\varepsilon$ can be improved further via a tighter analysis for the sizes of the sets $|U|, |V|$ and $|S^-|$ in the event $E$.

### B.3 Proof of Corollary 1

The following proof closely follows along the lines of the proof of Theorem 2 above.

**Proof.** In this proof, we will assume that $n \geq 300$, $d \geq \log(10)2^n + 1$ and the initial point $w_1 = 0$. Assume, for the sake of contradiction, that there exists a regularizer $R : \mathbb{R}^d \to \mathbb{R}$ such that for any SCO instance over $\mathbb{R}^d$ satisfying Assumption II with $L, B = 1$, there exists a regularization parameter $\lambda$ such that the expected suboptimality gap for the point $w_{\mathrm{RERM}} = \mathrm{argmin}_{w \in \mathcal{W}} F_S(w) + \lambda R(w)$ is at most $\varepsilon = 1/20000$. Then, by Markov's inequality, with probability at least $0.9$ over the choice of sample set $S$, the suboptimality gap is at most $10\varepsilon$.

We next define the functions $f(w; z)$, the distribution $\mathcal{D}_1$ and the population loss function $F(w)$ identical to the the corresponding quantities in the proof of Theorem 2. Furthermore, we also define the points $w_j$ for $j \in [d]$, the coordinates $j^*$ and the event $E$ identical to the corresponding definitions in the proof of Theorem 2. As we argued in the proof of Theorem 2 above, we note that any $10\varepsilon$-suboptimal minimizer $w'$ for $F(\cdot)$ must satisfy

$$\|w' - e_{j^*}\| \leq 100\varepsilon. \tag{16}$$

Thus, the point $w_{\widehat{j}}^*$ (where $\widehat{j}$ is defined in the event $E$) is not an $10\varepsilon$-suboptimal point for the population loss $F(\cdot)$, and thus for any regularization parameter $\lambda$ (that can even depend on the dataset $S$) should not correspond to the RERM solution (as the point $w_{\mathrm{RERM}}$ is $10\varepsilon$-suboptimal with respect to $F(\cdot)$). Since, any RERM must satisfy condition (16), we have that for any regularization parameter $\lambda$ that can even depend on the dataset $S$,

$$
\begin{aligned}
\widehat{F}(w_{\widehat{j}}^*) + \lambda R(w_{\widehat{j}}^*) &> \min_{w:\, \|w - e_{j^*}\| \leq 100\varepsilon} \left( \widehat{F}(w) + \lambda R(w) \right) \\
&\geq \min_{w:\, \|w - e_{j^*}\| \leq 100\varepsilon} \widehat{F}(w) + \min_{w:\, \|w - e_{j^*}\| \leq 100\varepsilon} \lambda R(w) \\
&\geq -100\varepsilon + \lambda R(w_{j^*}^*), \tag{17}
\end{aligned}
$$

where the inequality in the last line follows from the definition of the point $w_{j^*}^*$ and by observing that $\widehat{F}(w) \geq -100\varepsilon$ for any $w$ for which $\|w - e_{j^*}\| \leq 100\varepsilon$ since $\widehat{F}$ is 1-Lipschitz and $\widehat{F}(e_{j^*}) = 0$. For the left hand side, similar to the proof of Theorem 2, we upper bound

$$\widehat{F}(w_{\widehat{j}}^*) \leq 100\varepsilon - \frac{3}{200}. \tag{18}$$

Combining the bounds in (17) and (18) and rearranging the terms, we get that

$$200\varepsilon \geq \frac{3}{200} + \lambda R(w_{j^*}^*) - \lambda R(w_{\widehat{j}}^*) \geq \frac{3}{200}.$$

where the second inequality above holds because $j^* \in \mathrm{argmax}_{j \in [d]} R(w_j^*)$ (by definition). Thus, $\varepsilon \geq 3/40000 > 1/20000$, a contradiction, as desired.

We remark that in the above proof the regularization parameter $\lambda$ can be arbitrary and can even depend on the dataset $S$, but $\lambda$ should not depend on $w$ as this will change the definition of the points $w_j^*$. Only the regularization function $R(\cdot)$ is allowed to depend on $w$. $\qquad\square$

## C  Missing proofs from Section 4

In this section, we first provide a learning problem for which, for any $\eta \in [1/n^2, 1)$ and $T \in [1, n^3)$, the point $\widehat{w}_{\eta,T}^{\mathrm{GD}}$ returned by running GD algorithm with step size $\eta$ for $T$ time steps has the lower bound

$$\mathbb{E}[F(\widehat{w}_{\eta,T}^{\mathrm{GD}})] - \inf_{w \in \mathbb{R}^d} F(w) = \Omega\left( \frac{1}{\log^4(n)} \min\left\{ \eta\sqrt{T} + \frac{1}{\eta T} + \frac{\eta T}{n}, 1 \right\} \right). \tag{19}$$

Then, we will provide a learning setting in which GD algorithm when run with step size $\eta \leq 1/64n^{5/4}$ has the lower bound of $\Omega(1/n^{3/8})$ for all $T \geq 0$. Our final lower bound for GD algorithm, given in Theorem 3 then follows by considering the above two lower bound constructions together.

## C.1 Modification of Amir et al. [2021] lower bound

Amir et al. [2021] recently provided the following lower bound on the performance of GD algorithm.

**Theorem 9** (Modification of Theorem 3.1, Amir et al. [2021]). *Fix any $n$, $\bar{\eta}$ and $\bar{T}$. There exists a function $f(w; z)$ that is 4-Lipschitz and convex in $w$, and a distribution $\mathcal{D}$ over the instance space $\mathcal{Z}$, such that for any $\eta \in [\bar{\eta}, \bar{\eta}\sqrt{3/2})$ and $T \in [\bar{T}, 2\bar{T})$, the point $\widehat{w}^{GD}[\eta, T]$ returned by running GD algorithm with a step size of $\eta$ for $T$ steps has excess risk*

$$\mathbb{E}[F(\widehat{w}^{GD}[\eta, T])] - \inf_{w \in \mathbb{R}^d} F(w) \geq \Omega\Big(\min\Big\{\eta\sqrt{T} + \frac{1}{\eta T}, 1\Big\}\Big), \tag{20}$$

*where $F(w) := \mathbb{E}_{z \sim \mathcal{D}}[f(w; z)]$. Additionally, there exists a point $w^* \in \arg\min_w F(w)$ such that $\|w^*\| \leq 1$.*

**Proof.** We refer the reader to Amir et al. [2021] for full details about the loss function construction and the lower bound proof, and discuss the modifications below:

- Amir et al. [2021] provide the lower bound for a fixed $\eta$ and $T$. In particular, their loss function construction given in eqn-(16) (in their paper) consists of parameters $\gamma_1, \gamma_2, \gamma_3, \varepsilon_1, \ldots, \varepsilon_3$ and $d$ which are chosen depending $\eta$ and $T$. However, a slight modification of these parameters easily extends the lower bound to hold for all $\eta \in [\bar{\eta}, \bar{\eta}\sqrt{3/2})$ and $T \in [\bar{T}, 2\bar{T})$. The modified parameter values are:

  - Set $d = \frac{2^n}{3\bar{\eta}^2}$.
  - Set $\gamma_1$ such that $\gamma_1(1 + 5\bar{\eta}\bar{T})\sqrt{d} \leq \frac{1}{64}\min\{\bar{\eta}\sqrt{T}, \frac{1}{3}\}$.
  - Set $\gamma_2 = 2\gamma_1\bar{\eta}\bar{T}$ and $\gamma_3 = 1$.
  - Set $0 < \varepsilon_1 < \cdots < \varepsilon_d < \frac{\gamma_1\bar{\eta}}{2n}$.

  Their proof of the lower bound in Theorem 3.1 follows using Lemma 4.1, Theorem 6.1, Lemma 6.2 and Claim 6.3 (in their paper respectively). We note that the above parameter setting satisfies the premise of Lemma 4.1 and Claim 6.3 for all $\eta \in [\bar{\eta}, \bar{\eta}\sqrt{3/2})$ and $T \in [\bar{T}, 2\bar{T})$. Furthermore, it can be easily verified that the proofs of Theorem 6.1 and Lemma 6.2 also follow through with the above parameter setting for all $\eta \in [\bar{\eta}, \bar{\eta}\sqrt{3/2})$ and $T \in [\bar{T}, 2\bar{T})$. Thus, the desired lower bound holds for all $\eta \in [\bar{\eta}, \bar{\eta}\sqrt{3/2})$ and $T \in [\bar{T}, 2\bar{T})$.

- Amir et al. [2021] consider GD with projection on the unit ball as their training algorithm. However, their lower bound also holds for GD without the projection step (as we consider in our paper; see (5)). In fact, as pointed out in the proof of Lemma 4.1 (used to Prove Theorem 3.1) in their paper, the iterates $w_t$ generated by the GD algorithm never leave the unit ball. Thus, the projection step is never invoked, and GD with projection and GD without projection produce identical iterates.

The upper bound on $\|w^*\|$ also follows from the provided proof in Amir et al. [2021]. As they show in the proof of Lemma 4.1 (page 28, last paragraph in the proof), all the GD iterates as well as the minimizer point $w^*$ lie a ball of unit radius.

Finally, note that the loss function in Theorem 9 is 4-Lipschitz, and bounded over the unit ball which contains all the iterates generated by GD algorithm. Thus, a simple application of the Markov's inequality suggests that the lower bound in (20) also holds with constant probability. However, for our purposes, the in-expectation result suffices. $\square$

The loss function construction in Theorem 9 depends on $\bar{\eta}$ and $\bar{T}$, and thus the lower bound above only holds when the GD algorithm is run with step size $\eta \in [\bar{\eta}, \bar{\eta}\sqrt{3/2})$ and for $T \in [\bar{T}, 2\bar{T})$. In the following, we combine together multiple such lower bound instances to get an anytime and any stepsize guarantee.

**Theorem 10.** *Fix any $n \geq 200$. There exists a function $f(w; z)$ that is 1-Lipschitz and convex in $w$, and a distribution $\mathcal{D}$ over $\mathcal{Z}$ such that, for any $\eta' \in [1/n^2, 1)$ and $T' \in [1, n^3)$, the point $\widehat{w}^{GD}[\eta', T']$*

*returned by running GD algorithm with a step size of $\eta'$ for $T'$ steps satisfies the lower bound*

$$\mathbb{E}[F(\widehat{w}^{GD}[\eta', T'])] - \inf_{w \in \mathbb{R}^d} F(w) \geq \Omega\left(\frac{1}{\log^4(n)} \min\left\{\eta'\sqrt{T'} + \frac{1}{\eta'T'}, 1\right\}\right), \qquad (21)$$

*where $F(w) := \mathbb{E}_{z \sim \mathcal{D}}[f(w; z)]$. Additionally, there exists a minimizer $w^* \in \arg\min_w F(w)$ such that $\|w^*\| = O(1)$.*

**Proof.** Set $\gamma := \sqrt{3/2}$. We consider a discretization of the intervals $[1/n^3, 1)$ for the step sizes (we take a slightly larger interval that the domain $[1/n^2, 1)$ of the step size) and $[1, n^3)$ for the time steps respectively. Define the set

$$\mathcal{N} := \left\{\frac{1}{n^3}, \frac{\gamma}{n^3}, \frac{\gamma^2}{n^3}, \cdots, \frac{\gamma^{\lceil 3\log(n)/\log(\gamma)\rceil}}{n^3}\right\}$$

of step sizes such that for any $\eta' \in [1/n^3, 1)$, there exists an $\eta \in \mathcal{N}$ that satisfies $\eta \leq \eta' < \gamma\eta$. Similarly, define the set

$$\mathcal{T} := \left\{1, 2, 4, \ldots, 2^{\lceil 3\log(n)\rceil}\right\}$$

of time steps such that for any $T' \in [1, n^3)$, there exists a $T \in \mathcal{T}$ that satisfies $T \leq T' < 2T$. Further, define $M = |\mathcal{N}||\mathcal{T}|$. Clearly, $M = \lceil 3\log(n)/\log(\gamma)\rceil \cdot \lceil 3\log(n)\rceil$ and for $n \geq 20$ satisfies the bound $40\log^2(n) \leq M \leq 80\log^2(n)$.

In the following, we first define the component function $f_{\eta,T}$ for every $\eta \in \mathcal{N}$ and $T \in \mathcal{T}$. We then define the loss function $f$ and show that it is convex and Lipschitz in the corresponding optimization variable. Finally, we show the lower bound for GD for this loss function $f$ for any step size $\eta \in [1/n^2, 1)$ and time steps $T \in [1, n^3)$.

**Component functions.** For any $\eta \in \mathcal{N}$ and $T \in \mathcal{T}$, let $\bar{w}_{\eta,T}$, $z_{\eta,T}$, $f_{\eta,T}$ and $\mathcal{D}_{\eta,T}$ denote the optimization variable, data instance, loss function and the corresponding data distribution in the lower bound construction in Theorem 9 where $\bar{\eta}$ and $\bar{T}$ are set as $\eta$ and $T$ respectively. We note that:

(a) For any $z_{\eta,T}$, the function $f_{\eta,T}(\bar{w}_{\eta,T}; z_{\eta,T})$ is 4-Lipschitz and convex in $\bar{w}_{\eta,T}$.

(b) For any $\eta'' \in [\eta, \gamma\eta)$ and $T'' \in [T, 2T)$, the output point[4] $\widehat{w}^{GD}_{\eta,T}[\eta'', T'']$ returned by running GD algorithm with a step size of $\eta''$ for $T''$ steps has excess risk

$$\mathbb{E}\left[F_{\eta,T}\left(\widehat{w}^{GD}_{\eta,T}[\eta'', T'']\right)\right] - \inf_{\bar{w}_{\eta,T}} F_{\eta,T}(\bar{w}_{\eta,T}) \geq \Omega\left(\min\left\{\eta''\sqrt{T''} + \frac{1}{\eta''T''}, 1\right\}\right), \qquad (22)$$

where the population loss $F_{\eta,T}(\bar{w}_{\eta,T}) := \mathbb{E}_{z_{\eta,T} \sim \mathcal{D}_{\eta,T}}[f_{\eta,T}(\bar{w}_{\eta,T}; z_{\eta,T})]$.

(c) There exists a point $\bar{w}^*_{\eta,T} \in \arg\min F_{\eta,T}(\bar{w}_{\eta,T})$ such that $\|\bar{w}^*_{\eta,T}\| \leq 1$.

**Lower bound construction.** We now present our lower bound construction:

- *Optimization variable*: For any $\eta$ and $T$, define $w_{\eta,T} := \bar{w}_{\eta,T}/\log(n)$. The optimization variable $w$ is defined as the concatenation of the variables $(w_{\eta,T})_{\eta \in \mathcal{N}, T \in \mathcal{T}}$.

- *Data instance*: $z$ is defined as the concatenation of the data instances $(z_{\eta,T})_{\eta \in \mathcal{N}, T \in \mathcal{T}}$.

- *Data distribution*: $\mathcal{D}$ is defined as the cross product of the distributions $(\mathcal{D}_{\eta,T})_{\eta \in \mathcal{N}, T \in \mathcal{T}}$. Thus, for any $\eta \in \mathcal{N}$ and $T \in \mathcal{T}$, the component $z_{\eta,T}$ is sampled independent from $\mathcal{D}_{\eta,T}$.

- *Loss function*: is defined as

$$f(w; z) = \frac{1}{M} \sum_{\eta \in \mathcal{N}, T \in \mathcal{T}} f_{\eta,T}(\bar{w}_{\eta,T}; z_{\eta,T}), \qquad (23)$$

  where recall that $\bar{w}_{\eta,T} = w_{\eta,T}\log(n)$. Additionally, we define the population loss $F(w) := \mathbb{E}_{z \sim \mathcal{D}}[f(w; z)]$.

---

[4]We use the notation $\widehat{w}^{GD}_{\eta,T}[\eta'', T'']$ to denote the value of the variable $\bar{w}_{\eta,T}$ computed by running GD algorithm with the step size $\eta''$ for $T''$ time steps. The subscripts $\eta, T$ are used to denote the fact that the corresponding variables are associated with the loss function construction in Theorem 9 where $\bar{\eta}$ and $\bar{T}$ are set as $\eta$ and $T$ respectively.

**$f$ is convex and 1-Lipschitz.** Since, for any $\eta \in \mathcal{N}$ and $T \in \mathcal{T}$, the function $f_{\eta,T}(\bar{w}_{\eta,T}; z_{\eta,T})$ is convex in $\bar{w}_{\eta,T}$ for every $z_{\eta,T}$, and since $w_{\eta,T} = \bar{w}_{\eta,T}/\log(n)$, we immediately get that the function $f(w; z)$ is also convex in $w$ for every $z$. Furthermore, for any $w$, $w'$ and $z$, we have

$$
\begin{aligned}
|f(w;z) - f(w';z)| &\overset{(i)}{\leq} \frac{1}{M} \sum_{\eta \in \mathcal{N}, T \in \mathcal{T}} \left| f_{\eta,T}(\bar{w}_{\eta,T}; z_{\eta,T}) - f_{\eta,T}(\bar{w}'_{\eta,T}; z_{\eta,T}) \right| \\
&\overset{(ii)}{\leq} \frac{4\log(n)}{M} \sum_{\eta \in \mathcal{N}, T \in \mathcal{T}} \|w_{\eta,T} - w'_{\eta,T}\| \\
&= \frac{4\log(n)}{M} \sum_{\eta \in \mathcal{N}, T \in \mathcal{T}} \sqrt{\|w_{\eta,T} - w'_{\eta,T}\|^2} \\
&\overset{(iii)}{\leq} 4\log(n) \sqrt{\frac{1}{M} \sum_{\eta \in \mathcal{N}, T \in \mathcal{T}} \|w_{\eta,T} - w'_{\eta,T}\|^2} \\
&= \frac{4\log(n)}{\sqrt{M}} \sqrt{\sum_{\eta \in \mathcal{N}, T \in \mathcal{T}} \|w_{\eta,T} - w'_{\eta,T}\|^2} \\
&= \frac{4\log(n)}{\sqrt{M}} \sqrt{\sum_{\eta \in \mathcal{N}, T \in \mathcal{T}} \|w_{\eta,T} - w'_{\eta,T}\|^2} \\
&\overset{(iv)}{=} \frac{4\log(n)}{\sqrt{M}} \sqrt{\|w - w'\|^2} \\
&\overset{(v)}{\leq} \|w - w'\|,
\end{aligned}
$$

where the inequality $(i)$ follows from Triangle inequality and the inequality $(ii)$ holds because $f_{\eta,T}$ is 4-Lipschitz in the variable $\bar{w}_{\eta,T}$ for every $\eta \in \mathcal{N}$ and $T \in \mathcal{T}$, and by using the fact that $\bar{w}_{\eta,T} = w_{\eta,T}\log(n)$. The inequality $(iii)$ above follows from an application of Jensen's inequality and using the concavity of square root. Furthermore, the equality $(iv)$ holds by the construction of the variable $w$ as concatenation of the variables $(w_{\eta,T})_{\eta \in \mathcal{N}, T \in \mathcal{T}}$. Finally, the inequality $(v)$ follows from the fact that $M \geq 16\log^2(n)$ for $n \geq 4$. Thus, the function $f(w; z)$ is 1-Lipschitz in $w$ for every $z$.

**Bound on the minimizer.** Since the components $(w_{\eta,T})_{\eta \in \mathcal{N}, T \in \mathcal{T}}$ do not interact with each other in the loss function $f$, any point $w^* \in \arg\min F(w)$ satisfies:

$$
\|w^*\| = \sqrt{\sum_{\eta \in \mathcal{N}, T \in \mathcal{T}} \|w^*_{\eta,T}\|^2} = \frac{1}{\log(n)} \sqrt{\sum_{\eta \in \mathcal{N}, T \in \mathcal{T}} \|\bar{w}^*_{\eta,T}\|^2},
$$

where $\bar{w}^*_{\eta,T}$ denote the corresponding minimizers of the population loss $F_{\eta,T}$. Due to [Theorem 9](#), we have that for any $\eta$ and $T$, there exists a $\bar{w}^*_{\eta,T}$ that satisfies $\|\bar{w}^*_{\eta,T}\| \leq 1$. Plugging these in the above, we get that there exists a $w^*$ for which

$$
\|w^*\| \leq \frac{1}{\log(n)}\sqrt{M} \leq 10,
$$

where the second inequality follows by using the fact that $M \leq 100\log^2(n)$ for $n \geq 20$.

**Lower bound proof.** We next provide the desired lower bound for the loss function in [(23)](#). First, note that when running GD update on the variable $w$, with a step size of $\eta'$ and using the loss function $f(w; z)$, each of the component variables $\bar{w}_{\eta,T}$ are updated as if independently performing GD on the function $f_{\eta,T}$ but with the step size of $\eta'/M$.

Now, suppose that we run GD on the loss function $f$ with step size $\eta' \in [1/n^2, 1)$ and for $T' \in [1, n^3)$ steps. For $n \geq 20$, the step size $\widetilde{\eta} := \eta'/M \leq \eta/40\log^2(n)$ with which each component is updated clearly satisfies $\widetilde{\eta} \in [1/n^3, 1]$. Thus, by construction of the sets $\mathcal{N}$ and $\mathcal{T}$, there exists some $\bar{\eta} \in \mathcal{N}$ and $\bar{T} \in \mathcal{T}$ such that $\bar{\eta} \leq \widetilde{\eta}' < \gamma\bar{\eta}$ and $\bar{T} \leq T' < 2\bar{T}$. Thus, due to [(23)](#), we have that for the

component function corresponding to $(\bar\eta, \bar T)$, the point $\widehat{w}_{\bar\eta,\bar T}^{\text{GD}}[\widetilde\eta, T']$ returned after running GD with step size $\widetilde\eta$ for $T'$ steps satisfies

$$\mathbb{E}\big[F_{\bar\eta,\bar T}\big(\widehat{w}_{\bar\eta,\bar T}^{\text{GD}}[\widetilde\eta, T']\big)\big] - \inf_{\bar w_{\bar\eta,\bar T}} F_{\bar\eta,\bar T}(\bar w_{\bar\eta,\bar T}) \geq \Omega\Big(\min\Big\{\widetilde\eta\sqrt{T'} + \frac{1}{\widetilde\eta\,T'}, 1\Big\}\Big)$$
$$\geq \Omega\Big(\frac{1}{M}\min\Big\{\eta'\sqrt{T'} + \frac{1}{\eta'T'}, 1\Big\}\Big), \qquad (24)$$

where the last line holds for $\widetilde\eta = \eta'/M$.

Our desired lower bound follows immediately from (24). Let $\widehat{w}^{\text{GD}}[\eta', T']$ be the output of running GD algorithm on the function $f$ with step size $\eta'$ and for $T'$ steps. We have that

$$\mathbb{E}\big[F\big(\widehat{w}^{\text{GD}}[\eta', T']\big)\big] - \min_w F(w) = \frac{1}{M}\sum_{\eta\in\mathcal{N}, T\in\mathcal{T}}\Big(\mathbb{E}\big[F_{\eta,T}\big(\widehat{w}_{\eta,T}^{\text{GD}}[\widetilde\eta, T']\big)\big] - \inf_{w_{\eta,T}} F_{\eta,T}(w_{\eta,T})\Big)$$
$$\geq \frac{1}{M}\Big(\mathbb{E}\big[F_{\bar\eta,\bar T}\big(\widehat{w}_{\bar\eta,\bar T}^{\text{GD}}[\widetilde\eta, T']\big)\big] - \inf_{w_{\bar\eta,\bar T}} F_{\bar\eta,\bar T}(w_{\bar\eta,\bar T})\Big)$$
$$= \Omega\Big(\frac{1}{M^2}\min\Big\{\eta'\sqrt{T'} + \frac{1}{\eta'T'}, 1\Big\}\Big),$$

where the equality in the first line holds because the variables $\{w_{\eta,T}\}$ do not interact with each other, the inequality in the second line follows by ignoring rest of the terms which are all guaranteed to be positive, and finally, the last line follows by plugging in (24). Using the fact that $M \geq 40\log^2(n)$ in the above, we get that for any $\eta' \in [1/n^2, 1)$ and for $T' \in [1, n^3)$, the point $\widehat{w}^{\text{GD}}[\eta', T']$ satisfies

$$\mathbb{E}\big[F(\widehat{w}^{\text{GD}}[\eta', T'])\big] - \min_w F(w) = \Omega\Big(\frac{1}{\log^4(n)}\min\Big\{\eta'\sqrt{T'} + \frac{1}{\eta'T'}, 1\Big\}\Big).$$

$\square$

## C.2   Lower bound of $\eta T/n$

The lower bound in (21) already matches the first two terms in our desired lower bound in (19). In the following, we provide a function for which GD algorithm has expected suboptimality of $\eta T/n$.

**Lemma 7.** *Fix any $n$. Let $w \in \mathbb{R}$ denote the optimization variable and $z$ denote a data sample from the instance space $\mathcal{Z} = \{-1, 1\}$. There exists a 2-Lipschitz function $f(w; z)$ and a distribution $\mathcal{D}$ over $\mathcal{Z}$ such that:*

(a) *The population loss $F(w) := \mathbb{E}_{z\sim\mathcal{D}}[f(w; z)]$ is convex in $w$. Furthermore, $w^* = 0$ is the unique minimizer of the population loss $F(w)$.*

(b) *For any $\eta$ and $T$, the point $\widehat{w}^{GD}[\eta, T]$ returned by running GD with a step size of $\eta$ for $T$ steps satisfies*

$$\mathbb{E}\big[F(\widehat{w}^{GD}[\eta, T])\big] - \inf_{w\in\mathbb{R}} F(w) \geq \Omega\Big(\frac{\eta T}{n}\Big).$$

**Proof.** For $z \in \{-1, 1\}$ and $w \in \mathbb{R}$, define the instance loss function $f$ as

$$f(w; z) := \Big(\frac{1}{4\sqrt{n}} + z\Big)|w|.$$

Define the distribution $\mathcal{D}$ such that $z = +1$ or $z = -1$ with probability $1/2$ each. Clearly, $f(w; z)$ is 2-Lispchitz w.r.t. $w$ for any $z \in \mathcal{Z}$. Furthermore, for any $w \in \mathbb{R}$,

$$F(w) = \mathbb{E}_{z\sim\mathcal{D}}[f(w; z)] = \mathbb{E}_z\Big[\Big(\frac{1}{4\sqrt{n}} + z\Big)|w|\Big] = \frac{1}{4\sqrt{n}}|w|,$$

where the last equality holds because $\mathbb{E}[z] = 0$. Thus, $F(w)$ is convex in $w$ and $w^* = 0$ is the unique minimizer of the population loss $F(w)$. This proves part-(a).

We now provide a lower bound for GD algorithm. Let $S = \{z_i\}_{i=1}^n$ denote a dataset of size $n$ sampled i.i.d. from $\mathcal{D}$. The update rule for GD algorithm from (5) implies that

$$w_{t+1}^{\text{GD}} \leftarrow w_t^{\text{GD}} - \eta \cdot \text{sign}(w_t^{\text{GD}}) \cdot \Big(\frac{1}{4\sqrt{n}} + \frac{1}{n}\sum_{i=1}^n z_i\Big), \tag{25}$$

and finally the returned point is given by $\widehat{w}^{\text{GD}}[\eta, T] = \frac{1}{n}\sum_{t=1}^T w_t^{\text{GD}}$.

For $i \in [n]$, define the random variables $y_i = (1 - z_i)/2$. Note that $y_i \sim \mathcal{B}(1/2)$, and thus Lemma 3 implies that

$$\sum_{i=1}^n y_i \geq \frac{n}{2} + \frac{\sqrt{n}}{4},$$

with probability at least $1/15e$. Rearranging the terms, we get that $\sum_{i=1}^n z_i \leq -\sqrt{n}/2$ with probability at least $1/15e$. Plugging this in (25), we have that with probability at least $1/15e$, for all $t \geq 0$,

$$w_{t+1}^{\text{GD}} \geq w_t^{\text{GD}} + \frac{\eta}{4\sqrt{n}}\text{sign}(w_t^{\text{GD}}).$$

Without loss of generality, assume that $w_1 > 0$. In this case, the above update rule implies that

$$w_t \geq w_1 + \frac{t\eta}{4\sqrt{n}}$$

for all $t \geq 0$, which further implies that

$$\widehat{w}^{\text{GD}}[\eta, T] = \frac{1}{T}\sum_{t=1}^T w_t \geq w_1 + \frac{\eta(T-1)}{8\sqrt{n}}.$$

Thus,

$$F(\widehat{w}^{\text{GD}}[\eta, T]) - \inf_w F(w) = F(\widehat{w}^{\text{GD}}[\eta, T]) \geq \frac{w_1}{4\sqrt{n}} + \frac{\eta(T-1)}{32n},$$

giving us the desired lower bound on the performance guarantee of the returned point $\widehat{w}^{\text{GD}}[\eta, T]$. The final in-expectation statement follows by observing that the above holds with probability at least $1/15e$. The proof follows similarly when $w_1 \leq 0$. $\qquad\square$

We next prove the lower bound in (19) which follows by combining the lower bound construction from Theorem 9 and Lemma 7.

**Theorem 11.** *Fix any $n \geq 200$. There exists a 3-Lipschitz function $f(w; z)$ and a distribution $\mathcal{D}$ over $z$, such that:*

(a) *The population loss $F(w) := \mathbb{E}_{z \sim \mathcal{D}}[f(w; z)]$ is convex in $w$. Furthermore, there exists a $w^* \in \arg\min_w F(w)$ such that $\|w^*\| = O(1)$.*

(b) *For any $\eta \in [1/n^2, 1)$ and $T \geq 1$, the point $\widehat{w}^{GD}[\eta, T]$ returned by running GD with a step size of $\eta$ for $T$ steps has excess risk*

$$\mathbb{E}\big[F(\widehat{w}^{GD}[\eta, T])\big] - \inf_{w \in \mathbb{R}^d} F(w) = \Omega\Big(\frac{1}{\log^4(n)}\min\Big\{\eta\sqrt{T} + \frac{1}{\eta T} + \frac{\eta T}{n}, 1\Big\}\Big). \tag{26}$$

**Proof.** We first define some additional notation. Let $w_{(1)}$, $z_{(1)}$, $f_{(1)}$ and $\mathcal{D}_{(1)}$ denote the optimization variable, the data sample, the instance loss and the distribution over $z_{(1)}$ corresponding to the lower bound construction in Theorem 10. Additionally, let $F_{(1)}(w_{(1)})$ denote the corresponding population loss under the distribution $D_{(1)}$. We note that the function $f_{(1)}$ is 1-Lipschitz in $w_{(1)}$ for any $z_{(1)}$. Furthermore, Theorem 10 implies that $F_{(1)}(w_{(1)})$ is convex in $w_{(1)}$, there exists a minimizer $w_{(1)}^* \in \arg\min_{w_{(1)}} F_{(1)}(w_{(1)})$ such that $\|w_{(1)}^*\| = O(1)$, and that for any $\eta \in [1/n^2, 1)$

and $T \in [1, n^3)$, the point $\widehat{w}_{(1)}^{\mathrm{GD}}[\eta, T]$ returned by running GD algorithm with step size $\eta$ for $T$ time steps satisfies

$$\mathbb{E}\big[F_{(1)}(\widehat{w}_{(1)}^{\mathrm{GD}}[\eta, T])\big] - \inf_{w_{(1)}} F(w_{(1)}) = \Omega\Big(\frac{1}{\log^4(n)} \min\Big\{\eta\sqrt{T} + \frac{1}{\eta T}, 1\Big\}\Big). \qquad (27)$$

Similarly, let $w_{(2)}$, $z_{(2)}$, $f_{(2)}$ and $\mathcal{D}_{(2)}$ denote the corresponding quantities for the lower bound construction in Lemma 7, and let $F_{(2)}(w_{(2)})$ denote the corresponding population loss under the distribution $\mathcal{D}_{(2)}$. We note that the function $f_{(2)}$ is 2-Lipschitz in $w_{(2)}$ for any $z_{(2)}$. Furthermore, Lemma 7 implies that $F_{(2)}(w_{(2)})$ is convex in $w_{(2)}$ with $w^* = 0$ being the unique minimizer, and that for any $\eta$ and $T$ the point $\widehat{w}_{(2)}^{\mathrm{GD}}[\eta, T]$ returned by running GD algorithm with step size $\eta$ for $T$ time steps satisfies

$$\mathbb{E}\big[F_{(2)}(\widehat{w}_{(2)}^{\mathrm{GD}}[\eta, T])\big] - \inf_{w_{(2)}} F(w_{(2)}) = \Omega\Big(\frac{\eta T}{n}\Big). \qquad (28)$$

Our desired lower bound follows by combining the lower bound constructions from Theorem 9 and Lemma 7 respectively.

**Lower bound construction.** Consider the following learning setting:

- *Optimization variable*: $w$ is defined as the concatenation of the variables $(w_{(1)}, w_{(2)})$.

- *Data instance*: $z$ is defined as the concatenation of the data instances $(z_{(1)}, z_{(2)})$.

- *Data distribution*: $\mathcal{D}$ is defined as $\mathcal{D}_{(1)} \times \mathcal{D}_{(2)}$, i.e. $z_{(1)}$ and $z_{(2)}$ are sampled independently from $\mathcal{D}_{(1)}$ and $\mathcal{D}_{(2)}$ respectively.

- *Loss function*: is defined as

$$f(w; z) := f_{(1)}(w_{(1)}; z_{(1)}) + f_{(2)}(w_{(2)}; z_{(2)}),$$

  Additionally, define the population loss $F(w) := \mathbb{E}_{z \sim \mathcal{D}}[f(w; z)]$.

Since, $f_{(1)}$ is 1-Lipschitz in $w_{(1)}$ and $f_{(2)}$ is 2-Lipschitz in $w_{(2)}$, we have that the function $f$ defined above is 3-Lipschitz in $w$. Furthermore, the population loss

$$F(w) = F_{(1)}(w_{(1)}) + F_{(2)}(w_{(2)}),$$

is convex in $w$ as both $F_{(1)}(w_{(1)})$ and $F_{(2)}(w_{(2)})$ are convex functions. Furthermore, since the components $(w_{(1)}, w_{(2)})$ do not interact with each other in the function $f$, we have that there exists a $w^* \in \mathrm{argmin}_w F(w)$ such that:

$$\|w^*\| \leq \|w_{(1)}^*\| + \|w_{(2)}^*\| = O(1),$$

where $w_{(1)}^*$ denotes a minimizer of $F_{(1)}$ with $\|w_{(1)}^*\| = O(1)$ and $w_{(1)}^* = 0$ denotes the unique minimizer of $F_{(2)}$.

**GD lower bound.** From the construction of the function $f$, we note that the variables $w_{(1)}$ and $w_{(2)}$ are updated independent to each other by GD algorithm. Thus, for any $\eta \in [1/n^2, 1)$ and $T \in [1, n^3)$, the point $\widehat{w}^{\mathrm{GD}}[\eta, T]$ returned by running GD algorithm on the function $f$ with step size $\eta$ for $T$ time steps satisfies

$$\mathbb{E}\big[F\big(\widehat{w}^{\mathrm{GD}}[\eta, T]\big)\big] - \min_w F(w) = \mathbb{E}\big[F_{(1)}\big(\widehat{w}_{(1)}^{\mathrm{GD}}[\eta, T]\big) + F\big(\widehat{w}_{(2)}^{\mathrm{GD}}[\eta, T]\big)\big] - \min_{w_{(1)}, w_{(2)}} F_{(1)}(w_{(1)}) + F_{(2)}(w_{(2)})$$

$$= \mathbb{E}\big[F_{(1)}\big(\widehat{w}_{(1)}^{\mathrm{GD}}[\eta, T]\big)\big] - \min_{w_{(1)}} F_{(1)}(w_{(1)}) + \mathbb{E}\big[F\big(\widehat{w}_{(2)}^{\mathrm{GD}}[\eta, T]\big)\big] - \min_{w_{(2)}} F(w_{(2)})$$

$$\overset{(i)}{=} \Omega\Big(\frac{1}{\log^4(n)} \min\Big\{\eta\sqrt{T} + \frac{1}{\eta T}, 1\Big\}\Big) + \Omega\Big(\frac{\eta T}{n}\Big)$$

$$= \Omega\Big(\frac{1}{\log^4(n)} \min\Big\{\eta\sqrt{T} + \frac{1}{\eta T} + \frac{\eta T}{n}, 1\Big\}\Big),$$

where the lower bound in $(i)$ follows from combining the lower bounds in (27) and (28).

Finally, we note that when $\eta \in [1/n^2, 1)$ and $T \geq n^3$, we have that

$$\mathbb{E}\big[F(\widehat{w}^{\mathrm{GD}}[\eta, T])\big] - \min_w F(w) \geq \mathbb{E}\big[F(\widehat{w}^{\mathrm{GD}}_{(2)}[\eta, T])\big] - \min_{(2)} F_{(2)}(w_{(2)})$$

$$= \Omega\Big(\frac{\eta T}{n}\Big)$$

$$= \Omega(1),$$

where the second line follows by using the lower bound in (28), and the last line holds for $T > n^3$ because $\eta \geq 1/n^2$. Thus, the desired lower bound holds for all $\eta \in [1/n^2, 1]$ and $T \geq 1$. $\qquad\square$

### C.3 Lower bound for small step size ($\eta < 1/64n^{5/4}$)

In the following, we provide a learning setting for which GD run with step size $\eta < 1/64n^{5/4}$ has lower bound of $\Omega(1/n^{3/8})$.

**Lemma 8.** *Let $w \in \mathbb{R}$ denote the optimization variable, and $z$ denote a data sample. There exists a function $f(w; z)$ and a distribution $\mathcal{D}$ over $z$ such that:*

(a) *The population loss $F(w) := \mathbb{E}_{z \sim \mathcal{D}}[f(w; z)]$ is $1$-Lipschitz and convex in $w$. Furthermore, there exists a point $w^* \in \arg\min_w F(w)$ such that $\|w^*\| = 1$.*

(b) *The variance of the gradient is bounded, i.e. $\mathbb{E}_{z \sim \mathcal{D}}[\|\nabla f(w; z) - \nabla F(w)\|^2] \leq 1$ for any $w \in \mathbb{R}$.*

(c) *If $\eta < 1/64n^{5/4}$, then for any $T > 0$, the point $\widehat{w}^{GD}_T$ returned by running GD algorithm with step size $\eta$ for $T$ steps satisfies*

$$\mathbb{E}[F(\widehat{w}^{GD}_T)] - \min_{w \in \mathbb{R}} F(w) = \Omega\Big(\frac{1}{n^{3/8}}\Big). \tag{29}$$

**Proof.** Before delving into the construction of the function $f$, we first define some auxiliary functions and notation. Define the kink function $h(w)$ as

$$h(w) := \begin{cases} 0 & \text{if} \quad w < 0 \\ -n^{5/8}w & \text{if} \quad 0 \leq w < \frac{1}{64n^{5/4}} \\ n^{5/8}w - \frac{2}{64n^{5/8}} & \text{if} \quad \frac{1}{64n^{5/4}} \leq w \leq \frac{3}{64n^{5/4}} \\ -n^{5/8}w + \frac{4}{64n^{5/8}} & \text{if} \quad \frac{3}{64n^{5/4}} < w \leq \frac{1}{16n^{5/4}} \\ 0 & \text{if} \quad \frac{1}{16n^{5/4}} \leq w \end{cases}, \tag{30}$$

and the corresponding gradients $\nabla h(w)$ as

$$\nabla h(w) := \begin{cases} 0 & \text{if} \quad w < 0 \\ -n^{5/8} & \text{if} \quad 0 \leq w < \frac{1}{64n^{5/4}} \\ n^{5/8} & \text{if} \quad \frac{1}{64n^{5/4}} \leq w \leq \frac{3}{64n^{5/4}} \\ -n^{5/8} & \text{if} \quad \frac{3}{64n^{5/4}} < w \leq \frac{1}{16n^{5/4}} \\ 0 & \text{if} \quad \frac{1}{16n^{5/4}} \leq w \end{cases}. \tag{31}$$

Additionally, define the set

$$H := \Big\{ \frac{1}{4} + \frac{1}{8n^{5/4}}, \frac{1}{4} + \frac{2}{8n^{5/4}}, \dots, \frac{3}{4} \Big\},$$

where the set $H$ has $4n^{5/4}$ numbers from the interval $[1/4, 3/4]$ spaced at a distance of $1/8n^{5/4}$.

We now present our learning setting:

- *Data sample:* $z$ consists of the tuple $(\beta, y)$ where $\beta \in H$ and $y \in \{-1, +1\}$.

- *Data Distribution:* $\mathcal{D}$ over the instances $z = (\beta, y)$ is defined such that

$$\beta \sim \text{Uniform}(H) \qquad \text{and} \qquad y \sim \text{Uniform}(\{-1, 1\}), \tag{32}$$

  where $\beta$ and $y$ are sampled independent of each other.

- *Loss function:* is defined as

$$f(w; z) := \frac{1}{n^{3/8}} \max\{-w, -1\} + y \cdot h(w + \beta), \tag{33}$$

  Additionally, define the population loss $F(w) := \mathbb{E}_{z \sim \mathcal{D}}[f(w; z)]$.

We next show the desired statements for this learning setting:

$(a)$ For the distribution $\mathcal{D}$ defined in (32), since $\mathbb{E}[y] = 0$ and $y$ is sampled independent of $\beta$, we have that the population loss

$$F(w) = \mathbb{E}_{z \sim \mathcal{D}}[f(w; z)] = \frac{1}{n^{3/8}} \max\{-w, -1\}.$$

Clearly, $F(w)$ is 1-Lipschitz and convex is $w$. Additionally, the point $w^* = 1$ is a minimizer of $F(w)$.

$(b)$ We next bound variance of the stochastic gradient. For any $w \in \mathbb{R}$, we have

$$\begin{aligned}
\mathbb{E}_{z \sim \mathcal{D}}[|\nabla f(w; z) - \nabla F(w)|^2] &= \mathbb{E}_{(\beta, y)}\left[|y \nabla h(w + \beta)|^2\right] \\
&= \mathbb{E}_{\beta}\left[|\nabla h(w + \beta)|^2\right] \\
&= \mathbb{E}_{\beta}\left[\mathbb{1}\left\{\beta \in \left[w - \frac{1}{16 n^{5/4}}, w\right]\right\} \cdot n^{5/4}\right] \\
&= n^{5/4} \Pr\left(\beta \in \left[w - \frac{1}{16 n^{5/4}}, w\right]\right)
\end{aligned}$$

where the equality in the first line holds because $y \in \{-1, 1\}$, and the second line follows from the construction of the function $h$ which implies that $|\nabla h(w + \beta)| \leq n^{5/8}$ for any $w$ and $\beta$ (see (31)). Using the fact that $\beta \sim \text{Uniform}(B)$ in the above, we get that

$$\mathbb{E}_{z \sim \mathcal{D}}[|\nabla f(w; z) - \nabla F(w)|^2] \leq 1/4.$$

$(c)$ We next show that GD algorithm when run with the step size of $\eta \leq 1/64 n^{5/4}$ fails to converge to a good solution.

Define $\beta_{(1)}, \ldots, \beta_{(n)}$ such that $\beta_{(j)}$ denotes the $j$th smallest item in the set $\{\beta_1, \ldots, \beta_n\}$, and define $y_{(j)}$ as the corresponding $y$ variable for the random variable $\beta_{(j)}$. An application of Lemma 9 implies that with probability at least $1 - 2/n^{1/4}$, there exists a $\widehat{j} \leq \lceil \log(n)/2 n^{1/4} \rceil$ such that:

  $(i)$ $\beta_{(1)} \neq \beta_{(2)} \neq \beta_{(3)} \neq \cdots \neq \beta_{(\widehat{j}+1)}$, and

  $(ii)$ $y_{(\widehat{j})} = +1$.

In the following, we condition on the occurrence of the above two events. The key idea of the proof is that at the level of the empirical loss, the first $\widehat{j}$ kinks (that arise as a result of the stochastic function $h$) would be isolated from each other due to event-$(i)$. Thus, the norm of the gradient of the empirical loss would be bounded by $2/n^{3/8}$ for all points before $\beta_{\widehat{j}}$. At the same time, since $y_{(\widehat{j})} = +1$ from event-$(ii)$, the empirical loss would be flat in a small interval around $\beta_{\widehat{j}}$. As we show in the following, when GD is run on the empirical loss with step size $\eta \leq 1/64 n^{5/4}$, some GD iterate will lie in this small flat region, and after that GD will fail to make progress because the gradient is 0; hence outputting a bad solution. On the other hand, GD / SGD run with a large step size e.g. $1/\sqrt{n}$ will easily jump over these kinks and converge to a good solution. We illustrate this intuition in Figure 1, and provide the proof below.

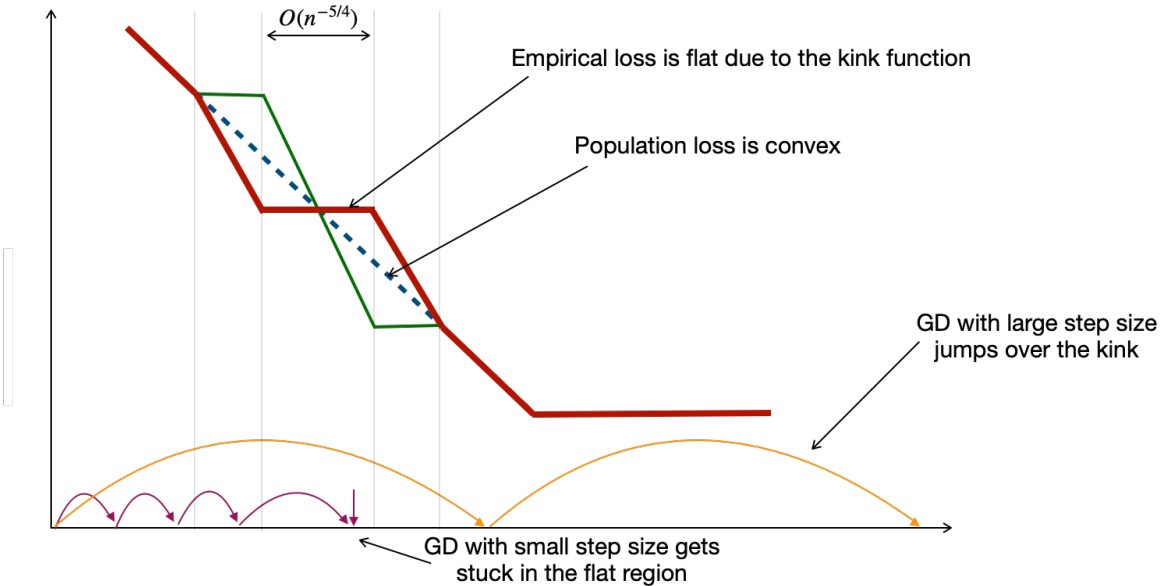

Figure 1: (Picture not drawn according to scale) The solid red line shows the empirical loss induced by the kink function when $y = +1$, the solid green line shows the empirical loss when $y = -1$, and the dotted blue line shows the convex population loss. The empirical loss when $y = +1$ has gradient 0 in a region of width $1/32n^{5/4}$. Gradient descent with step size smaller than $1/64n^{5/4}$, shown in the bottom, will get stuck in this flat region and thus fail to find a good solution. On the other hand, gradient descent with large step size will jump over the kink and find a good solution.

Recall that the empirical loss on the dataset $S$ is given by:

$$\widehat{F}(w) = \frac{1}{n^{3/8}} \max\{-w, -1\} + \frac{1}{n} \sum_{j=1}^{n} y_j(h(w + \beta_j))$$

Note that for any $\beta \in H$, the set of $w$ for which $h(w + \beta)$ is non-zeros is given by the interval $[\beta, \beta + 1/16n^{5/4}]$. Furthermore, any two numbers in the set $H$ are at least $1/8n^{5/4}$ apart from each other. Thus, the event-$(i)$ above implies that non-zero parts of the functions $\{h(w + \beta_{(1)}), \ldots, h(w + \beta_{(\widehat{j})})\}$, i.e. the first $\widehat{j}$ kinks do not overlap with each other. This implies that for $w < \beta_{(\widehat{j}+1)}$,

$$|\nabla \widehat{F}(w)| \leq \frac{1}{n^{3/8}} + \frac{1}{n} \sum_{j=1}^{n} |\nabla h(w + \beta_j)|$$

$$\leq \frac{1}{n^{3/8}} + \frac{n^{5/8}}{n} \leq \frac{2}{n^{3/8}}, \tag{34}$$

where the first inequality in the above follows from Triangle inequality and the second inequality holds because at most one of the $n$ terms in the stochastic component $\sum_{j=1}^{n} y_j(h(w + \beta_j))$ above is non-zero for $w < \beta_{(\widehat{j}+1)}$. Thus, $\widehat{F}(w)$ is 1-Lipschitz for $w \in [0, \beta_{(\widehat{j}+1)})$.

Next, the event-$(ii)$ above implies that $y_{(\widehat{j})} = 1$. Define the interval $\widetilde{\mathcal{W}} := (\beta_{(\widehat{j})} + 1/64n^{5/4}, \beta_{(\widehat{j})} + 3/64n^{5/4})$ and note that for any $w \in \widetilde{\mathcal{W}}$,

$$\nabla \widehat{F}(w) = -\frac{1}{n^{3/8}} + \frac{1}{n} \nabla h(w + \beta_{(\widehat{j})})$$

$$= -\frac{1}{n^{3/8}} + \frac{1}{n} \cdot n^{5/8} = 0, \tag{35}$$

where the first equality holds because $y_{(\widehat{j})} = +1$ and using the event-$(i)$ above. This implies that the empirical loss $\widehat{F}(w)$ has 0 gradient for $w \in \widetilde{\mathcal{W}}$, and thus GD algorithm will stop updating if any iterate reaches $\widetilde{\mathcal{W}}$.

In the following, we will show that for $\eta \leq 1/64n^{5/4}$, GD algorithm is bound to get stuck in the interval $\widetilde{\mathcal{W}}$, and will thus fail to find a good solution. Consider the dynamics of the GD algorithm:

$$w_{t+1} \leftarrow w_t - \eta \nabla \widehat{F}(w_t),$$

where the initial point $w_1 = 0$. Suppose there exists some time $\tau$ for which $w_\tau > \beta_{(\widehat{j}+1)}$, and let $t_0 > 0$ denote the smallest such time. Thus, for any $t < t_0$ and $\eta \leq 1/64n^{5/4}$, we have that

$$|w_{t+1} - w_t| = \eta |\nabla \widehat{F}(w)| \leq \eta \leq 1/64n^{5/4},$$

where the first inequality holds due to (34). This implies that any two consecutive iterates produced by the GD algorithm for $t < t_0$ are at most $1/64n^{5/4}$ apart from each other. However, note that the interval $\widetilde{\mathcal{W}} \subseteq [0, \beta_{(\widehat{j}+1)})$ and has width of $2/64n^{5/4}$. Thus, there exists some time $t' \leq t_0$ for which $w_{t'}$ will lie in the set $\widetilde{\mathcal{W}}$. However, recall that $\nabla F(w) = 0$ for any $w \in \widetilde{\mathcal{W}}$ as shown in (35). Thus, once $w_{t'} \in \widetilde{\mathcal{W}}$, GD will not update any further implying that for all $t > t'$, $w_t = w_{t'}$. This shows via contradiction that no such time $\tau$ exists for which $w_\tau > \beta_{(\widehat{j}+1)}$.

Hence for $\eta < 1/64n^{5/4}$, all iterates $\{w_t\}_{t \geq 0}$ generated by the GD algorithm will lie in the set $[0, \beta_{(\widehat{j}+1)})$. Thus, for any $T > 1$, the returned point $\widehat{w}_T^{\mathrm{GD}}$ satisfies

$$\widehat{w}_T^{\mathrm{GD}} \leq \beta_{(\widehat{j}+1)} \leq \frac{3}{4},$$

where the second inequality holds because $\beta_{(\widehat{j}+1)} \in H$ and is thus smaller than $3/4$. Thus,

$$F(\widehat{w}_T^{\mathrm{GD}}) - \min_{w \in \mathbb{R}} F(w) \geq \frac{1}{4n^{3/8}}. \tag{36}$$

Since, the events $(i)$ and $(ii)$ occur simultaneously with probability at least $1 - 2/n^{1/4}$, we have that for any $T \geq 0$,

$$\mathbb{E}[F(\widehat{w}_T^{\mathrm{GD}})] - \min_{w \in \mathbb{R}} F(w) = \Omega\left(\frac{1}{n^{3/8}}\right).$$

$\square$

**Lemma 9.** *Suppose $(\beta_1, y_1), \ldots, (\beta_n, y_n)$ be $n$ samples drawn independently from $Uniform(H) \times Uniform(\{-1, 1\})$, where the set $H := \{\frac{1}{4} + \frac{1}{8n^{5/4}}, \frac{1}{4} + \frac{2}{8n^{5/4}}, \ldots, \frac{3}{4}\}$. Further, define $\beta_{(j)}$ to denote the jth smallest number in the set $\{\beta_1, \ldots, \beta_n\}$, and define $y_{(j)}$ as the corresponding $y$ variable for $\beta_{(j)}$. Then, with probability at least $1 - 2/n^{1/4}$, there exists a $\widehat{j} \leq \lceil \log(n)/2n^{1/4} \rceil$ such that:*

*(i)* $\beta_{(1)} \neq \beta_{(2)} \neq \beta_{(3)} \neq \cdots \neq \beta_{(\widehat{j}+1)}$, *and*

*(ii)* $y_{(\widehat{j})} = +1$.

**Proof.** Let $k = \lceil \log(n)/2n^{1/4} \rceil$. We give the proof by translating our problem into a "ball and bins" problem.

Let the set $H$ denote $m = 4n^{5/4}$ distinct bins, and let there be $n$ distinct balls. Each of the $n$ balls are tossed independently into one of the $m$ bins drawn uniformly at random. For any $i, j$, the event where the ball $j$ is tossed into the bin $i$ corresponds to the event that $\beta_j = \frac{1}{4} + \frac{i}{8n^{5/4}}$. In addition to this, the balls are such that they change color after they are tossed into a bin. In particular, after being tossed into a bin, a ball changes color to either red or blue with equal probability. For any $i, j$, the event where the ball $j$ takes red color corresponds to the event that $y_j = 1$; Similarly, blue color corresponds to $y = -1$. Thus, in the balls and bins model, the position of the $n$ balls

in the bins and their corresponding colors after tossing reveals the value of the random variables $(\beta_1, y_1), \ldots, (\beta_n, y_n)$.

In the following, we first show that with probability at least $1 - \frac{2}{n^{1/4}}$, the first $k$ bins will have at most one ball each, and there will be some bin amongst the first $k$ bins that contains a red color ball. Define $E_i$ to denote the event that the bin $i$ gets more than one ball. Thus,

$$\Pr(E_i) = 1 - \Pr(\text{Bin } i \text{ has 0 or 1 ball in it})$$

$$= 1 - \left(1 - \frac{1}{n^{5/4}}\right)^n - n\left(1 - \frac{1}{n^{5/4}}\right)^{n-1}\frac{1}{n^{5/4}}$$

$$\leq 1 - \left(1 - \frac{1}{n^{1/4}}\right) - \frac{1}{n^{1/4}}\left(1 - \frac{n-1}{n^{5/4}}\right)$$

$$= \frac{n-1}{n^{3/2}} \leq \frac{1}{\sqrt{n}} \tag{37}$$

where the first inequality in the above follows from the fact that $(1 - \alpha)^n \leq 1 - \alpha n$ for any $\alpha \geq -1$ and $n \geq 1$. Let $A_k$ denote the event that there exists some bin among the first $k$ bins that has more than one ball. Taking a union bound, we get

$$\Pr(A_k) = \Pr(\cup_{i=1}^k E_i) \leq \sum_{i=1}^k \Pr(E_i) \leq \frac{k}{\sqrt{n}}.$$

Next, let $B_k$ denote the event that there is no red ball in the first $k$ bins after $n$ tosses. When we throw a ball, the probability that it will fall in the first $k$ bins is given by $k/n^{5/4}$, and the probability that it takes the color red is $1/2$. Since the bin chosen and the final color are independent of each other for each ball, the probability that a ball falls into the first $k$ bins and is of red color is given by $k/2n^{5/4}$. Furthermore, since the balls are thrown independent to each other, the probability that if we throw $n$ balls, no ball falls into the first $k$ bins that takes the red color is given by

$$\Pr(B_k) = \left(1 - \frac{k}{2n^{5/4}}\right)^n.$$

Finally, let us define $C_k$ to denote the event that there is at most one ball per bin amongst the first $k$ bins, and that there exists a bin amongst the first $k$ bins with a red ball in it. By the definition of the events $A_k$ and $B_k$, we note that

$$\Pr(C_k) = \Pr(A_k^c \cap B_k^c)$$

$$= 1 - \Pr(A_k \cup B_k)$$

$$\geq 1 - \Pr(A_k) - \Pr(B_k)$$

$$\geq 1 - \frac{k}{\sqrt{n}} - \left(1 - \frac{k}{2n^{5/4}}\right)^n$$

$$\geq 1 - \frac{k}{\sqrt{n}} - e^{-k/2n^{1/4}},$$

where the first inequality in the above follows from the union bound, the second inequality holds by plugging in the corresponding bounds for $P(A_k)$ and $P(B_k)$, and the last line is due to the fact that $(1 + \alpha)^n \leq e^{\alpha n}$ for any $\alpha$. Plugging in the value of $k = \lceil \log(n)/2n^{1/4} \rceil$ in the above, we get that

$$\Pr(C_k) \geq 1 - \frac{2}{n^{1/4}}.$$

Thus, with probability at least $1 - 2/n^{1/4}$, the first $k$ bins have at most one ball each, and there exists some bin $\widehat{j} \in [k]$ that contains a red ball; when this happens, the corresponding random variables $(\beta_1, y_1), \ldots, (\beta_n, y_n)$ satisfy:

(a) $\beta_{(1)} \neq \beta_{(2)} \neq \beta_{(3)} \neq \cdots \neq \beta_{(\widehat{j}+1)}$, and

(b) $y_{\widehat{j}} = +1$.

$\square$

## C.4 Proof of Theorem 3

In this section, we combine the lower bound constructions in Theorem 11 and Lemma 8 to provide an instance of a SCO problem on which, for any step size $\eta$ and time steps $T$, GD has the lower bound of $\Omega(1/n^{5/12})$.

**Proof of Theorem 3 .** Throughout the proof, we assume that $n \geq 200$ and the initial point $w_1 = 0$.

We first define some additional notation: Let $w_{(1)}$, $z_{(1)}$, $f_{(1)}$ and $\mathcal{D}_{(1)}$ denote the optimization variable, the data sample, the instance loss and the distribution over $z_{(1)}$ corresponding to the lower bound construction in Theorem 11. The statement of Theorem 11 implies that:

(a) $f_{(1)}(w_{(1)}; z_{(1)})$ is 3-Lipschitz in the variable $w_{(1)}$ for any $z_{(1)}$.

(b) The population loss $F_{(1)}(w_{(1)}) := \mathbb{E}_{z_{(1)} \sim \mathcal{D}_{(1)}}[f(w_{(1)}; z_{(1)})]$ is 3-Lipschitz and convex in $w_{(1)}$. Additionally, there exists a point $w_{(1)}^* \in \mathrm{argmin}_{w_{(1)}} F_{(1)}(w_{(1)})$ such that $\|w_{(1)}^*\| = O(1)$.

(c) For any $\eta \in [1/n^2, 1)$ and $T \geq 1$, the point $\widehat{w}_{(1)}^{\mathrm{GD}}[\eta, T]$ returned by running GD algorithm with a step size of $\eta$ for $T$ time steps satisfies:

$$\mathbb{E}\big[F_{(1)}\big(\widehat{w}_{(1)}^{\mathrm{GD}}[\eta, T]\big)\big] - \min_{w_{(1)}} F_{(1)}(w_{(1)}) = \Omega\Big(\frac{1}{\log^4(n)} \min\Big\{\eta\sqrt{T} + \frac{1}{\eta T} + \frac{\eta T}{n}, 1\Big\}\Big). \quad (38)$$

Similarly, let $w_{(2)}$, $z_{(2)}$, $f_{(2)}$ and $\mathcal{D}_{(2)}$ denote the optimization variable, the data sample, the instance loss and the distribution over $z_{(2)}$ corresponding to the lower bound construction in Lemma 8. The statement of Lemma 8 implies that:

(a) The population loss $F_{(2)}(w_{(2)}) := \mathbb{E}_{z_{(2)} \sim \mathcal{D}_{(2)}}\big[f(w_{(2)}; z_{(2)})\big]$ is 1-Lipschitz and convex in $w_{(2)}$. Additionally, there exists a point $w_{(2)}^* \in \mathrm{argmin}_{w_{(2)}} F(w_{(2)})$ such that $\|w_{(2)}^*\| \leq 1$.

(b) The variance of the gradient is bounded, i.e. for any $w_{(2)}$,

$$\mathbb{E}_{z_{(2)} \sim \mathcal{D}_{(2)}}\big[\|\nabla f_{(2)}(w_{(2)}; z_{(2)}) - \nabla F_{(2)}(w_{(2)})\|^2\big] \leq 1. \quad (39)$$

(c) If $\eta < 1/64n^{5/4}$, then for any $T > 0$, the point $\widehat{w}_{(2)}^{\mathrm{GD}}[\eta, T]$ returned by running GD algorithm with step size $\eta$ for $T$ steps satisfies

$$\mathbb{E}[F_{(2)}\big(\widehat{w}_{(2)}^{\mathrm{GD}}[\eta, T]\big)] - \min_{w_{(2)}} F_{(2)}(w_{(2)}) = \Omega\Big(\frac{1}{n^{3/8}}\Big). \quad (40)$$

**Lower bound construction.** Consider the following learning setting:

- *Optimization variable*: $w$ is defined as the concatenation of the variables $(w_{(1)}, w_{(2)})$.

- *Data instance*: $z$ is defined as the concatenation of the data instances $(z_{(1)}, z_{(2)})$.

- *Data distribution*: $\mathcal{D}$ is defined as $\mathcal{D}_{(1)} \times \mathcal{D}_{(2)}$, i.e. $z_{(1)}$ and $z_{(2)}$ are sampled independently from $\mathcal{D}_{(1)}$ and $\mathcal{D}_{(2)}$ respectively.

- *Loss function*: is defined as

$$f(w; z) := f_{(1)}(w_{(1)}; z_{(1)}) + f_{(2)}(w_{(2)}; z_{(2)}),$$

Additionally, define the population loss $F(w) := \mathbb{E}_{z \sim \mathcal{D}}[f(w; z)] = F_{(1)}(w_{(1)}) + F_{(2)}(w_{(2)})$.

**Excess risk guarantee for SGD.** We first show that the above learning setting is in SCO. Note that the population loss

$$F(w) = F_{(1)}(w_{(1)}) + F_{(2)}(w_{(2)}).$$

Clearly, $F(w)$ is convex in $w$ since the functions $F_{(1)}(w_{(1)})$ and $F_{(2)}(w_{(2)})$ are convex in $w_{(1)}$ and $w_{(2)}$ respectively. Next, note that for any $w$ and $w'$,

$$
\begin{aligned}
|F(w) - F(w')| &\overset{(i)}{\leq} |F_{(1)}(w_{(1)}) - F_{(1)}(w'_{(1)})| + |F_{(2)}(w_{(2)}) - F_{(2)}(w'_{(2)})| \\
&\overset{(ii)}{\leq} 3\|w_{(1)} - w'_{(1)}\| + \|w_{(2)} - w'_{(2)}\| \\
&\leq 3\left(\sqrt{\|w_{(1)} - w'_{(1)}\|^2} + \sqrt{\|w_{(2)} - w'_{(2)}\|^2}\right) \\
&\overset{(iii)}{\leq} 3\sqrt{2} \cdot \sqrt{\|w_{(1)} - w'_{(1)}\|^2 + \|w_{(2)} - w'_{(2)}\|^2} \\
&\overset{(iv)}{=} 3\sqrt{2} \cdot \sqrt{\|w - w'\|^2} \\
&= 3\sqrt{2}\|w - w'\|,
\end{aligned}
$$

where the inequality $(i)$ above follows from Triangle inequality and the inequality $(ii)$ is given by the fact that $F_{(1)}$ is 3-Lipschitz in $w_{(1)}$ and that $F_{(2)}$ is 1-Lipschitz in $w_{(2)}$. The inequality in $(iii)$ follows from an application of Jensen's inequality and using concavity of square-root. Finally, the equality in $(iv)$ is by construction of the variable $w$ as concatenation of the variables $(w_{(1)}, w_{(2)})$. Thus, the population loss $F(w)$ is $3\sqrt{2}$-Lipschitz in $w$.

We next show a bound on the variance of the gradient. Note that for any $w$,

$$
\begin{aligned}
\mathbb{E}_{z \sim \mathcal{D}}&\left[\|\nabla f(w; z) - \nabla F(w)\|^2\right] \\
&= \mathbb{E}_{z \sim \mathcal{D}}\left[\|\nabla f_{(1)}(w_{(1)}; z_{(1)}) + \nabla f_{(2)}(w_{(2)}; z_{(2)}) - \nabla F_{(1)}(w_{(1)}) - \nabla F_{(2)}(w_{(2)})\|^2\right] \\
&\overset{(i)}{\leq} 2\,\mathbb{E}_{z_{(1)} \sim \mathcal{D}_{(1)}}\left[\|\nabla f_{(1)}(w_{(1)}; z_{(1)}) - \nabla F_{(1)}(w_{(1)})\|^2\right] \\
&\qquad + 2\,\mathbb{E}_{z_{(2)} \sim \mathcal{D}_{(2)}}\left[\|\nabla f_{(2)}(w_{(2)}; z_{(2)}) - \nabla F_{(2)}(w_{(2)})\|^2\right] \\
&\overset{(ii)}{\leq} 72 + 2\,\mathbb{E}_{z_{(2)} \sim \mathcal{D}_{(2)}}\left[\|\nabla f_{(2)}(w_{(2)}; z_{(2)}) - \nabla F_{(2)}(w_{(2)})\|^2\right] \overset{(iii)}{\leq} 74,
\end{aligned}
$$

where the inequality $(i)$ above follows from the fact that $(a + b)^2 \leq 2a^2 + 2b^2$ for any $a, b$. The inequality $(ii)$ holds because the function $f_{(1)}(w_{(1)}; z_{(1)})$ is 3-Lipchitz in $w_{(1)}$ for any $z_{(1)}$, and because $F_{(1)}(w_{(1)})$ is 3-lipschitz in $w_{(1)}$. Finally, the inequality $(iii)$ is due to the bound in (39).

We next show that there exists a $w^* \in \operatorname{argmin}_w F(w)$ such that $\|w^*\| = O(1)$. Since the components $(w_{(1)}, w_{(2)})$ do not interact with each other in the above construction of the loss function $f$, we note that the point $w^* = (w^*_{(1)}, w^*_{(2)})$ is a minimizer of $F(w) = F_{(1)}(w_{(1)}) + F_{(1)}(w_{(1)})$. This point $w^*$ satisfies

$$
\|w^*\| \leq \|w^*_{(1)}\| + \|w^*_{(2)}\| = O(1),
$$

where we used the fact that $\|w^*_{(1)}\| = O(1)$ and $\|w^*_{(2)}\| = O(1)$.

Combining the above derived properties, we get that:

(a) The population loss $F(w)$ is $3\sqrt{2}$-Lipschitz and convex in $w$.

(b) There exists a point $w^* \in \operatorname{argmin}_w F(w)$ such that $\|w^*\| = O(1)$.

(c) For any $w$, the gradient variance $\mathbb{E}_{z \sim \mathcal{D}}\left[\|\nabla f(w; z) - \nabla F(w)\|^2\right] \leq 72$.

Thus, as a consequence of Theorem 8, we get that running SGD with step size $\eta = 1/\sqrt{n}$ and initialialization $w_1 = 0$, returns the point $\widehat{w}_n^{\mathrm{SGD}}$ that satisfies

$$
\mathbb{E}\left[F(\widehat{w}_n^{\mathrm{SGD}})\right] - \min_w F(w) = O\left(\frac{1}{\sqrt{n}}\right). \tag{41}
$$

**Lower bound for GD.**   We next show that GD algorithm fails to match the performance guarantee of SGD in (41), for any step size $\eta$ and time step $T$.

Let $\widehat{w}^{\mathrm{GD}}[\eta, T]$ denote the point returned by running GD algorithm on the function $f$ with step size $\eta$ for $T$ steps. Since the components $w_{(1)}$ and $w_{(2)}$ do not interact with each other in the GD update step due to the construction of the function $f$, we have that

$$\mathbb{E}\big[F(\widehat{w}^{\mathrm{GD}}[\eta, T])\big] - \min_{w} F(w) = \mathbb{E}[F_{(1)}(\widehat{w}_{(1)}^{\mathrm{GD}}[\eta, T])] - \min_{w_{(1)}} F_{(1)}(w_{(1)})$$
$$+ \mathbb{E}[F_{(2)}(\widehat{w}_{(2)}^{\mathrm{GD}}[\eta, T])] - \min_{w_{(2)}} F_{(2)}(w_{(2)}) \quad (42)$$

The key idea behind the lower bound for GD is that first components has a lower bound of $\Omega(1/n^{5/12})$ when $\eta$ is larger than $1/64n^{5/4}$. Specifically, in order to improve the excess risk bound over the rate of $1/n^{5/12}$ w.r.t. the variable $w_{(1)}$, we need $\eta$ to be smaller than $1/64n^{5/4}$. However, any choice of $\eta < 1/64n^{5/4}$ fails to find a good solution w.r.t. the component $w_{(2)}$. We formalize this intuition by considering the two cases (a) $\eta < 1/64n^{5/4}$, and (b) $\eta \geq 1/64n^{5/4}$ separately below.

- **Case 1: $\eta < 1/64n^{5/4}$.** Using the fact that $\mathbb{E}[F_{(1)}(\widehat{w}_{(1)}^{\mathrm{GD}}[\eta, T])] - \min_{w_{(1)}} F_{(1)}(w_{(1)}) \geq 0$ in (42), we get that

$$\mathbb{E}\big[F(\widehat{w}^{\mathrm{GD}}[\eta, T])\big] - \min_{w} F(w) \geq \mathbb{E}\big[F_{(2)}\big(\widehat{w}_{(2)}^{\mathrm{GD}}[\eta, T]\big)\big] - \min_{w_{(2)}} F_{(2)}(w_{(2)})$$
$$= \Omega\Big(\frac{1}{n^{3/8}}\Big), \quad (43)$$

 where the inequality in the second line above is due to the lower bound in (40) which holds for all $\eta < 1/64n^{5/4}$ and $T \geq 1$.

- **Case 2: $\eta \geq 1/64n^{5/4}$.** Using the fact that $\mathbb{E}[F_{(2)}(\widehat{w}_{(2)}^{\mathrm{GD}}[\eta, T])] - \min_{w_{(2)}} F_{(2)}(w_{(2)}) \geq 0$ in (42), we get that

$$\mathbb{E}\big[F(\widehat{w}_{\eta, T}^{\mathrm{GD}})\big] - \min_{w} F(w) \geq \mathbb{E}\big[F_{(1)}\big(\widehat{w}_{(1)}^{\mathrm{GD}}[\eta, T]\big)\big] - \min_{w_{(1)}} F_{(1)}(w_{(1)}). \quad (44)$$

The lower bound in (38) suggests that for $\eta \in [1/n^2, 1)$ and $T \geq 1$,

$$\mathbb{E}\big[F_{(1)}\big(\widehat{w}_{(1)}^{\mathrm{GD}}[\eta, T]\big)\big] - \min_{w_{(1)}} F_{(1)}(w_{(1)}) = \Omega\Big(\frac{1}{\log^4(n)} \min\Big\{\eta\sqrt{T} + \frac{1}{\eta T} + \frac{\eta T}{n}, 1\Big\}\Big)$$
$$= \Omega\Big(\frac{1}{\log^4(n)} \min\Big\{\eta\sqrt{T} + \frac{1}{2\eta T} + \frac{1}{2\eta T} + \frac{\eta T}{n}, 1\Big\}\Big)$$
$$\overset{(i)}{=} \Omega\Big(\frac{1}{\log^4(n)} \min\Big\{\eta\sqrt{T} + \frac{1}{2\eta T} + \frac{1}{\sqrt{2n}}, 1\Big\}\Big)$$
$$\quad (45)$$
$$\overset{(ii)}{=} \Omega\Big(\frac{1}{\log^4(n)} \min\Big\{\eta^{1/3} + \frac{1}{\sqrt{n}}, 1\Big\}\Big)$$

where $(i)$ follows from an application of the AM-GM inequality for the last two terms, and $(ii)$ holds by setting $T = 1/\eta^{4/3}$ which minimizes the expression in (45). Plugging the above lower bound in (44), we get that

$$\mathbb{E}\big[F\big(\widehat{w}^{\mathrm{GD}}[\eta, T]\big)\big] - \min_{w} F(w) \geq \Omega\Big(\frac{1}{\log^4(n)} \min\Big\{\eta^{1/3} + \frac{1}{\sqrt{n}}, 1\Big\}\Big).$$

Finally, using the fact that $\eta \geq 1/64n^{5/4}$ in the above bound, we get

$$\mathbb{E}\big[F\big(\widehat{w}^{\mathrm{GD}}[\eta, T]\big)\big] - \min_{w} F(w) \geq \Omega\Big(\frac{1}{\log^4(n)} \min\Big\{\frac{1}{n^{5/12}} + \frac{1}{\sqrt{n}}, 1\Big\}\Big). \quad (46)$$

Combining the lower bound from (43) and (46) for the two cases above, we get that for all $\eta \geq 0$ and $T \geq 1$, the point $\widehat{w}^{\mathrm{GD}}[\eta, T]$ returned by running GD algorithm on the function $f$ with step size $\eta$ for $T$ steps satisfies:

$$\mathbb{E}\big[F\big(\widehat{w}^{\mathrm{GD}}[\eta, T]\big)\big] - \min_w F(w) \geq \Omega\Big(\frac{1}{\log^4(n)} \min\Big\{\frac{1}{n^{5/12}}, 1\Big\}\Big).$$

$\square$

# D  Missing proofs from Section 5

The pseudocode for multi-pass SGD algorithm given in Algorithm 1 is slightly different from the description of the algorithm given at the beginning of Section 5. In particular, at the start of every epoch, Algorithm 1 uses the following projection operation:

$$\Pi_{w_1, B}(w) = \begin{cases} w & \text{if } \|w - w_1\| \leq B \\ w_1 + \frac{B}{\|w - w_1\|}(w - w_1) & \text{otherwise.} \end{cases}$$

This ensures that the iterate at the start of every epoch has bounded norm. Rest of the algorithm is the same as in the description in the main body.

---

**Algorithm 1** Multi-pass SGD algorithm

---

**Input:** Dataset $S = \{z_i\}_{i=1}^n$, number of passes $k$, initial point $w_1$.
1: Define $m := n/2$, $S_1 := \{z_i\}_{i=1}^m$ and $S_2 := S \setminus S_1$
2: Initialize $\widehat{\mathcal{W}} \leftarrow \emptyset$
3: **for** $j = 1, \ldots, k$ **do**                                                  ▷ Multiple passes
4:     $w_{m(j-1)+1} \leftarrow \Pi_{w_1, B}(w_{m(j-1)+1})$
5:     $\eta_j \leftarrow \frac{1}{\sqrt{nj}}$
6:     **for** $i = 1, 3, \ldots, m$ **do**
7:         $w_{m(j-1)+i+1} \leftarrow w_{m(j-1)+i} - \eta_j \nabla f(w_{m(j-1)+i}; z_i)$.
8:     $\widehat{w}_j \leftarrow \frac{1}{mj} \sum_{t=1}^{mj} w_j$
9:     $\widehat{\mathcal{W}} \leftarrow \widehat{\mathcal{W}} \cup \{\widehat{w}_j\}$
10: Return the point $\widehat{w}^{\mathrm{MP}} \in \operatorname{argmin}_{w \in \widehat{\mathcal{W}}} F_{S_2}(W) := \frac{1}{m} \sum_{i=1}^m f(w; z_{m+1})$      ▷ Validation

---

## D.1  Proof of Proposition 1

**Proof.** The following proof uses the fact that the loss function is bounded over the domain of interest. This is automatically ensured for our multi-pass SGD algorithm due to the projection step in Algorithm 1. Assume that $f$ is bounded by $M$. Since $S_2$ is independent of $S_1$, we note $S_2$ is also independent of the set of points $\widehat{\mathcal{W}}$. Hoeffding's inequality (Lemma 1) thus implies that, with probability at least $1 - \delta$, for all $w \in \widehat{\mathcal{W}}$,

$$|F_{S_2}(w) - F(w)| \leq M\sqrt{\frac{\log(2k/\delta)}{n}}.$$

Thus, the returned point $\widehat{w}^{\mathrm{MP}} \in \operatorname{argmin}_{w \in \widehat{\mathcal{W}}} F_{S_2}(w)$ satisfies:

$$F(\widehat{w}^{\mathrm{MP}}) \leq \min_{w \in \widehat{\mathcal{W}}} F(w) + 2M\sqrt{\frac{\log(2k/\delta)}{n}}$$

$$\leq F(\widehat{w}_1) + 2M\sqrt{\frac{\log(2k/\delta)}{n}}.$$

Observing that the point $\widehat{w}_1$ denotes $\widehat{w}_{n/2}^{\mathrm{SGD}}$, and converting the above high probability statement into an in-expectation result (since $f$ is bounded by $M$) gives us the desired statement. $\square$

### D.2 Proof of Theorem 4

For the upper bound of part (b) in Theorem 4, we need the following slight generalization of Theorem 1.

**Lemma 10.** *Consider any SCO problem and initial point $w_1$ satisfying Assumption I in (2) and Assumption II in (3). Suppose starting from the point $w_1$, we run SGD algorithm with step size $\eta$ for $n$ steps. Then the average iterate $\widehat{w}_n^{\mathrm{SGD}} := \frac{1}{n} \sum_{i=1}^{n} w_i$ enjoys the bound*

$$\mathbb{E}_S[F(\widehat{w}_n^{\mathrm{SGD}})] - \inf_w F(w) \leq \eta(\sigma^2 + L^2) + \frac{1}{2\eta n} \|w_1 - w^*\|^2,$$

*for any point $w^* \in \operatorname{argmin}_w F(w)$.*

**Proof.** The proof follows exactly along the lines of the proof of Theorem 8 given on page 14. $\qquad\square$

**Theorem 12.** *Consider any SCO problem satisfying Assumption I in (2) and Assumption II in (3). Suppose we run the following variant of SGD: for some integer $k \geq 1$, we run Algorithm 1 for $nk$ steps with the only change being that fresh samples $z_t$ are used in each update step (line 5) instead of reusing samples. Then the average iterate $\widehat{w}_{nk}^{\mathrm{SGD}}$ enjoys the bound*

$$\mathbb{E}_S[F(\widehat{w}_{nk}^{\mathrm{SGD}})] - \inf_{w:\, \|w - w_1\| \leq B} F(w) \leq 2(B^2 + L^2 + \sigma^2)\sqrt{nk}.$$

**Proof.** The proof is completely standard, we just include it here for completeness. Let $w^* \in \mathbb{R}^d$ be any point such that $\|w_1 - w^*\| \leq B$. We use $\|w_t - w^*\|^2$ as a potential function. Within epoch $j$, we have

$$\|w_{t+1} - w^*\|^2 = \|w_t - w^*\|^2 - 2\eta_j \langle \nabla f(w_t; z_t), (w_t - w^*) \rangle + \eta_j^2 \|\nabla f(w_t; z_t)\|^2.$$

Since $\|\nabla f(w_t; z_t)\|^2 \leq 2L^2 + 2\sigma^2$, by rearranging the above, taking expectation over $z_t$ conditioned on $w_t$, and using convexity of $F$ we have

$$F(w_t) - F(w^*) \leq \frac{\|w_t - w^*\|^2 - \mathbb{E}[\|w_{t+1} - w^*\|^2 | w_t]}{2\eta_j} + \eta_j(L^2 + \sigma^2).$$

Now taking expectation over the randomness in $w_t$, and summing up the inequality for all iterations in epoch $j$, we get

$$\sum_{t=n(j-1)+1}^{nj} F(w_t) - F(w^*) \leq \frac{\mathbb{E}[\|w_{n(j-1)+1} - w^*\|^2] - \mathbb{E}[\|w'_{nj+1} - w^*\|^2]}{2\eta_j} + \eta_j(L^2 + \sigma^2)n,$$

where we use the notation $w'_{nj+1}$ to denote the iterate computed by SGD before the $\Pi_{w_1, B}$ projection to generate $w_{nj+1}$. Summing over all the $k$ epochs, we have

$$\sum_{t=1}^{nk} F(w_t) - F(w^*) \leq \frac{\|w_1 - w^*\|^2}{2\eta_1} + \sum_{j=1}^{k-1}\left(\frac{\mathbb{E}[\|w_{nj+1} - w^*\|^2]}{2\eta_{j+1}} - \frac{\mathbb{E}[\|w'_{nj+1} - w^*\|^2]}{2\eta_j}\right)$$

$$- \frac{\mathbb{E}[\|w_{nk+1} - w^*\|^2]}{2\eta_k} + \sum_{j=1}^{k} \eta_j(L^2 + \sigma^2)n$$

$$\overset{(i)}{\leq} \frac{\|w_1 - w^*\|^2}{2\eta_1} + \sum_{j=1}^{k-1}\left(\frac{1}{2\eta_{j+1}} - \frac{1}{2\eta_j}\right)\mathbb{E}[\|w_{nj+1} - w^*\|^2] + \sum_{j=1}^{k} \eta_j(L^2 + \sigma^2)n$$

$$\overset{(ii)}{\leq} \frac{2B^2}{\eta_k} + \sum_{j=1}^{k} \eta_j(L^2 + \sigma^2)n$$

$$\overset{(iii)}{\leq} 2(B^2 + L^2 + \sigma^2)\sqrt{nk}.$$

Here, $(i)$ follows since $\|w_{nj+1} - w^*\|^2 \leq \|w'_{nj+1} - w^*\|^2$ since the $\Pi_{w_1, B}$ projection can only reduce the distance to $w^*$, and $(ii)$ follows since $\|w_{nj+1} - w^*\|^2 \leq 4B^2$ for all $j = 0, 1, \ldots, k-1$, and telescoping, and finally $(iii)$ follows by plugging in $\eta_j = 1/\sqrt{nj}$.

Finally, the stated bound on $F(\widehat{w}_{nk}^{\mathrm{SGD}})$ follows via an application of Jensen's inequality to the convex function $F$. The dependence on problem specific constants ($\sigma$, $L$ and $B$) in the above bound can be improved further with a different choice of the step size $\eta$; getting the optimal dependence on these constants, however, is not the focus of this work. $\qquad\square$

We now prove Theorem 4. To begin, we first define the instance space, the loss function, and the data distribution.

**Instance space and loss function.** We define the instance space $\mathcal{Z} = \{0,1\}^{kd} \times \{\pm 1\}^k \times \{0, e_1, \ldots, e_d\}^k$. That is, each instance $z \in \mathcal{Z}$ can be written as a 3-tuple $z = (x, y, \alpha)$ where $x \in \{0,1\}^{kd}$, $y \in \{\pm 1\}^k$, and $\alpha \in \{0, e_1, \ldots, e_d\}^k$. For each $s \in [k]$, we define $x_s, y_s, \alpha_s$ to be the $s$-th parts of $x, y, \alpha$ respectively when these vectors are split into $k$ contiguous equal sized blocks of sizes $d, 1, 1$ respectively. Define the function $f_{(C)} : \mathbb{R}^{d+n+2} \times \mathcal{Z} \to \mathbb{R}$ on the variables $u \in \mathbb{R}, v \in \mathbb{R}^{n+1}, \tau \in \mathbb{R}^d$ and instance $z = (x, y, \alpha) \in \mathcal{Z}$ as follows. First, define the intervals $I_1 = (-\infty, \frac{1}{k}], I_s = (\frac{(s-1)}{k}, \frac{s}{k}]$ for $s = 2, 3, \ldots, k-1$, and $I_k = (\frac{(k-1)}{k}, \infty)$. Then define

$$f_{(C)}((u, v, \tau); z) := f_N(v) + \sum_{s=1}^{k} \mathbb{1}[u \in I_s] f_{(A)}(\tau; (x_s, y_s, \alpha_s)) - \tfrac{2}{\sqrt{kn}} \min\{u, 1\} + c_1, \qquad \text{(C)}$$

where the function $f_N : \mathbb{R}^{n+1} \to \mathbb{R}$ is defined as

$$f_N(v) := \left(\tfrac{\sqrt{n+1}}{1+\sqrt{n+1}}\right) \max_{i \in [n+1]} v_i + \left(\tfrac{1}{2+2\sqrt{n+1}}\right) \|v\|^2,$$

and the constant $c_1 = \frac{2}{\sqrt{kn}} + \frac{1}{2+2\sqrt{n+1}}$. Finally, we define the variable $w$ to denote the tuple $(u, v, \tau)$. We also assume that $n \geq 300$ and $d \geq \log(10)2^n + 1$.

For the purpose of defining the SGD update on the function $f_{(C)}(\cdot; z)$ for any given $z \in \mathcal{Z}$, we make the following convention for defining subgradients: when $u \in I_s$ for some $s \in [k]$, we use a subgradient of

$$f_N(v) + f_{(A)}(\tau; (x_s, y_s, \alpha_s)) - \tfrac{2}{\sqrt{kn}} \min\{u, 1\}.$$

It is easy to check that with the above convention, the norm of the subgradient is always $O(1)$.

**Input distribution.** The samples $z \in \mathcal{Z}$ are drawn by sampling $\langle (x_s, y_s, \alpha_s) \rangle_{s=1}^k \sim \mathcal{D}(\frac{1}{10}, \frac{1}{2}, e_{j^*})^{\otimes k}$, for some $j^* \in [d]$ that will be defined in the proof (see Definition 2 for the definition of $\mathcal{D}$).

**Proof of Theorem 4.** Let $\mathcal{D}'$ denote the distribution specified over the instance space $\mathcal{Z}$. The exact choice of $j^*$ will be specified later in this proof. Note that due to the indicators in (C), the population loss under $f_{(C)}$ when the data are sampled from $\mathcal{D}'$ can be written as

$$F(w) = \mathbb{E}_{z \sim \mathcal{D}'}[f_{(C)}((u, v, \tau); z')] = f_N(v) + \mathbb{E}_{z \sim \mathcal{D}'}[f_{(A)}(\tau; z)] - \tfrac{2}{\sqrt{kn}} \min\{u, 1\}. \qquad (47)$$

Note that by Lemma 5, $\mathbb{E}_{z \sim \mathcal{D}'}[f_{(A)}(\tau; z)]$ is a convex function of $\tau$. Hence, $F$ is convex, 1-Lipschitz and $\tau = e_{j^*}$ denotes its unique minimizer. Also note that $F$ nicely separates out as convex functions of $u, v, \tau$ and hence it is minimized by minimizing the component functions separately. In particular, it is easy to check by computing the subgradient that the optima are

$$u = 1, \qquad v = -\tfrac{1}{\sqrt{n+1}} \mathbf{1}, \qquad w = e_{j^*},$$

where $\mathbf{1}$ is the all 1's vector. The corresponding optimal values are $-\frac{2}{\sqrt{kn}}$, $-\frac{1}{2+2\sqrt{n+1}}$, and 0. This implies that $F^* = \min_w F(w) = 0$. More importantly, the suboptimality gap also decomposes as the sum of suboptimality gaps for the three functions, a fact we will use repeatedly in our analysis.

Finally, we will assume that the initial value of the variables $u, v, \tau$ is 0.

**Proof of part-(a).** Since $\|\nabla f(u, v, \tau; z)\| \leq O(1)$ for any $z \in \mathcal{Z}$ and $u, v, \tau$, and the optimal values of $u, v, \tau$ are $O(1)$ in magnitude, and $F(\omega)$ is convex in $\omega = (u, v, \tau)$, Theorem 1 implies that single pass SGD run for $n$ steps with a step size of $\frac{1}{\sqrt{n}}$ will obtain $O(\frac{1}{\sqrt{n}})$ suboptimality gap in expectation.

As for the lower bound, note that the function $f_N$ is exactly the same one constructed in [Nesterov, 2014, Section 3.2.1]. There, it is shown (in Theorem 3.2.1) that a certain class of gradient based methods (including GD/SGD) have suboptimality at least $\Omega(1/\sqrt{n})$ after $n$ iterations, which completes the proof of part-(a) of Theorem 4.

**Proof of part-(b).** In the following, we give convergence guarantee for multi-pass SGD algorithm that does $k$ passes over the dataset $S_1$ of $n/2$ samples (see Algorithm 1). First, note that the deterministic components of the function $f_{(C)}$, viz. $f_N$ and $-\frac{2}{\sqrt{kn}} \min\{u, 1\}$, are unaffected by the randomized component in the iterates produced by SGD update. Since these deterministic components are 1-Lipschitz, their optimal values are $O(1)$ in magnitude and the corresponding population losses are convex, we conclude via Theorem 12 that the multi-pass SGD, which is equivalent to GD, attains a suboptimality gap of $O(\frac{1}{\sqrt{kn}})$ on these components. Specifically, the points $\widehat{u}_k$ and $\widehat{v}_k$ returned after the $k$-th pass satisfy

$$f_N(\widehat{v}_k) - \frac{2}{\sqrt{kn}} \min\{\widehat{u}_k, 1\} - \min_{v, u}\Big(f_N(v) - \frac{2}{\sqrt{kn}} \min\{u, 1\}\Big) \leq O\Big(\frac{1}{\sqrt{nk}}\Big).$$

Coming to the randomized component of $f_{(C)}$, we note that as long as $u < 1$, the gradient of $f_{(C)}$ with respect to $u$ is always $-2/\sqrt{kn}$. Thus, $u$ keeps monotonically increasing at each step of SGD with an increment equal to the step size times $2/\sqrt{kn}$. Suppose we run $k$ pass SGD with step size set to $1/\sqrt{kn}$ and $u$ starting at 0, where in each pass we take $n/2$ steps of SGD using the dataset $S_1$. It is easy to see that for all $s \in [k]$, the value of $u$ stays in $I_s$ in the $s$-th pass, and traverses to the next interval $I_{s+1}$ as soon as the $(s+1)$-th pass starts. Thus, within each pass $s \in [k]$ and for all $i \in [n]$, multi-pass SGD encounters a fresh i.i.d. sample $(x_{i,s}, y_{i,s}, \alpha_{i,s}) \sim \mathcal{D}$ for every update. This is thus equivalent to running SGD with $kn/2$ such i.i.d. samples over the first $kn/2$ iterations. An application of Theorem 12 (where we set $n$ to be $n/2$) implies that the suboptimality gap of the iterate $\widehat{\tau}_k$ generated after the $k$th-pass of SGD algorithm on the randomized component of $f_{(C)}$ is

$$\mathbb{E}_{z \sim \mathcal{D}}[f_{(A)}(\widehat{\tau}_k; z)] - \min_w \mathbb{E}_{z \sim \mathcal{D}}[f_{(A)}(w; z)] \leq O\Big(\frac{1}{\sqrt{nk}}\Big).$$

Taking the above two bounds together, we get that the point $\widehat{w}_k = (\widehat{u}_k, \widehat{v}_k, \widehat{\tau}_k)$ satisfies

$$F(\widehat{w}_k) = F(\widehat{w}_k) - \min_w F(w) \leq O\Big(\frac{1}{\sqrt{nk}}\Big), \tag{48}$$

where the equality in the first line above follows from using the fact that $\min_w F(w) = 0$ by construction.

Finally, note that the returned point $\widehat{w}^{\mathrm{MP}} \in \operatorname{argmin}_{w \in \widehat{\mathcal{W}}} F_{S_2}(W)$ where $\mathcal{W} = \{\widehat{w}_1, \ldots, \widehat{w}_k\}$. Lemma 4 thus implies that the point $\widehat{w}^{\mathrm{MP}}$ satisfies

$$F(\widehat{w}^{\mathrm{MP}}) \leq \min_{j \in [k]} F(\widehat{w}_j) + O\Big(\frac{L \log(k/\delta)}{n} + \sqrt{\frac{\min_{j \in [k]} F(\widehat{w}_j) \, L \log(k/\delta)}{n}}\Big)$$

$$\leq F(\widehat{w}_k) + O\Big(\frac{L \log(k/\delta)}{n} + \sqrt{\frac{F(\widehat{w}_k) \, L \log(k/\delta)}{n}}\Big)$$

$$\leq \frac{1}{\sqrt{nk}} + O\Big(\frac{\log(k/\delta)}{n} + \frac{1}{n}\sqrt{\frac{\log(k/\delta)}{k}}\Big),$$

where $L$-denotes the Lipschitz constant for the function $f$ and the inequality in the last line follows by plugging in the bound on $F(\widehat{w}_k)$ from (48) and using the fact that $L = O(1)$. Finally, observing that $\min_w F(w) = 0$, the above bound implies that

$$F(\widehat{w}^{\mathrm{MP}}) - \min_w F(w) \leq O\Big(\frac{1}{\sqrt{nk}}\Big)$$

for $k = o(n)$; proving the desired claim.

**Proof of part-(c).** The proof follows exactly along the lines of the proof of Theorem 2 in Appendix B.2. Recall that $n \geq 300$ and $d \geq \log(10)2^n + 1$.

Assume, for the sake of contradiction, that there exists a regularizer $R : \mathbb{R} \times \mathbb{R}^{n+1} \times \mathbb{R}^d \to \mathbb{R}$ such that for any distribution $\mathcal{D} = \mathcal{D}(\frac{1}{10}, \frac{1}{2}, e_{j^*})$ (see Definition 2) for generating the components of $z$, the expected suboptimality gap for RERM is at most $\varepsilon = 1/20000$. Then, by Markov's inequality, with probability at least $0.9$ over the choice of sample set $S$, the suboptimality gap is at most $10\varepsilon$.

First, since the population loss separates out nicely in terms of losses for $u, v, \tau$ in (47), we conclude that if $(u', v', \tau')$ is a $10\varepsilon$-suboptimal minimizer for the population loss when components of $z$ are drawn from $\mathcal{D}$, then $u'$, $v'$, and $\tau'$ must be individually $10\varepsilon$-suboptimal minimizers for the functions $-\frac{1}{\sqrt{kn}}\min\{u, 1\}$, $f_N(v)$ and $\mathbb{E}_{z \sim \mathcal{D}}[f_{(A)}(\tau; z)]$ respectively. Additionally, from Lemma 5, we must have that $\|\tau' - e_{j^*}\| \leq 100\varepsilon$. With this insight, for any $j \in [d]$, define the set

$$G_j := \left\{ (u, v, \tau) : -\frac{1}{\sqrt{kn}}\min\{u, 1\} \leq -\frac{1}{\sqrt{kn}} + 10\varepsilon, \ f_N(v) \leq -\frac{1}{2+2\sqrt{n+1}} + 10\varepsilon, \ \|\tau - e_j\| \leq 100\varepsilon \right\}.$$

This set covers all possible $10\varepsilon$-suboptimal minimizers of the population loss. Also, for convenience, we use the notation

$$w = (u, v, \tau).$$

As in the proof of Theorem 2, define the points $w_j^*$ for $j \in \{0, 1, \ldots, d\}$ to be

$$w_j^* \in \operatorname*{argmin}_{w \in G_j} R(w)$$

Now we are ready to define $j^*$: let $j^* \in [d]$ be any element of $\operatorname{argmax}_{j \in [d]} R(w_j^*)$. The proof now follows similarly to the proof of Theorem 2. In particular, in the following, we condition on the occurrence of the event $E$ defined in the proof of Theorem 2.

Now define $c := -\frac{1}{\sqrt{kn}} - \frac{1}{2+2\sqrt{n+1}}$. It is easy to see that for any $w = (u, v, \tau) \in G_j$ for any $j \in [d]$, we have

$$c \leq -\frac{1}{\sqrt{kn}}\min\{u, 1\} + f_N(v) \leq c + 20\varepsilon. \tag{49}$$

Next, consider the point $w_{\widehat{j}}^*$ defined in (12) for the special coordinate $\widehat{j}$. Reasoning similarly to the proof of Theorem 2, we conclude that that $w_{\widehat{j}}^*$ cannot be an $\varepsilon$-suboptimal minimizer of $F$, and thus $w_{\widehat{j}}^*$ can not be a minimizer of the regularized empirical risk (as all RERM solutions are $10\varepsilon$-suboptimal w.r.t. the population loss $F$). Thus, we must have

$$\widehat{F}(w_{\widehat{j}}^*) + R(w_{\widehat{j}}^*) > \min_{w \in G_{j^*}} \left( \widehat{F}(w) + R(w) \right)$$

$$\geq \min_{w \in G_{j^*}} \widehat{F}(w) + \min_{w \in G_{j^*}} R(w)$$

$$\geq R(w_{j^*}^*) + c - 100\varepsilon, \tag{50}$$

where $\widehat{F}$ denotes the empirical loss on $S$,, and the last inequality above follows from (49) and due to the fact that: if $w^* = (u^*, v^*, \tau^*)$ is the minimizer of $\widehat{F}(w)$ over $G_{j^*}$, and $s \in [k]$ is the index such that $u^* \in I_s$, then the function $\tau \mapsto \frac{1}{n}\sum_{i=1}^n y_{i,s}\|\tau \odot x_{i,s}\|$ is 1-Lipschitz and takes the value 0 at $\tau = e_{j^*}$.

On the other hand, if $w_{\widehat{j}}^* = (\widehat{u}, \widehat{v}, \widehat{\tau})$ and $s \in [k]$ is the index such that $\widehat{u} \in I_s$, then using (49), we have

$$\widehat{F}(w_{\widehat{j}}^*) \leq \frac{1}{n}\sum_{i=1}^n y_{i,s}\|(\widehat{\tau} - e_{j^*}) \odot x_{i,s}\| - \frac{1}{\sqrt{kn}}\min\{\widehat{u}, 1\} + f_N(\widehat{v})$$

$$\leq \frac{1}{n}\sum_{i=1}^n y_{i,s}\|(\widehat{\tau} - e_{j^*}) \odot x_{i,s}\| + c + 20\varepsilon. \tag{51}$$

Now, since we are conditioning on the occurrence of the event $E$, using the same chain of inequalities leading to (15), we conclude that

$$\frac{1}{n}\sum_{i=1}^n y_{i,s}\|(\widehat{\tau} - e_{j^*}) \odot x_{i,s}\| \leq 100\varepsilon - \frac{3}{200}. \tag{52}$$

Combining (50), (51), and (52) and rearranging the terms, we get

$$220\varepsilon \geq R(w_{j^*}^*) - R(w_{\widehat{j}}^*) + \frac{3}{200} \geq \frac{3}{200}. \tag{53}$$

where the second inequality above holds because $j^* \in \operatorname{argmax}_{j \in [d]} R(w_j^*)$ (by definition). Thus, $\varepsilon \geq 3/44000 > 1/20000$, a contradiction, as desired. $\qquad \square$

# E    Missing proofs from Section 6

Throughout this section, we assume that a data sample $z$ consists of $(x, \alpha)$, where $x \in \{0, 1\}^d$ and $\alpha \in \{0, e_1, \ldots, e_d\}$. The loss function $f_{(B)} : \mathbb{R}^d \times \mathcal{Z}$ is given by:

$$f_{(B)}(w; z) = \frac{1}{2}\|(w - \alpha) \odot x\|^2 - \frac{c_n}{2}\|w - \alpha\|^2 + \max\{1, \|w\|^4\}, \tag{54}$$

where $c_n := n^{-(\frac{1}{4} - \gamma)}$ for some $\gamma > 0$. We will also assume that $c_n \leq \frac{1}{4}$. Furthermore, since $f_{(B)}(w; z)$ is not differentiable when $\|w\| = 1$, we make the following convention to define the sub-gradient:

$$\partial f(w; z) = (w - \alpha) \odot x - c_n(w - \alpha) + 4\mathbf{1}\{\|w\| > 1\}\|w\|^2 w. \tag{55}$$

Whenever clear from the context in the rest of the section, we will ignore the subscript $(B)$, and denote the loss function by $f(w; z)$. Additionally we define the following distribution over the samples $z = (x, \alpha)$.

**Definition 3.** *For parameters $\delta \in [0, \frac{1}{2}]$, $c_n \in [0, 1]$, and $v \in \{0, e_1, \ldots, e_d\}$, define the distribution $\bar{\mathcal{D}}(\delta, c_n, v)$ over $z = (x, \alpha)$ as follows:*

$$x \sim \mathcal{B}(c_n + \delta)^{\otimes d} \qquad \text{and} \qquad \alpha = v.$$

*The components $x$ and $\alpha$ are sampled independently.*

## E.1    Supporting technical results

Before delving into the proofs, we first establish some technical properties of the empirical loss $f(w; z)$ and the population loss $F(w)$. The following lemma states that for any distribution $\mathcal{D}$, the minimizers of the population loss $F(w) := \mathbb{E}_{z \sim \mathcal{D}}[f(w; z)]$ are bounded in norm.

**Lemma 11.** *Suppose $c_n \leq \frac{1}{4}$. Then for any data distribution $\mathcal{D}$, the minimizer of $F(w) := \mathbb{E}_{z \sim \mathcal{D}}[f(w; z)]$ has norm at most 1.*

**Proof.** Note that for any $w$ such that $\|w\| > 1$, we have

$$f(w; z) \geq -c_n(\|w\|^2 + \|\alpha\|^2) + \|w\|^4 \geq -c_n\|w\|^2 - c_n + \|w\|^4 > 1 - 2c_n.$$

Thus, $F(w) > 1 - 2c_n$. On the other hand, $f(0; z) = \frac{1}{2}\|\alpha \odot x\|^2 - \frac{c_n}{2}\|\alpha\|^2 \leq \frac{1-c_n}{2}$. Hence, $F(0) \leq \frac{1-c_n}{2} \leq 1 - 2c_n$ since $c_n \leq \frac{1}{4}$. This implies that such a $w$ is not a minimizer of $F(w)$.    $\square$

We show now that the single stochastic gradient descent update keeps the iterates bounded as long as the learning rate $\eta$ is chosen small enough.

**Lemma 12.** *Suppose that the learning rate $\eta \leq \frac{1}{100}$ and the point $w$ satisfies $\|w\| \leq 2.5$. Let $w^+ = w - \eta \cdot \partial f(w; z)$ for an arbitrary data point $z$ in the support of the data distribution. Then $\|w^+\| \leq 2.5$.*

**Proof.** We prove the lemma via a case analysis:

1. **Case 1: $\|w\| < 2$.** Using (55) and the Triangle inequality, we get that

$$\|\partial f(w; z)\| \leq \|(w - \alpha) \odot x\| + \|c_n(w - \alpha)\| + 4\|w\|^3$$
$$\leq (1 + c_n)(\|w\| + \|\alpha\|) + 4\|w\|^3 \leq 36, \tag{56}$$

where the last inequality follows from the fact that the iterate $w$ satisfies $\|w\| \leq 2$, and that the parameter $c_n \leq \frac{1}{4}$ and $\|\alpha\| \leq 1$.

Now, for the gradient descent update rule $w_{t+1} = w_t - \eta \partial f(w; z)$, an application of the triangle inequality implies that

$$\|w_{t+1}\| = \|w - \eta \partial f(w; z)\| \leq \|w\| + \eta\|\partial f(w; z)\|. \tag{57}$$

Plugging in the bounds on $\|\partial f(w; z)\|$ derived above, we get that

$$\|w^+\| \leq 2 + 36\eta \leq 2.5 \tag{58}$$

since $\eta \leq \frac{1}{100}$.

2. **Case 2: $2 \leq \|w\| \leq 2.5$.** We start by observing that the new iterate $w^+$ satisfies

$$
\begin{aligned}
\|w^+\|^2 &= \|w_t - \eta \partial f(w; z)\|^2 \\
&= \|w\|^2 + \eta^2 \|\partial f(w; z)\|^2 - 2\eta \langle w, \partial f(w; z) \rangle. \quad\quad (59)
\end{aligned}
$$

Reasoning as in case 1 above, using the fact that $\|w\| \leq 2.5$, we have $\|\partial f(w; z)\| \leq 70$. Furthermore,

$$
\begin{aligned}
\langle w, \partial f(w; z) \rangle &= \langle w, (w - \alpha) \odot x - c_n(w - \alpha) + 4\|w\|^2 w \rangle \\
&= 4\|w\|^4 + \langle w, w \odot x \rangle - \langle w, \alpha \odot x + c_n(w - \alpha) \rangle \\
&\overset{(i)}{\geq} 4\|w\|^4 - \langle w, \alpha \odot x + c_n(w - \alpha) \rangle \\
&\overset{(ii)}{\geq} 4\|w\|^4 - \|w\|((1 + c_n)\|\alpha\| + c_n\|w\|)
\end{aligned}
$$

where the inequality in $(i)$ follows from the fact that $\langle w, w \odot x \rangle \geq 0$, the inequality in $(ii)$ is given by an application of Cauchy-Schwarz inequality followed by Triangle inequality. Next, using the fact that $c_n \leq \frac{1}{4}$ and $\|\alpha\| \leq 1$, we get

$$
\langle w, \partial f(w; z) \rangle \geq 4\|w\|^4 - \|w\|\left(\frac{5}{4} + \frac{1}{4}\|w\|\right) \geq 60, \quad\quad (60)
$$

where the last inequality holds as the polynomial $f(a) := 4a^4 - a(1.25 + 0.25a)$ is an increasing function of $a$ over the domain $a \in [2, \infty)$.

Plugging the bound $\|\partial f(w; z)\| \leq 70$ and and (60) in (59), we have

$$
\|w^+\|^2 \leq \|w\|^2 + 4900\eta^2 - 60\eta \leq \|w\|^2,
$$

since $\eta \leq \frac{1}{100}$. Thus, $\|w^+\| \leq \|w\| \leq 2.5$.

Thus, in either case, $\|w^+\| \leq 2.5$, completing the proof. $\qquad\square$

**Corollary 2.** *Suppose that the learning rate $\eta \leq 1/100$ and the initial point $w_1$ be such that $\|w_1\| \leq 2.5$. Then the iterates obtained when running either full gradient descent on the empirical risk, or by running SGD, have norm bounded by $2.5$ at all times $t \geq 0$.*

**Proof.** The iterates obtained during the running of any of the gradient descent variants described in the statement of the lemma can be seen as convex combinations of single sample gradient descent updates. Thus, the bound on the norm follows immediately from Lemma 12 via the convexity of the norm. $\qquad\square$

The following lemma shows that the loss function $f(w; z)$ is Lipschitz in the domain of interest, i.e. the ball of radius 2.5. This is the region where all iterates produced by gradient based algorithms and the global minimizers of the population loss (for any distribution $\mathcal{D}$) are located in.

**Lemma 13.** *In the ball of radius $2.5$ around $0$, and for any data point $z$, the function $w \mapsto f(w; z)$ is $70$-Lipschitz.*

**Proof.** For $w$ such that $\|w\| \leq 2.5$, using (55) and the Triangle inequality, we get that

$$
\begin{aligned}
\|\partial f(w; z)\| &\leq \|(w - \alpha) \odot x\| + \|c_n(w - \alpha)\| + 4\|w\|^3 \\
&\leq (1 + c_n)(\|w\| + \|\alpha\|) + 4\|w\|^3 \leq 70.
\end{aligned}
$$

$\qquad\square$

### E.2 Proof of Proposition 2

**Proof.** The proof is rather simple. To show this statement, we consider two distributions $\mathcal{D}_1$ and $\mathcal{D}_2$ on the instance space. What we will show is that if a learning algorithm succeeds with a rate any better than $c_n$ on one distribution, then it has to have a worse rate on the other distribution. Thus, we will conclude that any learning algorithm cannot obtain a rate better than $c_n$.

Without further delay, let us define the two distributions we will use in this proof. The first distribution $\mathcal{D}_1$ is given by:

$$x \sim \mathcal{B}\left(\frac{1}{2}\right)^{\otimes d} \quad \text{and} \quad \alpha = 0$$

and the second distribution we consider is $\mathcal{D}_2$ given as follows, first, we draw $\tilde{j} \sim \mathrm{Unif}[d]$, next we set $x[\tilde{j}] = 0$ deterministically, finally, on all other coordinates $i \neq \tilde{j}$, $x[\tilde{j}] \sim \mathrm{unif}\{0,1\}$. We also set $\alpha = 0$ deterministically.

Now the key observation is the following. Since $d > 2^n$, when we draw $n$ samples from distribution $\mathcal{D}_1$ with constant probability there is a coordinate $\hat{j}$ such that $x_t[\hat{j}] = 0$ for all $t \in [n]$. Further, $\hat{j}$ can be any one of the $d$ coordinates with equal probability. However, on the other hand, if we draw $n$ samples from distribution $\mathcal{D}_2$, then by definition of $\tilde{j}$, we have that $x_t[\tilde{j}] = 0$ for any $t \in [n]$. The main observation is that the learning algorithm is agnostic to the distribution on instances. Now since a draw of $n$ samples from $\mathcal{D}_1$ has a coordinate $\hat{j}$ that to the algorithm is indistinguishable from coordinate $\tilde{j}$ when $n$ samples are drawn from $\mathcal{D}_2$, the algorithm cannot identify which of the two distributions the sample if from.

Hence, with constant probability both samples from $\mathcal{D}_1$ and $\mathcal{D}_2$ will be indistinguishable. However note that for distribution $\mathcal{D}_1$ we have

$$F_1(w) = \mathbb{E}_{z \sim \mathcal{D}_1} f(w, z) = \frac{1}{2}\left(\frac{1}{2} - c_n\right)\|w\|^2 + \max\{1, \|w\|^4\},$$

and for distribution $\mathcal{D}_2$,

$$F_2(w) = \mathbb{E}_{z \sim \mathcal{D}_2} f(w, z) = \frac{1}{2}\left(\frac{1}{2} - c_n\right)\|w_{[d]\setminus\{\tilde{j}\}}\|^2 - \frac{c_n}{2}|w[\tilde{j}]|^2 + \max\{1, \|w\|^4\}.$$

Hence notice that for any $w$,

$$F_1(w) - \inf_w F_1(w) = \frac{1}{2}\left(\frac{1}{2} - c_n\right)\|w\|^2 + \max\{1, \|w\|^4\} - 1$$

$$\geq \frac{1}{2}\left(\frac{1}{2} - c_n\right)\|w\|^2 \geq \frac{1}{2}\left(\frac{1}{2} - c_n\right)|w[\tilde{j}]|^2$$

and

$$F_2(w) - \inf_w F_2(w) = \frac{1}{2}\left(\frac{1}{2} - c_n\right)\|w_{[d]\setminus\{\tilde{j}\}}\|^2 - \frac{c_n}{2}|w[\tilde{j}]|^2 + \max\{1, \|w\|^4\} + \frac{c_n}{2} - 1$$

$$\geq \frac{1}{2}\left(\frac{1}{2} - c_n\right)\|w_{[d]\setminus\{\tilde{j}\}}\|^2 - \frac{c_n}{2}|w[\tilde{j}]|^2 + \frac{c_n}{2}$$

$$\geq -\frac{c_n}{2}|w[\tilde{j}]|^2 + \frac{c_n}{2}$$

Now as mentioned before, they key observation is that with constant probability we get a sample using which we cant distinguish between whether we got samples from $\mathcal{D}_1$ or from $\mathcal{D}_2$. Hence in this case, if we want to obtain a good suboptimality, we need to pick a common solution $w$ for which both $F_1(w) - \inf_w F_1(w)$, and $F_2(w) - \inf_w F_2(w)$ are small. However, note that if we want a $w$ for which $F_2(w) - \inf_w F_2(w) \leq c_n/4$, then it must be the case that

$$\frac{c_n}{4} \geq -\frac{c_n}{2}|w[\tilde{j}]|^2 + \frac{c_n}{2}$$

and hence, it must be the case that, $|w[\tilde{j}]| \geq \frac{1}{\sqrt{2}}$. However, for such a $w$ from the above we clearly have that

$$F_1(w) - \inf_w F_1(w) \geq \frac{1}{2}\left(\frac{1}{2} - c_n\right)|w[\tilde{j}]|^2 \geq \frac{1}{4}\left(\frac{1}{2} - c_n\right) \geq \frac{1}{9}$$

(as long as $c_n = o(1)$). Thus we can conclude that no learning algorithm can attain a rate better than $c_n/4$. $\qquad\square$

### E.3 Proof of Theorem 5

We prove part-(a) and part-(b) separately below. The following proof of Theorem 5-(a) is similar to the proof of Theorem 2 given in Appendix B.2.

**Proof of Theorem 5-(a).** In the following proof, we will assume that $n$ is large so that $nc_n^2 \geq 200$ and that $d \geq \ln(10)(1 - c_n)^{-n} + 1$.

Assume, for the sake of contradiction, that there exists a regularizer $R : \mathbb{R}^d \to \mathbb{R}$ such that for any distribution $\mathcal{D} \in \mathscr{D}_c$ the expected suboptimality gap for the RERM solution is at most $\varepsilon/10$. Then, by Markov's inequality, with probability at least $0.9$ over the choice of sample set $S$, the suboptimality gap is at most $\varepsilon$. We will show that $\varepsilon$ must be greater than $c_n^2/3200$ for some distribution in the class $\mathscr{D}_c$, hence proving the desired claim.

We first define additional notation. Set $\delta = c_n/10$ and define $\tilde{\varepsilon} := \varepsilon/\delta$. For the regularization function $R(\cdot)$, define the points $w_j^*$ for $j \in [d]$ such that

$$w_j^* \in \underset{w \text{ s.t. } \|w - e_j\|^2 \leq \tilde{\varepsilon}}{\operatorname{argmin}} R(w). \tag{61}$$

and define the index $j^* \in [d]$ such that $j^* \in \operatorname{argmax}_{j \in [d]} R(w_j^*)$. We are now ready to prove the desired claim.

Consider the data distribution $\mathcal{D}_1 := \bar{\mathcal{D}}(\delta, c_n, e_{j^*})$ (see Definition 3) and suppose that the dataset $S = \{z_i\}_{i=1}^n$ is sampled i.i.d. from $\mathcal{D}_1$. The population loss $F(w)$ corresponding to $\mathcal{D}_1$ is given by

$$F(w) = \mathbb{E}_{z \sim \mathcal{D}_1}[f(w; z)] = \frac{\delta}{2}\|w - e_{j^*}\|^2 + \max\{1, \|w\|^4\}.$$

Clearly, $F(w)$ is convex in $w$ and thus the distribution $\mathcal{D}_1 \in \mathscr{D}_c$. Furthermore, $e_{j^*}$ is the unique minimizer of $F(\cdot)$, and any $\varepsilon$-suboptimal minimizer $w'$ for $F(\cdot)$ must satisfy

$$\|w' - e_{j^*}\|^2 \leq \frac{2\varepsilon}{\delta} = 2\tilde{\varepsilon} \leq \frac{1}{5}. \tag{62}$$

To see the above, note that if $w'$ is an $\varepsilon$-suboptimal minimizer of $F(w)$, then $F(w) \leq F(e_{j^*}) + \varepsilon = 1 + \varepsilon$. However, we also have that $F(w') \geq \delta\|w - e_{j^*}\|^2 + 1$. Taking the two inequalities together, and using the fact that $\tilde{\varepsilon} \leq 1/10$, we get the desired bound on $\|w - e_{j^*}\|$.

Next, define the event $E$ such that the following hold:

$(a)$ For the coordinate $j^*$, we have $\sum_{z \in S} x[j^*] \leq n(c_n + 2\delta)$.

$(b)$ There exists $\widehat{j}$ such that $\widehat{j} \neq j^*$ and $x[\widehat{j}] = 0$ for all $z \in S$.

$(c)$ RERM (with regularization $R(\cdot)$) using the dataset $S$ returns an $\varepsilon$-suboptimal solution for the test loss $F(w)$.

Since $x[j^*] \sim \mathcal{B}(c_n + \delta)$, Hoeffding's inequality (Lemma 1) implies that the event (a) above occurs with probability at least $0.8$ for $n \geq 2/\delta^2 = 200/c_n^2$. Furthermore, Lemma 2 gives us that the event (b) above occurs with probability at least $0.9$ for $d \geq \ln(10)(1 - c_n)^{-n}$. Finally, the assumed performance guarantee for RERM with regularization $R(\cdot)$ implies that the event (c) above occurs with probability at least $0.9$. Thus, the event $E$ occurs with probability at least $0.6$. In the following, we condition on the occurrence of the event $E$.

Consider the point $w_{\widehat{j}}^*$, defined in (61) corresponding to the coordinate $\widehat{j}$ (that occurs in event $E$). By definition, we have that $\|w_{\widehat{j}}^* - e_{\widehat{j}}\|^2 \leq \tilde{\varepsilon}$, and thus

$$\|w_{\widehat{j}}^* - e_{j^*}\|^2 \geq \frac{1}{2}\|e_{\widehat{j}} - e_{j^*}\|^2 - \|w_{\widehat{j}}^* - e_{\widehat{j}}\|^2$$

$$\geq 1 - \tilde{\varepsilon} \geq \frac{1}{4},$$

where the first line above follows from the identity that $(a + b)^2 \leq 2a^2 + 2b^2$ for any $a, b > 0$, and the second line holds because $\widehat{j} \neq j^*$ and because $\tilde{\varepsilon} \leq 1/10$. As a consequence of the above bound

and the condition in (62), we get that the point $w_{\widehat{j}}^*$ is not an $\varepsilon$-suboptimal point for the population loss $F(\cdot)$, and thus would not be the solution of the RERM algorithm (as the RERM solution is $\varepsilon$-suboptimal w.r.t $F(w)$). Since, any RERM must satisfy condition (62), we have that

$$
\begin{aligned}
\widehat{F}(w_{\widehat{j}}^*) + R(w_{\widehat{j}}^*) &> \min_{w:\ \|w - e_{j^*}\|^2 \leq \tilde{\varepsilon}} \left( \widehat{F}(w) + R(w) \right) \\
&\geq \min_{w:\ \|w - e_{j^*}\|^2 \leq \tilde{\varepsilon}} \widehat{F}(w) + \min_{w:\ \|w - e_{j^*}\|^2 \leq \tilde{\varepsilon}} R(w) \\
&\overset{(i)}{\geq} \min_{w:\ \|w - e_{j^*}\|^2 \leq \tilde{\varepsilon}} \left( -\frac{c_n}{2} \|w - e_{j^*}\|^2 + \max\{1, \|w\|^4\} \right) + R(w_{j^*}^*) \\
&\geq \min_{w:\ \|w - e_{j^*}\|^2 \leq \tilde{\varepsilon}} \left( -\frac{c_n}{2} \|w - e_{j^*}\|^2 + 1 \right) + R(w_{j^*}^*) \\
&= -\frac{c_n}{2} \cdot \tilde{\varepsilon} + 1 + R(w_{j^*}^*) \\
&\overset{(ii)}{\geq} -\frac{c_n}{20} + 1 + R(w_{j^*}^*)
\end{aligned}
\tag{63}
$$

where $\widehat{F}(w) := \frac{1}{n} \sum_{i=1}^n f(w; z_i)$ denotes the empirical loss on the dataset $S$, the inequality $(i)$ follows by ignoring non-negative terms in the empirical loss $\widehat{F}(w)$, and the inequality $(ii)$ is due to the fact that $\tilde{\varepsilon} \leq 1/10$. For the left hand side, we note that

$$
\begin{aligned}
\widehat{F}(w_{\widehat{j}}^*) &= \frac{1}{2n} \sum_{(x,\alpha) \in S} \|(w_{\widehat{j}}^* - e_{j^*}) \odot x\|^2 - \frac{c_n}{2} \|(w_{\widehat{j}}^* - e_{j^*})\|^2 + \max\{1, \|w_{\widehat{j}}^*\|^4\} \\
&\overset{(i)}{\leq} \tilde{\varepsilon} + \frac{1}{2n} \sum_{(x,\alpha) \in S} \|(e_{\widehat{j}} - e_{j^*}) \odot x\|^2 - \frac{c_n}{2} \|(e_{\widehat{j}} - e_{j^*})\|^2 + \max\{1, (1 + \tilde{\varepsilon})^4\} \\
&\overset{(ii)}{\leq} 16\tilde{\varepsilon} + \frac{1}{2n} \sum_{(x,\alpha) \in S} \|(e_{\widehat{j}} - e_{j^*}) \odot x\|^2 - \frac{c_n}{2} \|(e_{\widehat{j}} - e_{j^*})\|^2 + 1 \\
&\overset{(iii)}{\leq} 16\tilde{\varepsilon} + \frac{1}{2}(c_n + 2\delta) - \frac{\sqrt{2}c_n}{2} + 1,
\end{aligned}
\tag{64}
$$

where the inequality $(i)$ above holds because $\|w_{\widehat{j}}^* - e_{\widehat{j}}\| \leq \tilde{\varepsilon}$ (by definition) and because $\|e_{\widehat{j}}\| = 1$. The inequality in $(ii)$ follows from the fact that $(1 + a)^4 \leq 1 + 15a$ for $a < 1$. Finally, the inequality $(iii)$ is due to the fact that $x[\widehat{j}] = 0$ for all $(x, \alpha) \in S$ and because $\sum_{(x,\alpha) \in S} x[j^*] \leq n(c_n + 2\delta)$ due to the conditioning on the event $E$.

Combining the bounds in (63) and (64), plugging in $\delta = c_n/10$, and rearranging the terms, we get that

$$
16\tilde{\varepsilon} \geq \frac{c_n}{20} + R(w_{j^*}^*) - R(w_{\widehat{j}}^*) \geq \frac{c_n}{20},
$$

where the second inequality above holds because $j^* \in \operatorname{argmax}_{j \in [d]} R(w_j^*)$ (by definition). Since, $\tilde{\varepsilon} = \varepsilon/\delta$, we conclude that

$$
\varepsilon \geq \delta \cdot \frac{c_n}{320} \geq \frac{c_n^2}{3200}.
$$

The above suggests that for data distribution $\mathcal{D}_1$, RERM algorithm with the regularization function $R(w)$ will suffer an excess risk of at least $\Omega(c_n^2)$ (with probability at least 0.9). Finally note that $R(w)$ could be any arbitrary function in the above proof, and thus we have the desired claim for any regularization function $R(w)$. $\qquad\square$

We now prove Theorem 5-(b). In the following, we give convergence guarantee for SGD algorithm given in (6) when run with step size $\eta = 1/20\sqrt{n} < 1/100$ using the dataset $S = \{z_i\}_{i=1}^n$ drawn

i.i.d. from a distribution $\mathcal{D}$. We further assume that assume that the initial point $w_1 = 0$, and thus $\|w_1\| \leq 2.5$. Note that having initial weights to be bounded is typical when learning with non-convex losses, for eg. in deep learning.

**Proof of Theorem 5-(b).** We consider the two cases, when $D \in \mathscr{D}_c$ and when $D \notin \mathscr{D}_c$, separately below:

**Case 1: When $\mathcal{D} \in \mathscr{D}_c$ .** In this case, we note the population loss $F(w) := \mathbb{E}_{z \sim \mathcal{D}}[f(w; z)]$ is convex in $w$ by the definition of the set $\mathscr{D}_c$ in (8). Further, since $\eta = 1/20\sqrt{n} < 1/100$, Lemma 12 and Lemma 13 imply that $f$ is 70-Lipschitz. Finally, due to Lemma 11, the initial point $w_1 = 0$ satisfies $\|w_1 - w^*\| \leq 2.5$. Thus, we satisfy both Assumption I and Assumption II in (2) and (3) respectively, and an application of Theorem 1 implies that the point $\widehat{w}_n^{\mathrm{SGD}}$ enjoys the performance guarantee

$$\mathbb{E}\big[F(\widehat{w}_n^{\mathrm{SGD}}) - w^* \mid E\big] \leq O\Big(\frac{1}{\sqrt{n}}\Big).$$

**Case 2: When $\mathcal{D} \notin \mathscr{D}_c$ .** In order to prove the performance guarantee of SGD algorithm in this case, we split up the loss function $f(w; z)$ into convex and non-convex parts $g$ and $\widetilde{g}$, defined as:

$$g(w; z) := \frac{1}{2}\|(w - \alpha) \odot x\|^2 + \max\{1, \|w\|^4\} \qquad \text{and} \qquad \widetilde{g}(w; z) := -c_n\|w - \alpha\|^2.$$

Further, we define the functions $G(w)$ and $\widetilde{G}(w)$ to denote their respective population counterparts, i.e. $G(w) = \mathbb{E}_{z \sim \mathcal{D}}[g(w; z)]$ and $\widetilde{G}(w) = \mathbb{E}_{z \sim \mathcal{D}}[\widetilde{g}(w; z)]$. Clearly,

$$F(w) = \mathbb{E}_{z \sim \mathcal{D}}[f(w; z)] = G(w) + \widetilde{G}(w). \tag{65}$$

Let $w^*$ be a minimizer of $F(w; \mathcal{D})$. By Lemma 11, we have $\|w^*\| \leq 1$. The folltowing chain of arguments follows along the lines of proof of SGD in Case 1 above (see the proof of Theorem 8 on page 14).

Let the sequence of iterates generated by SGD algorithm be given by $\{w_t\}_{t=1}^T$. We start by observing that for any $t \geq 0$,

$$\begin{aligned}
\|w_{t+1} - w^*\|_2^2 &= \|w_{t+1} - w_t + w_t - w^*\|_2^2 \\
&= \|w_{t+1} - w_t\|_2^2 + \|w_t - w^*\|_2^2 + 2\langle w_{t+1} - w_t, w_t - w^*\rangle \\
&= \|-\eta\nabla f_w(w_t; z_t)\|_2^2 + \|w_t - w^*\|_2^2 + 2\langle -\eta\nabla f_w(w_t; z_t), w_t - w^*\rangle,
\end{aligned}$$

where the last line follows from plugging in the update step $w_{t+1} = w_t - \eta\nabla f_w(w_t; z_t)$. Rearranging the terms in the above, we get that

$$\begin{aligned}
\langle \nabla f(w_t; z_t), w_t - w^*\rangle &\leq \frac{\eta}{2}\|\nabla f(w_t; z_t)\|_2^2 + \frac{1}{2\eta}\big(\|w_t - w^*\|_2^2 - \|w_{t+1} - w^*\|_2^2\big) \\
&\leq 2450\eta + \frac{1}{2\eta}\big(\|w_t - w^*\|_2^2 - \|w_{t+1} - w^*\|_2^2\big), \tag{66}
\end{aligned}$$

where the second inequality in the above follows from the bound on the Lipschitz constant of the function $f$ when the iterates stay bounded in a ball of radius 2.5 around 0 (see Lemma 13). We split the left hand side as:

$$\langle \nabla f(w_t; z_t), w_t - w^*\rangle = \langle \nabla g(w_t; z_t), w_t - w^*\rangle + \langle \nabla \widetilde{g}(w_t; z_t), w_t - w^*\rangle.$$

This implies that

$$\langle \nabla g(w_t; z_t), w_t - w^*\rangle \leq 2450\eta + \frac{1}{2\eta}\big(\|w_t - w^*\|_2^2 - \|w_{t+1} - w^*\|_2^2\big) + \langle \nabla \widetilde{g}(w_t; z_t), w^* - w_t\rangle.$$

Taking expectation on both the sides with respect to the data sample $z_t$, while conditioning on the point $w_t$ and the occurrence of the event $E$, we get

$$\langle \nabla G(w_t), w_t - w^*\rangle \leq 2450\eta + \frac{1}{2\eta}\mathbb{E}\big[\|w_t - w^*\|_2^2 - \|w_{t+1} - w^*\|_2^2\big] + \langle \nabla \widetilde{G}(w_t), w^* - w_t\rangle.$$

Next, note that the function $G(w)$ is convex (by definition). This implies that $G(w^*) \geq G(w_t) - \langle \nabla G(w_t), w_t - w^* \rangle$, plugging which in the above relation gives us

$$G(w_t) - G(w^*) \leq 2450\eta + \frac{1}{2\eta} \mathbb{E}\big[\|w_t - w^*\|_2^2 - \|w_{t+1} - w^*\|_2^2\big] + \langle \nabla \widetilde{G}(w_t), w^* - w_t \rangle.$$

Telescoping the above for $t$ from 1 to $n$, we get that

$$\sum_{t=1}^{n}(G(w_t) - G(w^*)) \leq 2450\eta n + \frac{1}{2\eta}\big(\|w_1 - w^*\|_2^2 - \|w_{n+1} - w^*\|_2^2\big) + \sum_{t=1}^{n}\langle \nabla \widetilde{G}(w_t), w^* - w_t \rangle$$

$$\leq 2450\eta n + \frac{\|w_1 - w^*\|_2^2}{2\eta} + \sum_{t=1}^{n}\|\nabla \widetilde{G}(w_t)\|\|w^* - w_t\|,$$

where the inequality in the second line follows by ignoring negative terms, and through an application of Cauchy-Schwarz inequality. Since, $\|w_1\| = 0$ and $\|w^*\| \leq 1$ (due to [Lemma 11]), we have that $\|w_1 - w^*\| \leq 1$. Setting $\eta = 1/20\sqrt{n}$, using the bound $\|w_1 - w^*\| \leq 1$ and by an application of Jensen's inequality in the left hand side, we get that the point $\widehat{w}_n^{\text{SGD}} := \frac{1}{n}\sum_{t=1}^{n} w_t$ satisfies

$$G(\widehat{w}_n^{\text{SGD}}) - G(w^*) \leq \frac{245}{\sqrt{n}} + \frac{1}{n}\sum_{t=1}^{n}\|\nabla \widetilde{G}(w_t)\|\|w^* - w_t\|. \tag{67}$$

Furthermore, by [Lemma 12], since $\eta = \frac{1}{20\sqrt{n}} \leq \frac{1}{100}$ and $\|w_1\| \leq 2.5$ (by construction), we have that $\|w_t\| \leq 2.5$ for all $t$. Thus, $\|w^* - w_t\| \leq 3.5$ for all $t$. Plugging this bound in (67), we get that

$$G(\widehat{w}_n^{\text{SGD}}) - G(w^*) \leq \frac{245}{\sqrt{n}} + 3.5 \max_{t \leq n} \|\nabla \widetilde{G}(w_t)\|.$$

Next, note that from the definition of the function $\widetilde{G}(w_t)$, we have that

$$\|\nabla \widetilde{G}(w_t)\| = c_n\|w_t - \alpha\| \leq 3.5c_n,$$

and thus

$$G(\widehat{w}_n^{\text{SGD}}) - G(w^*) \leq \frac{245}{\sqrt{n}} + 3.5c_n. \tag{68}$$

Finally, using the relation (65) and taking expectations on both the sides, we have that

$$\mathbb{E}\big[F(\widehat{w}_n^{\text{SGD}}) - F(w^*) \mid E\big] = \mathbb{E}\big[G(\widehat{w}_n^{\text{SGD}}) - G(w^*)\big] + \mathbb{E}\big[\widetilde{G}(\widehat{w}_n^{\text{SGD}}) - \widetilde{G}(w^*)\big]$$

$$\leq \frac{245}{\sqrt{n}} + 3.5c_n + c_n\, \mathbb{E}_z[w^* - \alpha\|^2]$$

$$\leq \frac{245}{\sqrt{n}} + 6c_n, \tag{69}$$

where the inequality in the second line is due to (68) and by using the definition of the function $\widetilde{G}(w)$ and by ignoring negative terms. The last line holds because $\|w^* - \alpha\| \leq \|w^*\| + \|\alpha\| \leq 2$. $\qquad \square$

# F   Missing proofs from [Section 7]

## F.1   $\alpha$-Linearizable functions

In the following, provide the proof of the performance guarantee for SGD for $\alpha$-Linearizable functions.

**Proof of [Theorem 6].** The proof follows along the lines of the proof of [Theorem 8] on page 14. Let $\{w_t\}_{t \geq 1}$ denote the sequence of iterates generated by the SGD algorithm. We note that for any $w^* \in \operatorname{argmin}_w F(w)$ and time $t \geq 1$,

$$\|w_{t+1} - w^*\|_2^2 = \|w_{t+1} - w_t + w_t - w^*\|_2^2$$

$$= \|w_{t+1} - w_t\|_2^2 + \|w_t - w^*\|_2^2 + 2\langle w_{t+1} - w_t, w_t - w^* \rangle$$
$$= \|-\eta \nabla f(w_t; z_t)\|_2^2 + \|w_t - w^*\|_2^2 + 2\langle -\eta \nabla f(w_t; z_t), w_t - w^* \rangle,$$

where the last line follows from plugging in the SGD update rule that $w_{t+1} = w_t - \eta \nabla f(w_t; z_t)$. Rearranging the terms in the above, we get that

$$\langle \nabla f(w_t; z_t), w_t - w^* \rangle \le \frac{\eta}{2} \|\nabla f(w_t; z_t)\|_2^2 + \frac{1}{2\eta} \big( \|w_t - w^*\|_2^2 - \|w_{t+1} - w^*\|_2^2 \big).$$

Taking expectation on both the sides, while conditioning on the point $w_t$, implies that

$$\langle \nabla F(w_t), w_t - w^* \rangle \le \frac{\eta}{2} \mathbb{E}\big[ \|\nabla f(w_t; z_t)\|_2^2 \big] + \frac{1}{2\eta} \big( \|w_t - w^*\|_2^2 - \|w_{t+1} - w^*\|_2^2 \big)$$
$$\le \eta \mathbb{E}\big[ \|\nabla f(w_t; z_t) - \nabla F(w_t)\|_2^2 \big] + \eta \|\nabla F(w_t)\|_2^2$$
$$+ \frac{1}{2\eta} \big( \|w_t - w^*\|_2^2 - \|w_{t+1} - w^*\|_2^2 \big)$$
$$\le \eta(\sigma^2 + L^2) + \frac{1}{2\eta} \big( \|w_t - w^*\|_2^2 - \|w_{t+1} - w^*\|_2^2 \big), \tag{70}$$

where the inequality in the second line is given by the fact that $(a - b)^2 \le 2a^2 + 2b^2$ and the last line follows from using Assumption II (see (3)) which implies that $F(w)$ is $L$-Lipschitz in $w$ and that $\mathbb{E}[\|\nabla f(w; z_t) - \nabla F(w)\|_2^2] \le \sigma^2$ for any $w$. Next, using the fact that $F$ is $\alpha$-Linearizable, we have that there exists a $w^* \in \operatorname{argmin}_w F(w)$ such that for any $w$,

$$F(w) - F(w^*) \le \alpha \langle \nabla F(w), w - w^* \rangle. \tag{71}$$

Setting $w^*$ to the one that satisfies (71), and using the above bound for $w = w_t$ with the bound in (70), we get that for any $t \ge 1$,

$$\frac{1}{\alpha}(F(w_t) - F^*) \le \eta(\sigma^2 + L^2) + \frac{1}{2\eta} \big( \|w_t - w^*\|_2^2 - \|w_{t+1} - w^*\|_2^2 \big),$$

where $F^* := F(w^*)$. Telescoping the above for $t$ from 1 to $n$, we get that

$$\sum_{t=1}^{n} \frac{1}{\alpha}(F(w_t) - F^*) \le \eta n(\sigma^2 + L^2) + \frac{1}{2\eta} \big( \|w_1 - w^*\|_2^2 - \|w_{n+1} - w^*\|_2^2 \big)$$
$$\le \eta n(\sigma^2 + L^2) + \frac{1}{2\eta} \|w_1 - w^*\|_2^2.$$

Dividing both the sides by $n$, we get that

$$\frac{1}{\alpha n} \sum_{t=1}^{n} (F(w_t) - F^*) \le \eta(\sigma^2 + L^2) + \frac{1}{2\eta n} \|w_1 - w^*\|_2^2.$$

An application of Jensen's inequality on the left hand side, implies that for the point $\widehat{w}_n^{\mathrm{SGD}} := \frac{1}{n} \sum_{t=1}^{n} w_t$, we have that

$$\frac{1}{\alpha} \mathbb{E}\big[ F(\widehat{w}_n^{\mathrm{SGD}}) - F^* \big] \le \eta(\sigma^2 + L^2) + \frac{1}{2\eta n} \|w_1 - w^*\|_2^2.$$

Setting $\eta = \frac{1}{\sqrt{n}}$ and using the fact that $\|w_1 - w^*\|_2 \le B$ in the above, we get that

$$\mathbb{E}\big[ F(\widehat{w}_n^{\mathrm{SGD}}) - F^* \big] \le \frac{\alpha}{\sqrt{n}} \big( \sigma^2 + L^2 + B^2 \big),$$

which is the desired claim. Dependence on the problem specific constants ($\sigma, L$ and $B$) in the above bound can be improved further with a different choice of the step size $\eta$; getting the optimal dependence on these constants, however, is not the focus of this work. $\qquad \square$

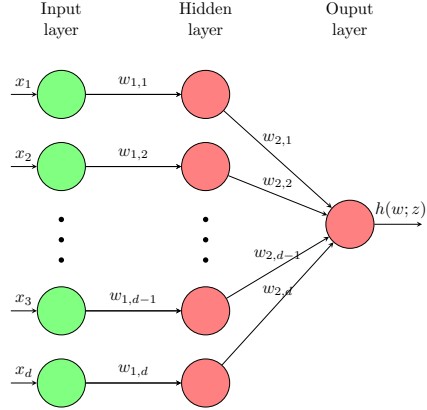

Figure 2: Two layer diagonal neural network. The weights are given by $w_1 \in \mathbb{R}^d$ and $w_2 \in \mathbb{R}^d$ for the first layer and the second layer respectively. The green nodes denote input nodes with linear activation function and red node denote hidden units with ReLU activation function.

### F.2 Proof of Theorem 7

Let the input $\mathcal{X} \in \{0,1\}^d$ and the label $\mathcal{Y} \in [-1,1]$. Consider a two layer neural network with ReLU activation and weights given by $w = (w_1, w_2)$ where $w_1 \in \mathbb{R}^d$ denotes the weights of the first layer and $w_2 \in \mathbb{R}^d$ denotes the weights of the second layer, as shown in Figure 2. When given the input $x$, the output of the network with weights $(w_1, w_2)$ is computed as

$$h(w;x) = \mathrm{ReLU}(w_2^\top \mathrm{ReLU}(w_1 \odot x)).$$

Here, the first layer of the neural network has sparse connections, i.e. each input node connects to only one hidden node. Such networks are denoted as diagonal two layer neural networks. We assume that the neural network is trained using absolute loss function. Specifically, the instantaneous loss on a sample $z = (x, y)$ is given by

$$f(w;z) = |y - h(w;x)| = |y - \mathrm{ReLU}(w_2^\top \mathrm{ReLU}(w_1 \odot x))|. \tag{72}$$

Since $f(w;z)$ is not smooth, for any weights $w = (w_1, w_2)$ and sample $z = (x, y)$, we define the gradient of $f(w;z)$ as

$$\nabla_{w_1} \ell(w;z)[i] = -\mathrm{sign}\{y - h(w;x)\} \cdot \mathbf{1}\{w_2^\top \mathrm{ReLU}(w_1 \odot x) > 0\} \cdot (w_{2,i} \cdot \mathbf{1}\{w_{1,i} \cdot x_i > 0\} \cdot x_i)$$

and

$$\nabla_{w_2} \ell(w;z)[i] = -\mathrm{sign}\{y - h(w;x)\} \cdot \mathbf{1}\{w_2^\top \mathrm{ReLU}(w_1 \odot x) > 0\} \cdot (\mathrm{ReLU}(w_{1,i} \cdot x_i)), \tag{73}$$

for $i \in [d]$. We next show that the population loss $F(w) := \mathbb{E}_{z \sim \mathcal{D}}[f(w;z)]$ is 1/2-Linearizable.

**Lemma 14.** *Let $\mathcal{D}$ be defined such that $x$ and $y$ are independent random variables with distributions*

$$x \sim Uniform(\{0,1\}^d) \qquad and \qquad y \sim \mathcal{B}(1/4).$$

*Then, the population loss $F(w) := \mathbb{E}_{z \sim \mathcal{D}}[f(w;z)]$ is 1/2-Linearizable.*

**Proof.** The population loss $F(w)$ is given by:

$$F(w) = \mathbb{E}_{x,y}[|y - h(w;x)|],$$

where $h(w;x) = \mathrm{ReLU}(w_2^\top \mathrm{ReLU}(w_1 \odot x))$. Using the fact that $y \in \{-1, 1\}$, and $\Pr(y = 1) = 1/4$ and is independent of $x$, the above can be written as:

$$F(w) = \mathbb{E}_x \left[ \frac{3}{4} |h(w;x)| + \frac{1}{4} |1 - h(w;x)| \right].$$

It is easy to verify that $F(w)$ is minimized when $h(w;x) = 0$ for every $x \in \{0,1\}^d$, which occurs at the point $w = 0$. Furthermore, $F(0) = 1/4$. Next, note that for any $w$, and sample $z = (x,y)$,

$$\langle w, \nabla_w f(w;z) \rangle = \langle w_1, \nabla_{w_1} f(w;z) \rangle + \langle w_2, \nabla_{w_2} f(w;z) \rangle$$

$$\overset{(i)}{=} -\sum_{i=1}^{d} w_{1,i} \cdot \operatorname{sign}\{y - h(w;x)\} \cdot \mathbf{1}\{w_2^\top \operatorname{ReLU}(w_1 \odot x) > 0\} \cdot (w_{2,i} \cdot \mathbf{1}\{w_{1,i} \cdot x_i > 0\} \cdot x_i)$$

$$\phantom{=} - \sum_{i=1}^{d} w_{2,d} \cdot \operatorname{sign}\{y - h(w;x)\} \cdot \mathbf{1}\{w_2^\top \operatorname{ReLU}(w_1 \odot x) > 0\} \cdot (\operatorname{ReLU}(w_{1,i} \cdot x_i))$$

$$= -\operatorname{sign}\{y - h(w;x)\} \cdot \mathbf{1}\{w_2^\top \operatorname{ReLU}(w_1 \odot x) > 0\} \cdot (w_2^\top \operatorname{ReLU}(w_1 \odot x))$$

$$\phantom{=} - \operatorname{sign}\{y - h(w;x)\} \cdot \mathbf{1}\{w_2^\top \operatorname{ReLU}(w_1 \odot x) > 0\} \cdot (w_2^\top \operatorname{ReLU}(w_1 \odot x))$$

$$= -2\operatorname{sign}\{y - h(w;x)\} \operatorname{ReLU}(w_2^\top \operatorname{ReLU}(w_1 \odot x))$$

$$= -2\operatorname{sign}\{y - h(w;x)\} h(w;x)$$

$$= 2|y - h(w;x)| - 2\operatorname{sign}\{y - h(w;x)\} y$$

where the equality $(i)$ follows from using the definition of $\nabla_{w_1} f(w;z)$ and $\nabla_{w_2} f(w;z)$ from (73). Taking expectations on both the sides with respect to $z$, we get that

$$\langle w, \nabla_w F(w) \rangle = 2\,\mathbb{E}_{x,y}[|y - h(w;x)|] - 2\,\mathbb{E}_{x,y}[\operatorname{sign}\{y - h(w;x)\} y]$$

$$= 2F(w) - 2\,\mathbb{E}_{x,y}[\operatorname{sign}\{y - h(w;x)\} y]$$

$$\geq 2F(w) - 2\,\mathbb{E}_{x,y}[|y|]$$

$$= 2(F(w) - F(0)),$$

where the last line follows by observing that $\mathbb{E}[|y|] = \mathbb{E}_{x,y}[|y| - h(0;x)] = F(0)$. Defining $w^* := 0$, the above implies that

$$F(w) - F(w^*) \leq \frac{1}{2} \langle w - w^*, \nabla_w F(w) \rangle,$$

thus showing that $F(w)$ is $1/2$-Linearizable. $\qquad \square$

**Proof of Theorem 7.** Consider the distribution $\mathcal{D}$ over the instance space $\mathcal{Z} = \{0,1\}^d \times \{0,1\}$ where

$$x \sim \operatorname{Uniform}(\{0,1\}^d), \qquad \text{and} \qquad y \sim \mathcal{B}(1/4). \tag{74}$$

We now prove the two parts separately below:

(a) In the following, we show that SGD algorithm, run with an additional projection on the unit norm ball after every update, learns at a rate of $O(1/\sqrt{n})$. In particular, we use the following update step for $t \in [n]$,

$$w_{t+1}^{\text{SGD}} \leftarrow \Pi_{w_1}\left(w_t^{\text{SGD}} - \eta \nabla f(w_t^{\text{SGD}}; z_t)\right)$$

where the initial point $w_1$ is chosen by first sampling $w_1' \sim \mathcal{N}(0, \mathbb{I}_d)$ and then setting $w_1 = w_1'/\|w_1\|$, and the projection operation $\Pi$ is given by

$$\Pi_{w_1}(w) = \begin{cases} w & \text{if } \|w - w_1\| \leq 1 \\ w_1 + \frac{1}{\|w - w_1\|}(w - w_1) & \text{otherwise.} \end{cases}$$

After taking $n$ steps, the point returned by the SGD algorithm is given by $\widehat{w}_n^{\text{SGD}} := \sum_{t=1}^{n} w_t^{\text{SGD}}/n$.

First note that, for the distribution given in (74), Lemma 14 implies that the population loss $F(w)$ is $1/2$-Liearizable w.r.t. the global minima $w^* = 0$. Next, note that for any point $w$ and data sample $z$,

$$\|\nabla f(w;z)\|^2 = \|\nabla_{w_1} f(w;z)\|^2 + \|\nabla_{w_2} f(w;z)\|^2$$

$$\leq \sum_{j=1}^{d} (w_{2,j} \cdot x_j)^2 + \sum_{j=1}^{d} (\text{ReLU}(w_{1,j} \cdot x_j))^2$$

$$\leq \sum_{j=1}^{d} w_{2,j}^2 + \sum_{j=1}^{d} w_{1,j}^2$$

$$= \|w\|^2$$

where the inequality in the second line follows by plugging in the definition of $\nabla_{w_1} f(w; z)$ and $\nabla_{w_2} f(w; z)$, and by upper bounding the respective indicators by 1. The inequality in the third line above holds because $\text{ReLU}(h) \leq |h|$ for any $h \in \mathbb{R}$, and by using the fact that $x_j \in \{0, 1\}$ for $j \in [d]$. Since, the iterates produced by SGD algorithm satisfy $\|w_t^{\text{SGD}}\| \leq 1$ due to the projection step, the above bound implies that $\|\nabla f(w_t^{\text{SGD}}; z)\| \leq 1$ for any $t \geq 0$, and thus $\|\nabla F(w_t^{\text{SGD}})\| \leq 1$.

The above bounds imply that Assumption II (in (3)) holds on the iterates generated by the SGD algorithm with $\max\{B, \sigma, L\} \leq 2$. Furthermore, the population loss $F(w)$ is $1/2$-Linearizable. Thus, repeating the steps from the proof of Theorem 6 on page 46, and using the fact that $\|\Pi_{w_1}(w) - w^*\| \leq \|w - w^*\|$ for $w^* = 0$ to account for the additional projection step, we get that the point $\widehat{w}_n^{\text{SGD}}$ returned by the SGD algorithm enjoys the performance guarantee

$$F(\widehat{w}_n^{\text{SGD}}) - F^* \leq O\left(\frac{1}{\sqrt{n}}\right).$$

(b) In the following, we will show that for $d > \log(10) 2^n + 1$, with probability at least 0.9 over the choice of $S \sim \mathcal{D}^n$, there exists an ERM solution for which

$$F(w_{\text{ERM}}) - \inf_{w \in \mathbb{R}^d} F(w) \geq \Omega(1).$$

Suppose that the dataset $S = \{(x_i, y_i)\}_{i=1}^n$ is sampled i.i.d. from the distribution $\mathcal{D}$ given in (74). A slight modification of Lemma 6 implies that for $n \leq \log_2(d/\log(10))$, with probability at least 0.9, there exists a coordinate $\widehat{j}$ such that $x_i[\widehat{j}] = y_i$ for all $i \in [n]$. In the following, we condition on the occurrence of such a coordinate $\widehat{j}$. Clearly, the empirical loss at the point $\widehat{w} = (e_{\widehat{j}}, e_{\widehat{j}})$ is:

$$\widehat{F}(\widehat{w}) = \sum_{i=1}^{n} |y_i - \text{ReLU}(e_{\widehat{j}} \text{ReLU}(e_{\widehat{j}} \odot x_i))| = \sum_{i=1}^{n} |y_i - x_i[\widehat{j}]| = 0,$$

where the last equality follows the fact that $x_i[\widehat{j}] = y_i$ for all $i \in [n]$. Since $\widehat{F}(w) \geq 0$ for any $w$, we get that the point $\widehat{w}$ is an ERM solution. Next, we note that the population loss at the point $\widehat{w}$ satisfies

$$F(\widehat{w}) - \min_{w} F(w) \overset{(i)}{=} F(\widehat{w}) - \frac{1}{4}$$

$$\overset{(ii)}{=} \mathbb{E}_{x,y}\left[\left|y - \text{ReLU}(e_{\widehat{j}} \text{ReLU}(e_{\widehat{j}} \odot x))\right|\right] - \frac{1}{4}$$

$$= \mathbb{E}_{x,y}[|y - x[\widehat{j}]|] - \frac{1}{4} \overset{(iii)}{=} \frac{1}{4}$$

where the equality $(i)$ follows by observing that $\min_w F(w) = 1/4$ (see the proof of Lemma 14 for details), the equality $(ii)$ holds for $\widehat{w} = (e_{\widehat{j}}, e_{\widehat{j}})$ and $(iii)$ follows by using the fact that $y \sim \mathcal{B}(1/4)$ and $x[\widehat{j}] \sim \mathcal{B}(1/2)$ and that $x$ and $y$ are sampled independent to each other. Thus, there exists an ERM solution $w_{\text{ERM}} = \widehat{w}$ for which the excess risk:

$$F(w_{\text{ERM}}) - \min_{w} F(w) = \frac{1}{4}.$$

The desired claim follows by observing that the coordinate $\widehat{j}$ above occurs with probability at least 0.9 over the choice of $S \sim \mathcal{D}^n$.

$$\square$$

### F.3 Expressing $f_{(A)}$ and $f_{(B)}$ using neural networks

In this section, we show that the loss functions $f_{(A)}$ and $f_{(B)}$ can be represented using restricted neural networks (where some of the weights are fixed to be constant and thus not trained) with poly($d$) hidden units. We first provide neural network constructions that in addition to ReLU, use square function $\sigma(a) = a^2$ and square root function $\widetilde{\sigma}(a) = \sqrt{a}$ as activation functions. We then give a general representation result in Lemma 15 which implies that the activations $\sigma$ and $\widetilde{\sigma}$ can be approximated both in value and in gradients simultaneously using poly($d$) number of ReLU units. This suggests that the functions $f_{(A)}$ and $f_{(B)}$ can be represented using restricted RELU networks of poly($d$) size.

Note that in our constructions, the NN approximates the corresponding functionst in both value and in terms of the gradient. Thus, running gradient based optimization algorithms on the corresponding NN representation will produce similar solutions as running the same algorithm on the actual function that it is approximating.

**Proposition 3.** *Function $f_{(A)}$ in Equation (A) can be represented as a restricted diagonal neural network with square and square root activation functions with $O(d)$ units and constant depth.*

**Proof.** In the following, we will assume that before passing to the neural network, each data sample $z = (x, \alpha, y)$ is preprocessed to get the features $\widetilde{x}$ defined as $\tilde{x} := (x, -\alpha \odot x)^T \in \mathbb{R}^{2d}$. The vector $\widetilde{x}$ is given as the input to the neural network.

We construct a three layer neural network with input $\widetilde{x} \in \mathbb{R}^{2d}$, and weight matrices $W_1 \in \mathbb{R}^{2d \times 2d}$, $W_2 \in \mathbb{R}^{2d \times d}$ and $W_3 \in \mathbb{R}^{d \times 1}$ for the first layer, the second layer and the third layer respectively. The activation function after the $i$th layer is given by $\sigma_i : \mathbb{R} \mapsto \mathbb{R}$. The neural network consists of $d$ trainable parameters, given by $w \in \mathbb{R}^d$, and is denoted by the function $h(w; \widetilde{x}) : \mathbb{R}^d \times \mathbb{R}^{2d} \mapsto \mathbb{R}$. In the following, we describe the corresponding weight matrices and activation functions for each layer of the network:

- **Layer 1:** Input: $\widetilde{x} \in \mathbb{R}^{2d}$. The weight matrix $W_1 : \mathbb{R}^{2d \times 2d}$ is given by

$$W_1[i, j] := \begin{cases} w[j] & \text{if } 1 \leq i = j \leq d \\ 1 & \text{if } d < i = j \leq 2d \\ 0 & \text{otherwise} \end{cases}.$$

  The activation function $\sigma_1$ is given by $\sigma_1(a) = a$. Let $h_1(w; \widetilde{x}) := \sigma_1(\widetilde{x}W_1)$ denote the output of this layer. We thus have that $h_1(w; \widetilde{x}) = (w \odot x, -\alpha \odot x)^T$.

- **Layer 2:** Input: $h_1(w; \widetilde{x}) \in \mathbb{R}^{2d}$. The weight matrix $W_2 : \mathbb{R}^{2d \times d}$ is given by

$$W_1[i, j] := \begin{cases} 1 & \text{if } i - j \in \{0, d\} \\ 0 & \text{otherwise} \end{cases}$$

  The activation function $\sigma_2$ is given by $\sigma_2(a) = a^2$. Let $h_2(w; \widetilde{x}) := \sigma_2(h_1(\widetilde{z}, w)W_2)$ denote the output of this layer. We thus have that $h_2(w; \widetilde{x})[j] = (w[j] \odot x[j] - \alpha[j] \odot x[j])^2$ for any $j \in [d]$.

- **Layer 2:** Input: $h_2(w; \widetilde{x}) \in \mathbb{R}^d$. The weight matrix $W_3 : \mathbb{R}^{d \times 1}$ is given by the vector $\mathbf{1}_d = (1, \ldots, 1)^T$.

  The activation function $\sigma_2$ is given by $\sigma_2(a) = \sqrt{a}$. Let $h_3(w; \widetilde{x}) := \sigma_3(h_2(w; \widetilde{x})W_3)$ denote the output of this layer. We thus have that $h_3(w; \widetilde{x})[j] = \|w \odot x - \alpha \odot x\|$.

Thus, the output of the neural network is

$$\begin{aligned} h(w; \widetilde{x}) &= h_3(w; \widetilde{x}) \\ &= \sigma_3(\sigma_2(\sigma_1(\widetilde{x}W_1)W_2)W_3) = \|(w - \alpha) \odot x\|. \end{aligned}$$

Except for the first $d$ diagonal elements of $W_1$, all the other weights of the network are kept fixed during training. Thus, the above construction represents a restricted neural network with trainable

parameters given by $w$. Also note that in this first layer, any input node connects with a single node in the second layer. Such networks are known as diagonal neural networks [Gunasekar et al., 2018b].

We assume that the network is trained using linear loss function, i.e. for the prediction $h(w; \widetilde{x})$ for data point $(\widetilde{x}, y)$, the loss is given by

$$\ell(w; \widetilde{x}) = y \cdot h(w; \widetilde{x}) = y\|(w - \alpha) \odot x\|, \tag{75}$$

Note that the above expression exactly represents $f_{(A)}(w; z)$. This suggests that learning with the loss function $f_{(A)}$ is equivalent to learning with the neural network $h$ (defined above with trainable parameter given by $w$) with linear loss. Furthermore, the network $h$ has a constant depth, and $O(d)$ units, proving the desired claim. □

We next show how to express the loss function $f_{(B)}$ using a neural network.

**Proposition 4.** *Function $f_{(B)}$ in Equation ($B$) can be represented as a restricted diagonal neural network with square and square root activation functions with $O(d)$ units and constant depth.*

**Proof of Proposition 4 .** In the following, we will assume that before passing to the neural network, each data sample $z = (x, \alpha)$ is preprocessed to get the features $\widetilde{x} := (x, -\alpha \odot x, \mathbf{1}_d, -\alpha, \mathbf{1}_d, \mathbf{1}_d)^T \in \mathbb{R}^{6d}$. The vector $\widetilde{x}$ is given as the input to the neural network.

We construct a four layer neural neural network with input $\widetilde{x} \in \mathbb{R}^{6d}$ and weight matrices $W_1 \in \mathbb{R}^{6d \times 6d}$, $W_2 \in \mathbb{R}^{6d \times 3d}$, $W_3 \in \mathbb{R}^{3d \times 2}$, $W_4 \in \mathbb{R}^{2 \times 1}$ for the four layers respectively. The activation function after the $i$th layer is given by $\sigma_i : \mathbb{R} \mapsto \mathbb{R}$. The neural network consists of $d$ trainable parameters, given by $w \in \mathbb{R}^d$, and is denoted by the function $h(w; \widetilde{x}) : \mathbb{R}^d \times \mathbb{R}^{6d} \mapsto \mathbb{R}$. In the following, we describe the corresponding weight matrices and activation functions for each layer of the network:

- **Layer 1:** Input: $\widetilde{x} \in \mathbb{R}^{4d}$. The weight matrix $W_1 : \mathbb{R}^{6d \times 6d}$ is given by

$$W_1[i, j] := \begin{cases} w[j] & \text{if } i = j \text{ and } j - \alpha d \leq d \text{ for } \alpha \in \{0, 2, 4\} \\ 1 & \text{if } i = j \text{ and } j - \alpha d \leq d \text{ for } \alpha \in \{1, 3, 5\} \\ 0 & \text{otherwise} \end{cases} .$$

  The activation function $\sigma_1$ is given by $\sigma_1(a) = a$. Let $h_1(w; \widetilde{x}) := \sigma_1(\widetilde{x} W_1)$ denote the output of this layer. We thus have that $h_1(w; \widetilde{x}) = (w \odot x, -\alpha \odot x, w, -\alpha, w, \mathbf{1}_d)^T$.

- **Layer 2:** Input: $h_1(w; \widetilde{x}) \in \mathbb{R}^{6d}$. The weight matrix $W_2 : \mathbb{R}^{6d \times 3d}$ is given by

$$W_1[i, j] := \begin{cases} 1 & \text{if } i - j \in \{0, d\} \text{ and } j \leq d \\ 1 & \text{if } i - j \in \{d, 2d\} \text{ and } d < j \leq 2d \\ 1 & \text{if } i = j + 3d \text{ and } 2d < j \\ 0 & \text{otherwise} \end{cases} .$$

  The activation function $\sigma_2$ is given by $\sigma_2(a) = a^2$. Let $h_2(w; \widetilde{x}) := \sigma_2(h_1(\widetilde{z}, w) W_2)$ denote the output of this layer. We thus have that $h_2(w; \widetilde{x})[j] = (w[j] \odot x[j] - \alpha[j] \odot x[j])^2$ for any $j \in [d]$, $h_2(w; \widetilde{x})[j] = (w[j] - \alpha[j])^2$ for any $d < j \leq 2d$ and $h_2(w; \widetilde{x})[j] = (w[j])^2$ for $2d < j \leq 3d$.

- **Layer 3:** Input: $h_2(w; \widetilde{x}) \in \mathbb{R}^{3d}$. The weight matrix $W_3 : \mathbb{R}^{3d \times 2}$ is given by

$$W_3[i, j] = \begin{cases} \frac{1}{2} & \text{if } j = 1 \text{ and } 1 \leq i \leq d \\ -\frac{c_n}{2} & \text{if } j = 1 \text{ and } d + 1 \leq i \leq 2d \\ 1 & \text{if } j = 2 \text{ and } 2d + 1 \leq i \leq 3d \end{cases} .$$

  For the first node in the output of this layer, we use the activation function $\sigma_2(a) = \sqrt{a}$ and for the second node, we use the activation function $\sigma_2(a) = \max\{1, a^2\}$. Let $h_3(w; \widetilde{x}) := \sigma_3(h_2(w; \widetilde{x}) W_3)$ denote the output of this layer. We thus have that $h_3(w; \widetilde{x})[j] = (\frac{1}{2}\|(w - \alpha) \odot x\|^2 - \frac{c_n}{2}\|(w - \alpha) \odot x\|^2, \max\{1, \|w\|^4\})$.

- **Layer 4:** Input: $h_3(w; \widetilde{x}) \in \mathbb{R}^2$. The weight matrix $W_4 : \mathbb{R}^{2 \times 1}$ is given by $W_4 = (1, 1)$, and the activation function $\sigma_4$ is given by $\sigma_4(a) = a$. Let $h_4(w; \widetilde{x}) := \sigma_3(h_2(w; \widetilde{x})W_3)$ denote the output of this layer. We thus have that $h_4(w; \widetilde{x})[j] = \frac{1}{2}\|(w - \alpha) \odot x\|^2 - \frac{c_n}{2}\|(w - \alpha) \odot x\|^2 + \max\{1, \|w\|^4\}$.

Thus, the output of the neural network is given by

$$
\begin{aligned}
h(w; \widetilde{x}) &= h_4(w; \widetilde{x}) \\
&= \sigma_4(\sigma_3(\sigma_2(\sigma_1(\widetilde{x}W_1)W_2)W_3)W_4) \\
&= \frac{1}{2}\|(w - \alpha) \odot x\|^2 - \frac{c_n}{2}\|(w - \alpha) \odot x\|^2 + \max\{1, \|w\|^4\}.
\end{aligned}
$$

In the above construction, the first layer can be thought of as a convolution with filter weights given by $\mathrm{diag}(w, \mathbf{1}_d)$ and stride $2d$. While training the neural network, we keep all the weights of the network fixed except for the ones that take values from $w$ (in the weight $W_1$). Thus, the above construction represents a restricted CNN with trainable parameters given by $w$.

Furthermore, for the prediction $h(w; \widetilde{x})$ for data point $\widetilde{x}$, we treat the output of the neural network as the loss, which is given by

$$
\begin{aligned}
\ell(w; \widetilde{x}) &= h(w; \widetilde{x}) + \max\{1, \|w\|^4\} \\
&= \frac{1}{2}\|(w - \alpha) \odot x\|^2 - \frac{c_n}{2}\|(w - \alpha) \odot x\|^2 + \max\{1, \|w\|^4\}.
\end{aligned}
$$

Note that the above expression exactly represents $f_{(B)}(w; z)$. This suggests that learning with the loss function $f_{(B)}$ is equivalent to learning with the neural network $h$ (defined above with trainable parameter given by $w$). Furthermore, the network $h$ has a constant depth, and $O(d)$ units, proving the desired claim. $\qquad\square$

We next provide a general representation result which implies that the activation functions $\sigma(a) = a^2$ (square function) and $\widetilde{\sigma}(a) = \sqrt{a}$ (square root function), used in the constructions above, can be approximated both in value and in gradients simultaneously using $\mathrm{poly}(d)$ number of ReLU units.

**Lemma 15.** *Let $f : [a, b] \to \mathbb{R}$ be an $L$-Lipschitz and $\alpha$-smooth function. Then for any $\varepsilon > 0$, there is a function $h : [a, b] \to \mathbb{R}$ that can be written as a linear combination $h$ of $\left\lceil \frac{(b-a)\max\{L, \alpha\}}{\varepsilon} \right\rceil + 1$ ReLUs with coefficients bounded by $2L$ such that for all $x \in [a, b]$, we have $|f(x) - h(x)| \le \varepsilon$, and if $h$ is differentiable at $x$, then $|f'(x) - h'(x)| \le \varepsilon$.*

**Proof.** Consider dividing up the interval $[a, b]$ into equal $n$ equal sized intervals of length $\delta = \frac{b-a}{n}$, for $n = \left\lceil \frac{(b-a)\max\{L, \alpha\}}{\varepsilon} \right\rceil$. Define $a_i = a + i\delta$ for $i = 0, 1, \ldots, n$. Let $h$ be a piecewise linear function that interpolates $f$ on the $n+1$ endpoints of the intervals. For any such interval $[a_i, a_{i+1}]$, by the mean value theorem, the slope of $h$ on the interval is equal to $f'(x_i)$ for some $x_i \in (a_i, a_{i+1})$, and hence is bounded by $L$ since $f$ is $L$-Lipschitz. Furthermore, by the $L$-Lipschitzness and $\alpha$-smoothness of $f$, for any $x \in (a_i, a_{i+1})$, we have

$$
|f(x) - h(x)| \le \delta L \le \varepsilon
$$

and

$$
|f'(x) - h'(x)| = |f'(x) - h'(x_i)| = |f'(x) - f'(x_i)| \le \delta\alpha \le \varepsilon.
$$

Now, by Lemma 16 we can represent $h$ as a linear combination of $n+1$ ReLUs with coefficients bounded by $2L$. $\qquad\square$

**Lemma 16.** *Any piecewise linear function $f : \mathbb{R} \to \mathbb{R}$ with $K$ segments can be written as a linear combination of $K + 2$ ReLUs with coefficients bounded by twice the maximum slope of any segment of $f$.*

**Proof.** A piecewise linear function $f : \mathbb{R} \to \mathbb{R}$ is fully determined by the endpoints $a_1 < a_2 < \cdots < a_K$ for some positive integer $K$ which define the segments of $f$, the value $f(a_1)$, and the slopes

$m_0, m_1, \ldots, m_K \in \mathbb{R}$, such that the slope of $f$ on the segment $(a_i, a_{i+1})$ is $m_i$, where we define $a_0 := -\infty$ and $a_{K+1} = +\infty$ for convenience. Specifically, we can write $f$ as the following:

$$f(x) = f(a_1) + \begin{cases} m_0(x - a_1) & \text{if } x < a_1 \\ \sum_{i=1}^{\ell-1} m_i(a_{i+1} - a_i) + m_\ell(x - a_\ell) & \text{if } x \in [a_\ell, a_{\ell+1}) \end{cases}$$

Now define $\sigma(x) = \max\{x, 0\}$ to be the ReLU function, and consider the function $h$ defined as

$$h(x) = f(a_1) - m_0\sigma(a_1 - x) + m_0\sigma(x - a_1) + \sum_{i=1}^{K}(m_i - m_{i-1})\sigma(x - a_i).$$

Since this is a linear combination of ReLUs, it is a piecewise linear function. By direct calculation, one can check that $h(a_1) = f(a_1)$, the endpoints of the segments of $h$ are $a_1, a_2, \ldots, a_K$, and the slopes of the segments are $m_0, m_1, \ldots, m_K$. Hence, the function $f = h$. $\qquad\square$