# OpenReview forum: "SGD: The Role of Implicit Regularization, Batch-size and Multiple-epochs"
_NeurIPS.cc/2021/Conference — NeurIPS 2021 Poster_

### Official Review · Reviewer_Esyv · 2021-07-12

**Rating:** 7
**Confidence:** 4

**Summary:**

The paper considers the problem of stochastic convex optimization where the population loss is convex, while the empirical loss is not necessarily convex. Specifically, this work examines SGD,GD and RERM under this setting.

The first result demonstrates a separation between SGD and RERM, in contrast to the more common SCO problem. It is well known that SGD ,given $n$ samples, obtains an $O(1/\sqrt{n})$-optimal solution (in population risk) after one pass over the dataset. They provide a lower bound of $\Omega(1)$ for RERM w.r.t any sample independent regularizer and any regularization parameter(possibly sample dependent).
The second result extends that of Amir et al. [2021] (in the slightly different setting of SCO), and they provide a lower bound in terms of sample complexity for GD of - $\tilde\Omega(1/n^{5/12})$.
The authors also provide an instance of SCO problem in which $k$-multi-pass SGD achieves a $O(1/\sqrt{nk})$ excess population risk, while RERM attains a lower bound of $\Omega(1)$.
Lastly, they provide a separation between RERM and SGD in the distribution free model, where SGD outperforms RERM (for any regularizer) for distributions where the population loss is convex - SGD upper bound of $O(1/\sqrt{n})$ vs RERM lower bound of $\Omega(1/n^{1/8})$. And also for distributions where the population loss is $\text{\emph{not}}$ convex - SGD upper bound of $O(1/n^{1/8})$ vs RERM lower bound of $\Omega(1/n^{1/8})$.

**Limitations And Societal Impact:**

There are no foreseen societal implications.

**Main Review:**

This paper is well-written and clearly explains the problem, the lower bounds, and the ideas behind the proofs. This paper affirms that SGD is indeed superior to GD and also surprisingly to RERM for any regularizer, in the specific regime this paper considers. This also substantiates the claim that implicit regularization is not the reason for the generalization properties of SGD.

Though some of the constructions rely on previous work(Amir et al [2021], Nesterov [2014]), I find the necessary extensions as novel and non-trivial.

One comment I have, in Theorem 7 you claim that $w^\star$ is in the unit ball. However, based on the construction this seems to be incorrect as for $w\in\mathcal{R}^d$ we have $\min_{w\in\mathcal{R}^d}F(w)\rightarrow -\infty$. Please address this issue in the rebuttal, as it might have implications on the following Theorem 8, Theorem 9 and ultimately Theorem 3.

-----Response-----
The concerns raised over the boundness of $F(w)$ were properly answered. As far as I understand, the point that the global minimizer is in the unit balI was not mentioned in the paper of Amir et al, so I would suggest addressing this point in the proof.

**Time Spent Reviewing:**

10

---

> ### Author Response · Authors · 2021-08-10
> **Response to your concern about boundedness of F(w)**
>
> Thank you for your time and review. Please find below our response to your concerns.
>
>
>    &nbsp;
>
>
> ***In Theorem 7 you claim that  $w^\star$ is in the unit ball..."***
>
>    The lower bound function construction in Amir et al [2021] satisfies that $\| w^* \|$ is in the unit ball and and that $F(w^*) > -1$. Amir et al [2021] explicitly discuss this in the last paragraph of the proof of Lemma 4.1 on page 28. We also mention this in lines 714-715 in the proof of Theorem 7 in the appendix of our paper, and will add more details to make it more clear. We provide the intuition below.
>
>    &nbsp;
>
>    ***Intuition:*** In the lower bound construction given in equation (16) in Amir et al [2021],  while instance losses $f(w;z)$ can be arbitrarily negative, the population loss $F(w)$ is bounded from below and is such that $\| \|w^* \|\| \leq 1$. The population loss contains a linear function of $w$ (middle term) which can be negative, and due to this one might conclude that $F(w)$ can be made arbitrarily small. However, note that whenever this term is negative, the first term takes a larger positive value (as it is a square root of a quadratic function of $w$). The negative and the positive terms balance each other in the population loss  giving a lower bound on $F(w^*)$.
>
> ***We are happy to provide a detailed technical proof through the openreview response system if the reviewer would like to see it.***

---

### Official Review · Reviewer_QRsG · 2021-07-16

**Rating:** 6
**Confidence:** 3

**Summary:**

The paper discusses the current training approaches of SGD from a theoretical perspective. On the negative side, the paper denies a popular conjecture that SGD has an implicit regularization like regularized ERM. On the positive side, the paper shows that SGD does outperform GD in some SCO problems, and that multiple-pass SGD does help in some SCO problems.

**Limitations And Societal Impact:**

The authors addressed the limitations and potential negative societal impact of their work.

**Main Review:**

The paper is overall well written and technically comprehensive. The result comparing SGD and RERM somewhat addresses a popular "misbelief" that SGD works because of implicit regularization and I think this information is worthy conveying. Similar for other results.

My major concern about the results is that the constructed SCO problems look so "manufactural" that it might not be able to reflect reality. For example, the dimension of the constructed negative example in Theorem 1 is about $\mathcal{O}(2^n)$. This brings two disadvantages: 1. In practice no dimension is in the scale of $\mathcal{O}(2^n)$, so the counter-example might not be enough to deny the possibly of SGD having implicit regularization; 2. Due to such a high dimension, the proof is essentially finding a dimension that "overfits" the finite data in the ERM setting but fails to fit the ground truth (which can be seen as a population setting). So actually in this case the difference of SGD and RERM is caused by the difference of population risk and ERM, and thus seems not related to SGD itself. Please correct me if I misunderstand something. I understand the theoretical flavor of this paper, but I'm just wondering whether the result can be improved to be more practical.

**Time Spent Reviewing:**

8 hours

---

> ### Author Response · Authors · 2021-08-10
> **Response to your concern about high dimensionality of our lower bound constructions**
>
> Thank you for your review. *To keep the response short, we did not add the proofs here but we are happy to follow up with complete technical details for any of the following using the openreview response system if the reviewer would like to see it.*
>
> &nbsp;
>
>
> ***Getting our results with d = O(n)***
>
> For the sake of cleaner presentation, we chose to use the simpler construction with $d = 2^n$. However, we can show similar separations in low dimensional settings as well, as discussed in lines 221-228 in the paper.
>
> &nbsp;
>
> Firstly, even in the current construction, for any dimension $d$, we can modify the data distributions to get a lower bound of $\sqrt{{\log(d)}/{n}}$ for RERM (which is the uniform convergence rate) while SGD always attains a $\sqrt{1/n}$ rate. When $d = n$, this leads to a lower bound of $\sqrt{{\log(n)}/{n}}$ for RERM which still shows a strict separation between SGD and RERM alebit only a $\sqrt{\log n}$ factor. We chose $d = 2^n$ so the separation is drastic in that SGD learns and RERM does not.
>
> &nbsp;
>
> Secondly, combining our ideas with the construction from Feldman [2016], we can construct an SCO instance with  $d = O(n)$ for which the exact lower bound in Theorem 2 holds. The construction in Feldman [2016] is based on error correcting codes, and is a bit more complicated to present (which is why we went with the easier route of setting $d = 2^n$). The key idea for the lower bound with $d = O(n)$ is to first modify their loss function construction by multiplying it with an independent variable $y \in \{-1, 1\}$  which takes the value $+1$ with probability $0.6$, as we currently do in Section 3. Next, adding a translation to the function we can effectively show the same result but with $d = O(n)$.  The main idea behind the proof for such construction is exactly the same and the proofs are similar. Since the original construction in Feldman [2016] is more involved, we simply chose to do the simpler construction to demonstrate the key ideas and only remarked about this.
>
> &nbsp;
>
>
> We will add a more detailed discussion on this in the final version of the paper and if the reviewer deems it also add the detailed proof in the appendix.

---

### Official Review · Reviewer_5qGy · 2021-07-17

**Rating:** 7
**Confidence:** 4

**Summary:**

This paper considers the sample complexity comparison between one-pass SGD vs. ERM/RERM/GD/multi-pass SGD in the setting of SCO (with Lipschitz, finite $\ell_2$-norm solution, bounded gradient variance). It shows that, for any problem instances in the considered SCO class, one-pass SGD always achieves good, $1/\sqrt{n}$ rate; in contrast, ERM, RERM (that is distribution-agnostic but is allowed to tune a multiplicative hyperparameter), GD (with any number of passes) could be arbitrarily bad in the worse-case instance in the considered problem class. Moreover, when used with a hold-out validation set, multi-pass SGD can outperform (at least no worse than) one-pass SGD. In conclusion, the paper reveals an important message, that (one-pass) SGD does have its own implicit bias, but it cannot be explained by a simple explicit regularizer.

**Limitations And Societal Impact:**

See above.

**Main Review:**

# Pros:
+ The paper is written very well!
+ A systematic (worse-case) sample complexity comparison between one-pass SGD vs. ERM/RERM/GD/multi-pass SGD in the setting of SCO problem class is conducted. The presented results are quite inspiring for understanding the implicit bias of (single-pass) SGD.

# Cons:
- The considered multi-pass SGD is with a hold-out validation set. However the considered GD does not utilize a similar setup. This is not fair  to the statement for GD. With that being said, I think the authors have done a good job of making things clear & the part of multi-pass SGD is also inspiring in its own.
- Some dimension factors are hidden in the constants in Assumption II. Since the considered dimension is large, this could limit the applicability of the presented results.

# Typos:
* l.201. ... is simple. (, -> .)
* l.327. imposes -> impose. need -> needs.

# Overall
I had the fortune to be one of the reviewers of (an old version of) this paper in another conference (where we had a very long discussion weighting the spirits and shortcomings). Compared with the old version that I had read (which already shows interesting spirits but has some shortcomings), the current version is way more clear and complete. Almost all the issues that I used to have are resolved in this version. Though there are a few (less-critical) points that can be improved, I am quite happy to score this version an acceptance.



------

Post-rebutal:  my issues are largely resolved after reading the authors reply. After reading the other reviewers comments, I will maintain my initial score and recommend to accept this paper.

**Time Spent Reviewing:**

4

---

> ### Author Response · Authors · 2021-08-10
> **Response to your concerns under "Cons" section**
>
> Thank you for a detailed review and your positive support. Please find below our response to your concerns.
>
> * ***"GD does not utilize a similar hold-out validation setup..."***
>    1. We show in Theorem 3 that GD algorithm when run with any step size and for any number of time steps can not match the performance of single pass SGD. The proof technique actually works even if the the step size or number of steps is chosen in a data dependent fashion. Hence, hold-out validation does not help GD. We will add more details on this in the final version.
>
>    &nbsp;
>
>    2. On the other hand, we show that with hold-out validation, multi-pass SGD is at least as good as single-pass SGD and could be much better than single-pass SGD for some problems.
>
>    &nbsp;
>
>    3. However, our current results are unable to show that there is a single problem instance for which multi-pass SGD is strictly better than single-pass SGD which is strictly better than GD with hold-out validation simultaneously. Explicitly showing this is a very interesting research direction.
>
>
> &nbsp;
>
> * ***"Some dimension factors are hidden in the constants in Assumption II...."***
>
>    We explicitly consider SCO problems for which $F$ is $L$-Lipschitz, $\|\| w - w^* \|\| \leq B$ and the variance in gradients is bounded by $\sigma^2$, where $L, B$, and $\sigma$ are independent of the dimension $d$. Furthermore, there are no additional dimension dependent factors hidden in the constants in any of our big-O or big-$\Omega$ notations.
>
>    &nbsp;
>
>    The assumption that $L, B$ and $\sigma$ are independent of the dimension $d$ is standard in stochastic optimization literature, and such SCO problems are easy to construct eg. SVMs, logistic regression, and kernel methods. In fact when one considers kernel methods the dimension can be infinite but typically $L$, $B$ and $\sigma$ are still finite.

---

### Official Review · Reviewer_7fEN · 2021-07-19

**Rating:** 6
**Confidence:** 4

**Summary:**

This paper studies the generalization error bound of multi-epoch, small-batch, SGD for learning over-parameterized models. In particular, the authors compare the generalization performance of SGD to GD and regularized empirical risk minimization (RERM). For RERM, the authors show that there is an SCO problem for which RERM will not give vanishing generalization error while SGD enjoys a $O(1/\sqrt{n})$ rate. For GD, the authors also show that for some SCO problems GD with any step size and number of iteration can only achieve a $\Omega(1/n^{5/12})$ rate, which is also worse than SGD. Lastly, the authors show that multi-epoch SGD can perform better than single-pass SGD.


**Limitations And Societal Impact:**

There is no potential negative societal impact.

**Main Review:**

Generally speaking, this paper is well written and the comparison to RERM and GD verifies the effectiveness of SGD. Detailed comments are as follows:

The claim that “this automatically rules out any implicit regularization based explanation for the success of SGD” is a bit overstated because the comparison of generalization error between SGD and RERM in this paper is only performed for a small class of problem instances. It is likely that given more conditions on the problem instance there could be a connection between RERM and SGD. For example, as shown in [1], for the least square problem, SGD (with iterate averaging) has a similar generalization performance compared to ridge regression solution in terms of the generalization error bound.

Following the previous comment, this paper only provides one side comparison between SGD and RERM/GD, i.e., only shows that for one single problem instance RERM/GD performs worse than SGD. Given this, it is also likely that for some problems RERM/GD can perform much better than SGD. A discussion on this opposite side may also need to be included.

Regarding the multi-epoch SGD algorithm, why do you only calculate the average of the first $nj/2$ iterates rather than all of them? Besides, the first $nj/2$ iterates are generated using different learning rates, would it be better to take this into consideration and use a weighted average?

I did not check the proof but I guess Proposition 1 requires that the loss function on $S_2$ can well concentrate around its expectation. But this may require the loss function is subgaussian or upper bounded. However, this paper only assumes that the loss function is Lipschitz rather than bounded.

Reference:

[1] Zou et. al., Benign overfitting of constant-stepsize sgd for linear regression, arXiv 2021


==================

I have read the authors' responses. Overall this paper is good and I keep my recommendation of acceptance.

**Time Spent Reviewing:**

4 hrs

---

> ### Author Response · Authors · 2021-08-10
> **Response to all your questions**
>
> Thank you for a detailed review. Please find resolutions for all your concerns below. If the reviewer wishes, we are happy to provide more technical details in a follow up discussion.
>
> &nbsp;
>
> * ***“The claim that “this automatically rules out any implicit regularization bias is a bit overstates.... ”***
>
>    We agree that adding more structure to the problem can make SGD solution coincide with that of an implicit regularized one. The example the reviewer mentions and many more prior implicit regularization works cited in our paper do exactly this under specific problem structures. We already discuss this in detail in lines 45-66 in the paper. Our goal in this paper was to explore the role of implicit regularization in explaining the success of SGD in a broader context. Specifically, in this paper we show that for SCO problems (the entire class of problems),  the success of SGD cannot always be explained via implicit regularization.
>
> &nbsp;
>
> * ***"It is also likely that for some problems RERM/GD can perform much better than SGD..."***
>
>    We agree that there can always be problem instances for which RERM/GD can outperform SGD. In fact, more generally, for any algorithm, there exists a problem instance where that algorithm will outperform SGD. What one can hope to show to be true theoretically is that for a class of problems, SGD algorithm outperforms RERM/GD, meaning the worst case rate for that class is better for SGD than GD/RERM. We show exactly this for the class of Stochastic Convex Optimization Problems, since SGD always provably achieves $n^{-1/2}$ rate where as RERM and GD are provably worse for some SCO problem.
>
> &nbsp;
>
> * ***Why average of $nj/2$ iterates and step size schemes ?***
>
>    The dataset S is first split into $S_1$ and $S_2$ with $n/2$ samples each, and then the multipass SGD algorithm is run on $S_1$. Since $S_1$ only has $n/2$ samples, the number of iterates generated after taking $j$ rounds is $nj/2$; we return the average of all these iterates.
>
>    &nbsp;
>
>    It is not clear if a different weighting scheme can further improve the performance of multi-pass SGD algorithm. However, this is a very interesting direction for future research.
>
>    &nbsp;
>
> * ***Proposition 1 requires that the loss function on $S_2$  can well concentrate around its expectation.***
>
>    The proof uses the fact that the loss function is bounded over the domain of interest. This is automatically ensured for our multi-pass SGD algorithm due to the projection step in Line 2 in Algorithm 1 (line 1038). We will clarify this in the final version of the paper.

---

### Decision · Program_Chairs · 2021-09-27

**Decision:**

Accept (Poster)

**Comment:**

This paper presents interesting results refining the relationship between SGD and GD, and also implicit regularization.  The reviewers are all in support, and I request the authors carefully consider the reviewers' suggestions in their revisions.